# TMS-evoked responses are driven by recurrent large-scale network dynamics

**Davide Momi[1]\*[†], Zheng Wang[1][†], John D Griffiths[1,2,3]\***

[1]Krembil Centre for Neuroinformatics, Centre for Addiction and Mental Health, Toronto, Canada; [2]Department of Psychiatry, University of Toronto, Toronto, Canada; [3]Institute of Medical Sciences, University of Toronto, Toronto, Canada

**Abstract** A compelling way to disentangle the complexity of the brain is to measure the effects of spatially and temporally synchronized systematic perturbations. In humans, this can be non-invasively achieved by combining transcranial magnetic stimulation (TMS) and electro-encephalography (EEG). Spatiotemporally complex and long-lasting TMS-EEG evoked potential (TEP) waveforms are believed to result from recurrent, re-entrant activity that propagates broadly across multiple cortical and subcortical regions, dispersing from and later re-converging on, the primary stimulation site. However, if we loosely understand the TEP of a TMS-stimulated region as the impulse response function of a noisy underdamped harmonic oscillator, then multiple later activity components (waveform peaks) should be expected even for an isolated network node in the complete absence of recurrent inputs. Thus emerges a critically important question for basic and clinical research on human brain dynamics: what parts of the TEP are due to purely local dynamics, what parts are due to reverberant, re-entrant network activity, and how can we distinguish between the two? To disentangle this, we used source-localized TMS-EEG analyses and whole-brain connectome-based computational modelling. Results indicated that recurrent network feedback begins to drive TEP responses from 100 ms post-stimulation, with earlier TEP components being attributable to local reverberatory activity within the stimulated region. Subject-specific estimation of neurophysiological parameters additionally indicated an important role for inhibitory GABAergic neural populations in scaling cortical excitability levels, as reflected in TEP waveform characteristics. The novel discoveries and new software technologies introduced here should be of broad utility in basic and clinical neuroscience research.

**\*For correspondence:**
momi.davide89@gmail.com
(DM);
john.griffiths@utoronto.ca (JDG)

[†]These authors contributed
equally to this work

**Competing interest:** The authors
declare that no competing
interests exist.

**Reviewing Editor:** Alex Fornito,
Monash University, Australia

## Editor's evaluation

This important work advances our understanding of the effects of focal perturbations with tran-scranial magnetic stimulation (TMS) on brain activity. By combining TMS, electroencephalography (EEG), and computational modelling, the authors provide solid evidence to indicate that early EEG signal changes result from local dynamics in the stimulated region whereas later signal changes are influenced by reverberating activity within more broadly connected networks. The work will be of interest to people researching the physiological effects of brain stimulation and biophysical models of large-scale neuronal activity.

## Introduction

The brain is a complex, nonlinear, multiscale, and intricately interconnected physical system, whose laws of motion and principles of organization have proven challenging to understand with currently available measurement techniques (*Bullmore and Sporns, 2009*). In such epistemic circumstances, the application of systematic perturbations, and measurement of their effects, is a central tool in the

scientific armory (*Deco et al., 2018*; *Freeman, 1975*). For human brains, the technological combination that best supports this non-invasive perturbation-based modus operandi is concurrent TMS and EEG (*Rogasch and Fitzgerald, 2013*; *Siebner et al., 2022*). TMS-EEG allows millisecond-level tracking of stimulation-evoked activity propagation throughout the brain (*Momi et al., 2021a*; *Thut and Miniussi, 2009*), originating from a target region that is perturbed by the secondary electrical currents of a focal (2–2.5 cm diameter), brief (1 ms), and powerful (1.5–2 T) magnetic field (*Hallett, 2007*). Trial-averaged TMS-EEG response waveforms (known as TMS-evoked potentials or TEPs) have been used to elucidate basic neurophysiology in the areas of brain connectivity (*Momi et al., 2021c*), axonal conduction delays (*Bortoletto et al., 2021*), and neural plasticity (*Chung et al., 2017*) and also as a clinical biomarker for several pathological conditions such as coma (*Rosanova et al., 2012*), stroke (*Borich et al., 2016*), depression (*Voineskos et al., 2021*), obsessive-compulsive disorder (*Cheng et al., 2022*), and schizophrenia (*Cao et al., 2021*). In addition to this wide variety of basic physiological and clinical applications, TEP measurements speak directly to a central theoretical question at the very heart of systems neuroscience: to what extent does stimulus-evoked neural activity propagate through the brain, via local and/or long-range projections, to affect activity in spatially distant brain regions? In the present paper, we are concerned with this question, and even more so with its equally interesting corollary: to what extent does stimulus-evoked activity propagate *back* from downstream areas to a primary stimulation site? This phenomenon of top-down or cyclic feedback within large-scale brain networks is known as *re-entry* or *recurrence* (*Edelman and Gally, 2013*; *Lau and Bi, 2005*; *Llinás et al., 1998*; *Lopes et al., 2021*).

Understanding the contribution of recurrent activity to TEPs, and stimulus-evoked activity in general, is critically important for proper interpretation of TMS-EEG experimental results and the design of clinical interventions. In the case of TMS, the direct physical and physiological effects at the primary stimulation site of an extracranially-applied magnetic perturbation are reasonably well-understood: secondary electrical currents initially depolarize the membranes of cells in the superficial neural tissue underneath the coil, causing action potentials and an associated local response in the stimulated brain region (*Pascual-Leone and Meador, 1998*). Concurrently, this local electrical activation propagates (as some combination of soma-originating and prodromic axon-originating action potentials) along white matter pathways to reach distant cortical and subcortical sites, resulting in predominantly excitatory effects with magnitudes depending on the strength of the anatomical connections (*Momi et al., 2021b*). The final EEG-measurable outcomes of this process appear as early (<100 ms) and late (>100 ms) responses at both the primary stimulation site and a broad set of interconnected brain regions, usually persisting for 300 ms, and showing reliable characteristic patterns but also high levels of inter-subject variability (*Rocchi et al., 2018*). A key challenge in interpreting these data is that it is impossible from the TMS-evoked EEG time series alone to disentangle whether TEP waveform components at the primary stimulation sites arise due to a 'local echo' - driven only by the recent history of that region, or to a 'global echo' - driven by the recurrent activity within the rest of network.

Here, we introduce a novel approach to addressing these questions around the physiological basis and spatiotemporal network dynamics of neural activity evoked by noninvasive brain stimulation, using a combination of empirical TMS-EEG data analyses and whole-brain, connectome-based neurophysiological modelling. An overview and conceptual framework are given in *Figure 1*. The logic proceeds as follows: In the first step, we fit a connectome-based model to individual-subject TEP data, achieving accurate replication of the measured channel- and source-level TMS-EEG patterns. Then, we introduce to the model a series of spatially and temporally specific 'virtual lesions' by setting to zero the weights of all connections leaving from and returning to the primary stimulation site, at specific times. These virtual lesions isolate the TMS-stimulated region from the rest of the brain for delineated periods, and allow us to ask what its dynamics would look like with and without recurrent feedback from downstream brain areas. Activity patterns at the stimulated node that are unchanged by a given virtual lesion that suppresses recurrent inputs are thus independent of those inputs, and can be understood as a 'local echo' of the stimulation that persists in time for long periods (dozens to hundreds of milliseconds). This framing implies two contrasting potential scenarios for the change in TEP waveform components after introducing a lesion that suppresses recurrent feedback to the stimulation site:

a. TEP components are still observed and entirely or largely unchanged, or
b. TEP components are substantially reduced or disappear

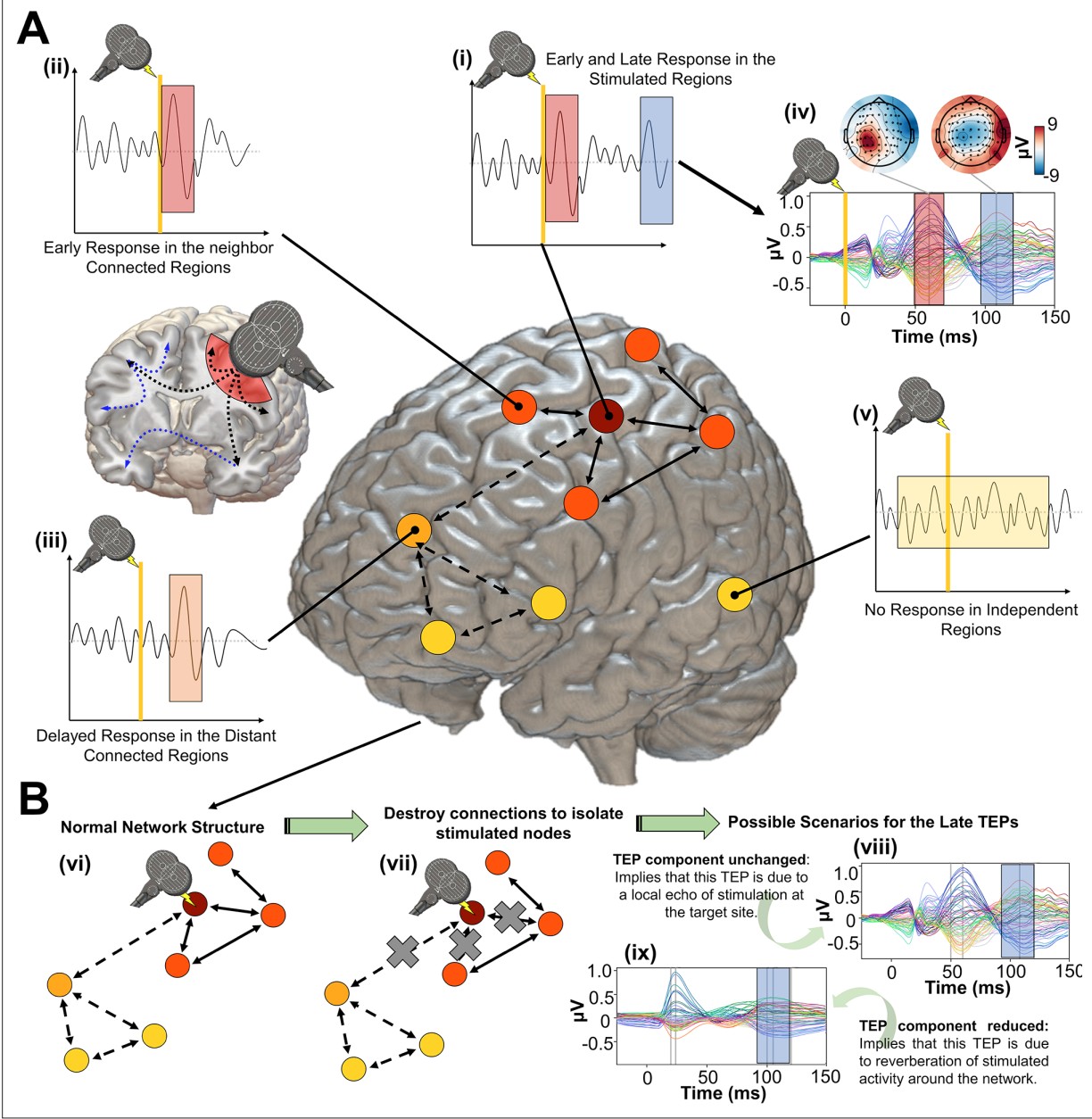

**Figure 1.** Studying the role of recurrent activity in stimulation-evoked neural responses with computational models. Shown here is a schematic overview of the hypotheses, methodology, and general conceptual framework of the present work. (**A**) Single transcranial magnetic stimulation (TMS) pulse (i - diagram, iv - real data) applied to a target region (in this case left M1) generates an early response (TMS-EEG evoked potential (TEP) waveform component) at electroencephalography (EEG) channels sensitive to that region and its immediate neighbors (ii). This also appears in more distal connected regions such as the frontal lobe (iii) after a short delay due to axonal conduction and polysynaptic transmission. Subsequently, second and sometimes third late TEP components are frequently observed at the target site (i, iv), but not in non-connected regions (v). Our central question is whether these late responses represent evoked oscillatory 'echoes' of the initial stimulation that are entirely locally driven and independent of the rest of the network, or whether they rather reflect a chain of recurrent activations dispersing from and then propagating back to the initial target site via the connectome. (**B**) In order to investigate this, precisely timed communication interruptions or 'virtual lesions' (vii) are introduced into an accurately fitting individual subject computational model of TMS-EEG stimulation responses (vi), and the resulting changes in the propagation pattern (vii) are evaluated. No change in the TEP component of interest (e.g. blue shaded area) would support the 'local echo' scenario (viii), whereas suppressed TEPs (e.g. blue shaded area) would support the 'recurrent activation' scenario (ix).

As noted, clear evidence of (a) would be consistent with these brain responses being simply a 'local echo' of the TMS perturbation, that activates only the stimulated area. In contrast, evidence of (b) would imply the local TEP response requires global network reverberation - recurrent activity propagating out from the stimulated region, via its distal interconnected notes, and back again to evoke or to amplify inflections at specific time points post-stimulation.

For modelling the empirical TMS-EEG TEP data following the investigative line described above, we use a newly-developed numerical simulation approach that draws on recent technical advances in the field of machine learning (*Griffiths et al., 2022*). Our novel modelling methodology allows accurate and robust individual subject-level TEP waveform fitting, allowing us to present here the first ever subject-specific, cortex-wide, connectome-based neurophysiological model of TEP generation. As we show, this allows us to pose and answer questions around both the shared structure and the well-known inter-subject variability of TMS-EEG responses (*Ozdemir et al., 2020*). We examine the general question of recurrent activity in relation to feedforward and feedback connections to primary stimulation targets, as well as to the broader graph topological structure of the anatomical connectome. Inter-subject variation in estimated physiological parameters offers candidate explanations for TEP phenomena in terms of excitatory/inhibitory population parameters that are consistent with known pharmaco-physiological effects (*Premoli et al., 2014*). We argue that this physiologically-based mathematical parameterization of brain stimulation responses offers an important new framework for understanding (and minimizing) inter-subject variability in TMS-EEG recordings for basic scientific and clinical applications.

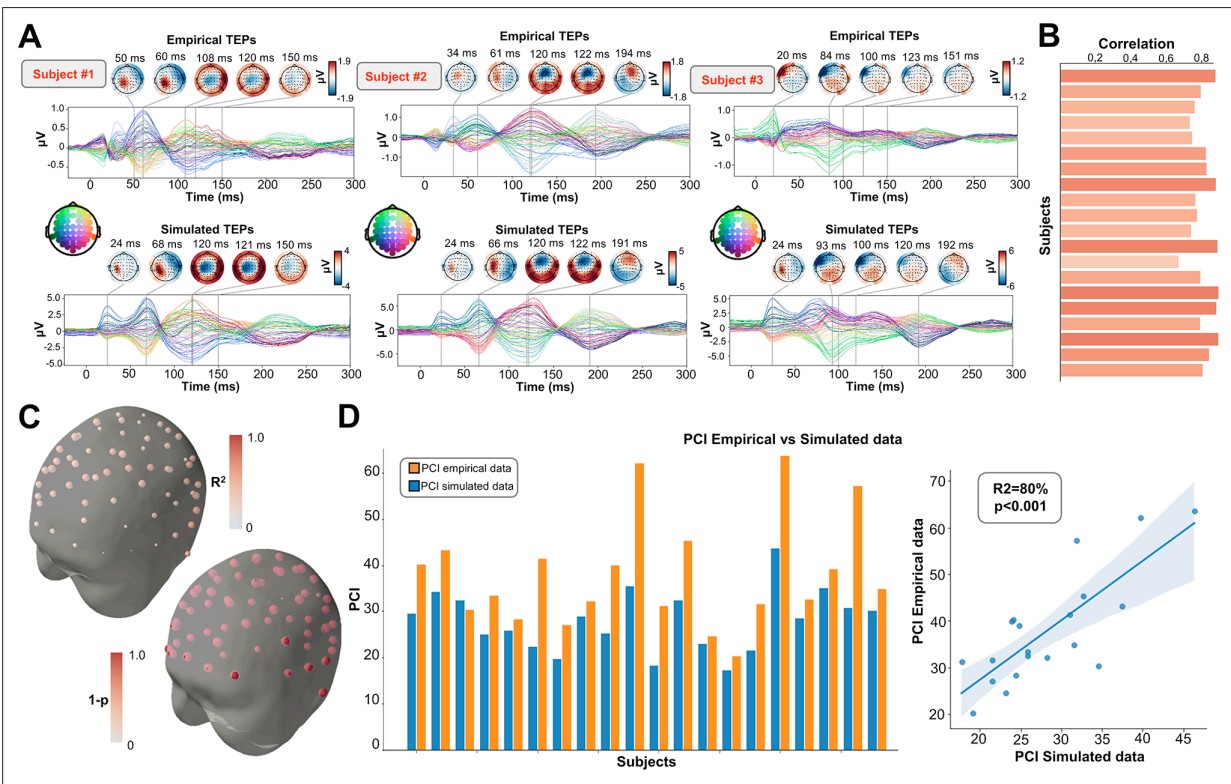

**Figure 2.** Comparison between simulated and empirical TMS-EEG data in channel space. (**A**) Empirical (upper row) and simulated (lower row) TMS-EEG butterfly plots with scalp topographies for three representative subjects, showing a robust recovery of individual empirical TMS-EEG evoked potential (TEP) patterns in model-generated activity electroencephalography (EEG) time series. (**B**) Pearson correlation coefficients between simulated and empirical TMS-EEG time series for each subject. (**C**) Time-wise permutation tests result showing the Pearson correlation coefficient (top) and the corresponding significant reversed p-values (bottom) for every electrode. (**D**) Perturbational complexity index (PCI) values extracted from the empirical (orange) and simulated (blue) TMS-EEG time series (left). A significant positive correlation (R²=80%, p<0.001) was found between the simulated and the empirical PCI (right), demonstrating high correspondence between empirical and simulated data.

## Results

### Connectome-based neurophysiological models accurately reproduce subject-specific TEP patterns

As an important preliminary to our primary research question, extensive testing confirmed that our new connectome-based neurophysiological model of TMS-EEG responses achieves robust and accurate recovery of TEP waveforms at both the group average and individual levels. This is demonstrated in *Figures 2 and 3* for both the EEG channel level and cortical surface source level, respectively. *Figure 2A* shows empirical and fitted (i.e. simulated, with optimized physiological parameters) TEP waveforms and selected topography maps for three example subjects (for the entire group, see *Appendix 2—figure 1*). It is visually evident in these figures that the model accurately captures several individually-varying features of these time series, such as the timing of the 50 ms and 100–120 ms TEP components, and the extent to which they are dominated by left/right and temporal/parietal/frontal channels. (For the latter, this can be seen by comparing the line colors in the upper and lower rows of corresponding columns in *Figure 2A*, and using the channel location references given by the channel color map on the top left of each TEP plot). Pearson correlations between empirical and simulated TMS-EEG time series confirmed that an excellent goodness-of-fit was observed at the whole-head level (*Figure 2B*) and individual channel level (*Figure 2C*), with time-wise permutation tests revealing a significant Pearson correlation coefficient for every electrode. As well as the millisecond-by-millisecond TEP comparisons and the timing of key waveform components, we also assessed the accuracy of the model in capturing holistic time series properties. As shown in *Figure 2D*, a significant positive correlation $R^2=80\%$, $p<0.0001$ was found between the Perturbational Complexity Index (PCI) (*Casali et al., 2013*) of the simulated and empirical waveforms.

Similarly to the single-subject fits, our model also showed accurate recovery of the grand mean empirical TEP waveform when the fitted TEPs were averaged over subjects (*Figure 3A*). These grand mean channel-level waveforms were further used to assess model fit in brain source space. As can be seen in the topoplots in *Figure 3B*, the same spatiotemporal activation pattern is observed both for empirical (top row) and model-generated (bottom row) time series. M1 stimulation begins with activation in left motor area at ~20–30 ms, then propagates to temporal, frontal, and homologous contralateral brain regions, resulting in a waveform peak at ~100–120 ms. Time-wise permutation tests on these data revealed a significant Pearson correlation coefficient in 75.63% of all vertices (*Figure 3C*). Finally, significant positive correlations were found between the dynamic Statistical Parametric Mapping (dSPM) values extracted from the seven canonical Yeo Networks, with stronger correspondences for the primary stimulation target (Somatomotor [SMN], $R^2=46\%$, $p=0.008$) compared to the non-stimulated Networks (Visual [VISN]: $R^2=38\%$, $p=0.01$; Dorsal Attention [DAN]: $R^2=38\%$, $p=0.004$; Anterior Salience [ASN]: $R^2=38\%$, $p=0.003$; Limbic [LIMN]: $R^2=40\%$, $p=0.01$; Fronto-parietal [FPN]: $R^2=41\%$, $p=0.006$; Default Mode [DMN]: $R^2=43\%$, $p=0.009$). This correspondence between empirical and simulated TEP data is clearly visible in the bar charts of *Figure 3D, E, F and G*.

### Later TEP responses are driven by recurrent large-scale network dynamics

Having established the accuracy of our model at replicating TEP waveforms across a wide range of subject-specific shapes (*Appendix 2—figure 1*), we now turn to the central question of the present study, as laid out in *Figure 1B*. Shown in *Figure 4* are effects on the simulated TEP of virtual lesions to recurrent incoming connections of the main activated regions at 20 ms, 50 ms, and 100 ms after single-pulse TMS stimulation of left M1. The leftmost column of *Figure 4*, which shows the simulated data grand average with no virtual lesion (re-plotting the data from the second row of *Figure 3B*), serves as a reference point for the other three columns. A key result here is that there is a reduction of source activation at the 50–100 ms time window in the central two panels (lesions at 20 ms and 50 ms, respectively), as compared to the rightmost (lesion at 100 ms) and leftmost (no lesion) panels. This reduction demonstrates the critical importance of network recurrence in generating later TEP components. These effects were evaluated statistically by extracting dSPM loadings from the source activity maps for each of the 7 canonical Yeo networks (*Thomas Yeo et al., 2011*) and entering them into an ANOVA with factors of NETWORK and TIME OF DAMAGE. Significant main effects were found for both NETWORK ($F_{(6,114)} = 114.73$, $p<0.0001$, $\eta^2_p = 0.85$) and TIME OF DAMAGE ($F_{(3,57)} =$

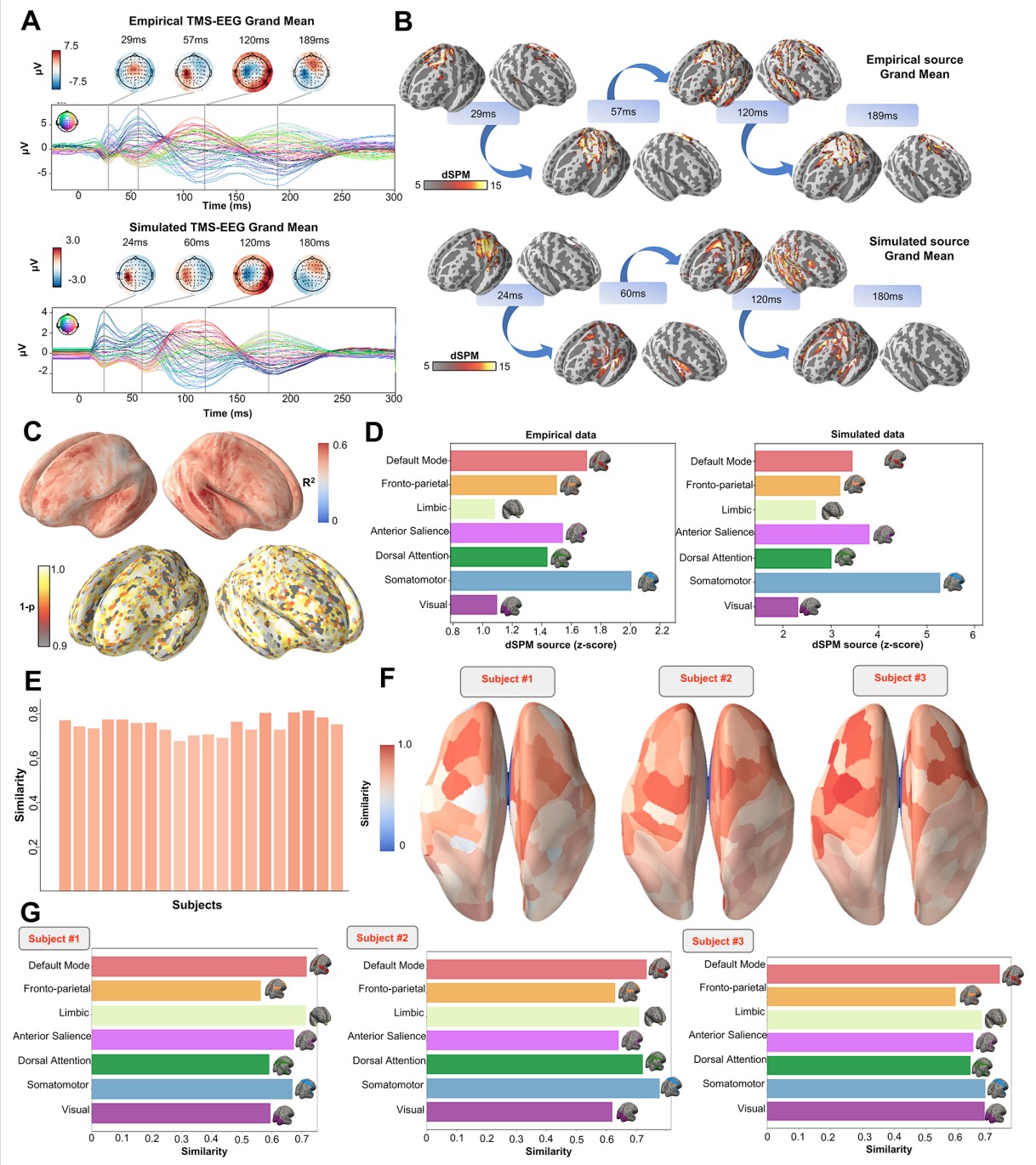

**Figure 3.** Comparison between simulated and empirical TMS-EEG data in source space. (**A**) TMS-EEG time series showing a robust recovery of grand-mean empirical TMS-EEG evoked potential (TEP) patterns in model-generated electroencephalography (EEG) time series. (**B**) Source reconstructed TMS-evoked propagation pattern dynamics for empirical (top) and simulated (bottom) data. (**C**) Time-wise permutation test results showing the significant Pearson correlation coefficient (top) and the corresponding reversed p-values (bottom) for every vertex. (**D**) Network averaged dSPM values extracted for the grand mean empirical (left) and simulated (right) source-reconstructed time series. (**E**) Bar plot showing high vertex-wise cosine similarity between empirical and simulated sources for all the subjects. (**F**) Parcels-wise cosine similarities plotted on surface for three representative subjects, showing a robust recovery of empirical TMS-evoked patterns in model-generated activity EEG time series. (**G**) Network-based cosine similarity for three representative subjects.

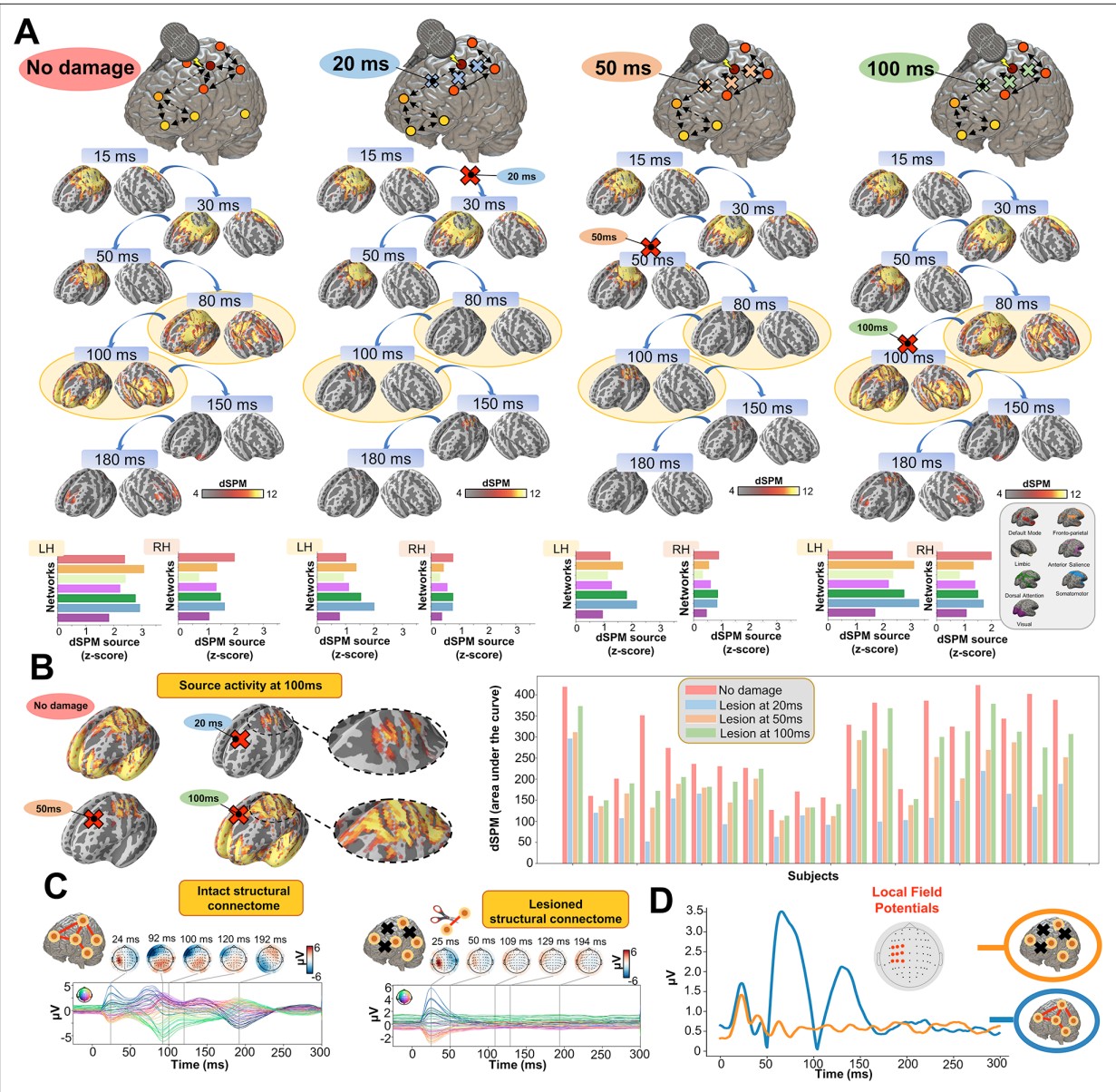

**Figure 4.** Removing recurrent connections from stimulated target nodes suppresses their late TMS-EEG evoked potential (TEP) activity. (**A**) We found that TMS-evoked propagation dynamics in the model change significantly depending on the specific time that a virtual lesion is applied (highlighted orange circle). Specifically, early significant reductions in the TMS-evoked activity (50 ms-100 ms time window) were found when important connections were removed at 20 ms (blue) and 50 ms (orange) after the transcranial magnetic stimulation (TMS) pulse, as compared to both a later virtual lesion (100 ms green) and no damage (red) conditions. These results are demonstrated also for network-based dynamic Statistical Parametric Mapping (dSPM) values (bottom row) extracted for all four conditions. (**B**) Zoom of the source localized electroencephalography (EEG) activities for the four different conditions showing how early lesions (e.g. 20 ms and 50 ms) compromised the propagation of the TMS-induced signal compared to control (no damage) and late lesion (100 ms) conditions (left). For all the subjects, we reported a reduced TMS-evoked activity (dSPM area under the curve) when the lesion was applied at 20 ms and 50 ms compared to the others conditions (right bar plot). (**C**) Demonstration of the network recurrence-based theory for one representative subject. Simulation of TMS-EEG dynamics run using the intact (left) and lesioned (right) anatomical connectome. In the latter case, the connection was removed 50 ms after the TMS pulse and external perturbation generate a local response that reverberates locally and terminates after ~50 ms. This demonstrates again how later TEPs are driven by recurrent network dynamics. (**D**) Local Mean Field Power (LMFP) at the stimulation site for intact (blue line) and lesioned (orange line) anatomical connectome. Red dots inside the scalp map correspond to the electrodes from where the LMFP was extracted.

254.05, p<0.0001, $\eta^2_p$ = 0.93) - indicating that the effects of virtual lesions vary depending on both the time administered and the site administered to, as well as the combination of these factors (significant interaction NETWORK*TIME OF DAMAGE ($F_{(18,342)}$ = 23.79, p<0.0001, $\eta^2_p$ = 0.55)). The specific results described above, with significant TEP suppression at the stimulation site occurring in the early and late TEP components for specific time windows, were verified through extensive post-hoc *t*-tests (see Appendix 2).

## Inhibitory synaptic activity affects inter-subject differences in TEP waveforms

One of the key advantages of physiologically-based brain modelling is the potential for making meaningful associations between major empirical data features and the physiological constructs

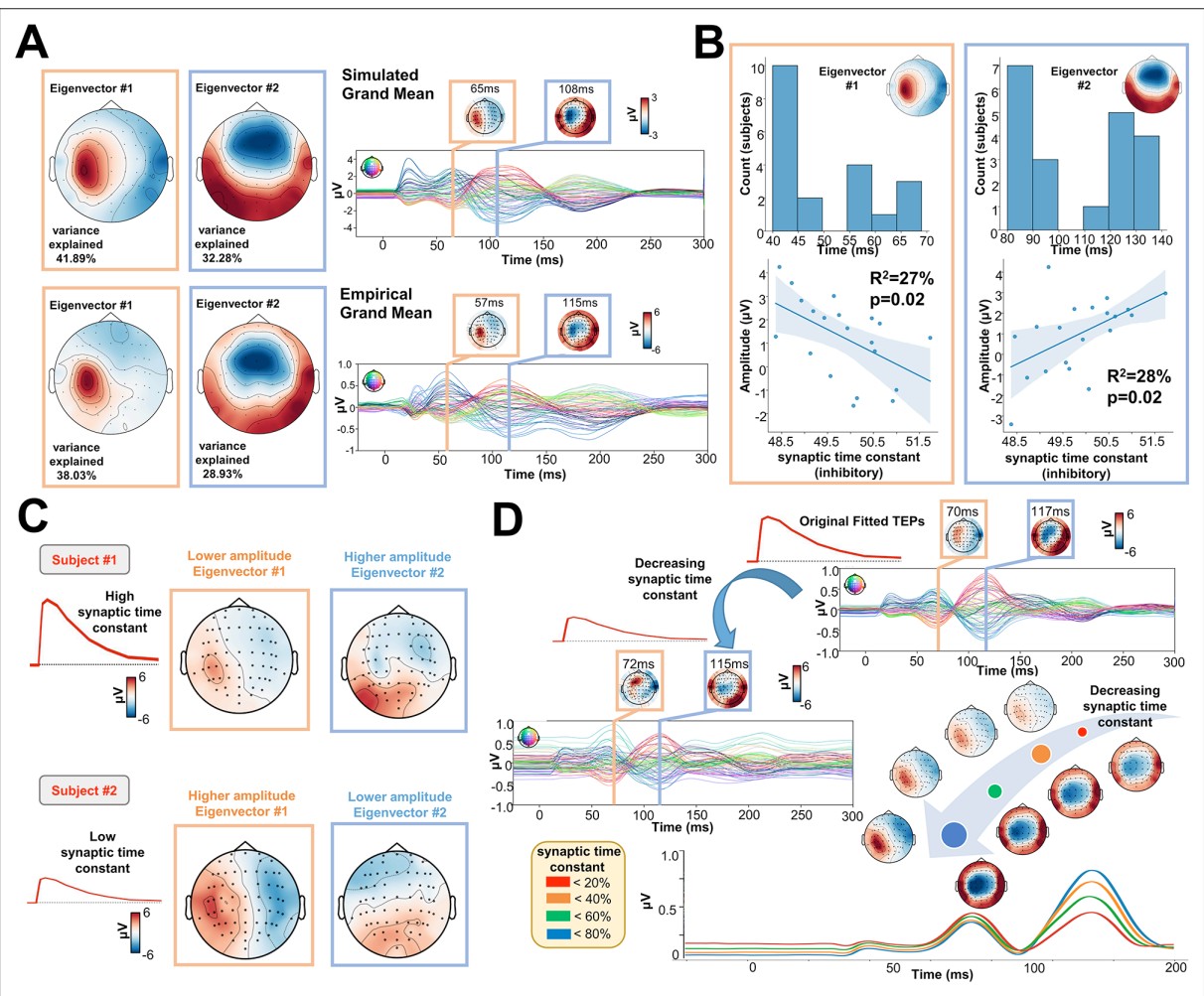

**Figure 5.** Synaptic time constants of inhibitory neural populations affect early and late TMS-EEG evoked potential (TEP) amplitudes. (**A**) Grand mean Singular value decomposition (SVD) topoplots for simulated (top) and empirical (bottom) TMS-EEG data. Results revealed that the first (orange outline) and the second (blue outline) SVD eigenmodes were located ~65 ms and ~110 ms after the transcranial magnetic stimulation (TMS) pulse, respectively. (**B**) First and second SVD temporal eigenmode latencies and amplitudes were extracted for every subject, and the distribution plots (top row) show the time window where the highest cosine similarity with the SVD spatial eigenvectors was found. Scatter plots (bottom row) show a significant negative (left) and positive (right) correlation between the synaptic time constant of the inhibitory population and the amplitude of the first and second eigenvectors. (**C**) Model-generated first and second SVD eigenmodes for two representative subjects with high (top) and low (bottom) estimated values for the synaptic time constant of the inhibitory population. The topoplots show that the magnitude of the synaptic time constant is closely coupled to the amplitude of the individual first and second SVD modes. (**D**) Model-generated TMS-EEG data were run using the optimal (top right) or 85% decreased (central left) value for the synaptic time constant of the inhibitory population. The bottom right panel shows absolute values for different magnitudes of this parameter. Results show an increase in the amplitude of the first, early, and local TEP components; and a decrease of the second, late, and global TEP components, as a function of the inhibitory synaptic time constant.

instantiated in the model's parameters. We explored this by examining the relationship between TEP waveform components and the physiological parameters of the Jansen-Rit model. To do this, singular value decompositions (SVDs) were performed on the channel x time TEP waveform matrices for both empirical and simulated data. The left and right singular vectors from this decomposition, respectively define the temporal and spatial expression of the channel-level TEP eigenmodes. The spatial part of each eigenmode takes the form of a loading pattern over channels that can be represented as a topoplot. As with the TEP waveform and PCI comparisons, this procedure also yielded high spatial similarity between empirical and simulated grand average data (*Figure 5A*), as well as similar levels of variance explained (74.14 and 66.96% cumulatively by the first two right SVD eigenvectors in simulated and empirical data, respectively). Inspecting the temporal peaks in the left singular vectors for the first two eigenmodes revealed that the first was maximally expressed in empirical[/simulated] data at 72 ms[/70 ms], and the second at 115 ms [/117 ms] post-stimulus. Thus the first two eigenvectors of the TEP waveform correspond quite closely to the canonical ~50 ms and ~100 ms TEP waveform components. As shown in *Figure 5B*, a significant negative correlation was found between the synaptic time constant of the Jansen-Rit inhibitory population and the amplitude of the first eigenmode at its peak ($R^2$=27%, p=0.02). Interestingly, we also observed a significant positive correlation between this parameter and the second eigenmode at its peak ($R^2$=28%, p=0.02). Indeed, as shown in *Figure 5C*, the topoplots for 2 representative subjects with high (top) and low (bottom) estimated values for the synaptic time constant of the inhibitory population show that the magnitude of the synaptic time constant is closely coupled to the amplitude of the individual first and second SVD modes. Interestingly, as shown in *Figure 5D*, by varying the optimal value of the inhibitory synaptic time constant, an increase in the amplitude of the first, early, and local TEP components; and a decrease of the second, late, and global TEP components was found. However, these significant results were not corrected for the Bonferroni corrected p-value of 0.007. For this reason, the reader should critically interpret such results.

For a comprehensive overview of individual timing and topographies of the first two eigenmodes, please refer to *Appendix 2—figure 5*. For a detailed description of the significant relationship between empirical data features and model's parameters, please refer to Appendix 2.

## Discussion

Using our novel computational framework for personalized TMS-EEG modelling, in this work we have presented new insights into the role of recurrent activity in stimulation-evoked brain responses. Characterizing these phenomena at a mechanistic level is important not only as a basic question in systems and cognitive neuroscience, but also as a foundation for clinical applications concerned with changes in excitability and connectivity due to neuropathologies or interventions.

We employed a `virtual dissection' approach (*Aerts et al., 2016*) to study the extent to which model-generated TMS-evoked stimulation patterns at the primary stimulation site relied on recurrent incoming connections from the rest of the brain, and at what times. These in-silico interventions resulted in substantial reductions in TMS-evoked activity when pivotal connections were inactivated. Specifically, compared to late (100 ms after the TMS pulse) virtual lesions, and compared to the control condition where no damage was applied, early (20 ms, 50 ms) damage of essential nodes' afferent and efferent pathways significantly reduced the amplitude of the 100 ms TEP component at the stimulation site (left M1) and its neighboring regions (*Figure 4*). In these early lesion conditions, some residual activity in the left M1 area was still observed at around 100 ms, indicating that a local echo of the TMS stimulus does indeed persist for tens to hundreds of milliseconds after stimulation. However, this purely locally-driven activity was low in amplitude, and does not appear to be the principal source of the commonly studied 100 ms TEP components in TMS-EEG recordings. In addition to recurrence at the stimulation site, we can also see that amplification of the TMS-evoked stimulation response occurs via network spreading and recruitment. Early lesions also compromised the propagation of the TMS-evoked activity to the contralateral homolog of the stimulated region (i.e. right M1), as well as bilateral frontal and parietal regions. This result clarifies not only *that* TMS-evoked activity in those regions depends on the presence of those specific cross-hemispheric and parieto-frontal pathways in the network, but also *when* propagation along them is critical for the subsequent response. Finally, in contrast to the 100 ms TEP component, the 50 ms TEP component at the target site was largely unaffected by lesions to recurrent connections at 20 ms and 50 ms, indicating that this earlier

part of the canonical TMS-EEG response can be attributed solely to the local impulse-response characteristics of a patch of cortical tissue.

Our results, and the framework for investigating such questions that we are introducing here, have clear and practical relevance to basic and clinical TMS-EEG research, but also have broader implications for the scientific understanding of functional brain organization. Variations on the concept of recurrence in systems neuroscience go back many decades, and have been developed in a wide number of areas and with a wide number of labels, including 're-entry,' 'reverberation,' 'feedback,' 'top-down control,' 'predictive coding,' 'functional/effective connectivity,' etc (*Edelman and Gally, 2013*; *Freeman, 1975*; *Llinás et al., 1998*; *Lopes et al., 2021*; *Massimini et al., 2005*; *Spiegler et al., 2016*). These framings vary a great deal on dimensions such as the spatial/temporal scale, role of corticothalamic interactions, association with cognitive functions, association with global brain state, level of physiological detail/abstraction, etc. In all these cases however the central shared intuition is that information or activity flows between network elements in the brain are bidirectional, but that the primary direction of travel may fluctuate dynamically over time. For example, the response of the visual system to images - a sensory stimulation-evoked response that is similar in many ways to electromagnetic stimulation-evoked responses - is widely understood to involve a period of feedforward activity propagation hierarchically up the ventral visual stream, followed rapidly by recurrent top-down feedback (*Ahlfors et al., 2015*; *Clarke et al., 2011*; *Lamme and Roelfsema, 2000*). Moreover, in vitro recordings have shown how a magnetic pulse delivered to a single ganglion cell generates a local early response that terminates after a few ms (*Bonmassar et al., 2012*) depicting a scenario similar to *Figure 4B–C*. The connectome-based neurophysiological modelling approach presented here could easily be deployed to investigate similar questions in these and other areas such as visual cognitive neuroscience or consciousness research, where feedback and recurrence play a central explanatory role in current theories.

The mathematical and theoretical neuroscientific context that has particularly informed the present study owes much to the ideas of Walter Freeman (*Freeman, 1975*) and of Andreas Spiegler and colleagues (*Spiegler et al., 2016*). Freeman's hierarchical `K Set' framework (*Freeman, 1975*) offers a rigorous technical and qualitative analysis of neuronal dynamics in systems progressing in complexity from a single excitatory neural population (K0e set), to ones with self-, uni-directional, and bi-directional excitation and inhibition (KI sets), and eventually adding network-level interactions and feedback (KII and KIII sets). Notably, Freeman's analysis provides both physiological and mathematical motivation for the central premise of our argument - that a local patch of cortical neural tissue can generate TEP-like damped oscillatory responses to a brief stimulation, without the need for feedback from other brain regions (this is also an implicit premise in all studies using second-order differential equations to model sensory-evoked potentials, such as *Freeman, 1975*, *Jansen and Rit, 1995*, *David et al., 2005*, and ourselves here). In these terms then, the questions we have posed and addressed are whether the 50 ms and 100 ms TEP components at the stimulation site represent KI set or KII set ensemble behavior. Complementing this, the nature of recurrent activity at the level of whole-brain connectome networks in particular is expressed more sharply in the work of *Spiegler et al., 2016*, who emphasizes how feedback loops within the connectome can lead to re-entrant activity, the result of which is to produce longer-lasting and temporally more complex evoked responses - consistent with our findings here. These authors also discovered from an exhaustive investigation comprising 37,000 simulation runs over 190 different stimulation targets that persistent, long-lasting activations tend to propagate within canonical resting-state networks. Interestingly, this prediction was later confirmed in our own experimental TMS-EEG work (*Eldaief et al., 2011*; *Ozdemir et al., 2020*), which demonstrated that the TEPs mainly propagate within distal cortical regions belonging to the same network. For example, stimulation of parietal default-mode network (DMN) nodes resulted in widespread sustained activity across the parietal, temporal, and frontal lobes - but this activity was primarily to be found within other DMN regions. The same result was also observed for nearby stimulation of dorsal attention network (DAN) nodes. More recently we obtained a similar result with anatomical connectivity (*Momi et al., 2021c*), namely that network-level anatomical connectivity is more relevant than local and global brain properties in shaping TMS signal propagation after the stimulation of two resting-state networks (again DMN and DAN). Whilst we did not study DMN or DAN stimulation in the present study, it can be seen from the Yeo network loadings in *Figures 3–5* that our results are also consistent with these experimental observations, with the somatomotor network

dominating for all our simulated M1 stimulations. Extending the present results to TEP measurements from additional target sites both anterior and posterior to the M1 target studied here is an important priority for future work with this model.

In addition to our scientific conclusions on the nature of recurrent activity in stimulation-evoked brain dynamics, the present work offers several technical advances over previous contributions in a number of areas. Our model is to our knowledge the first connectome-based neurophysiological model for TMS-EEG that demonstrates accurate single-subject reconstruction of TEP waveforms at the sensor and source level. Related work has focused on stimulation-evoked functional connectivity patterns (*Spiegler et al., 2016*) and stimulation-evoked time-frequency responses (*Cona et al., 2011*) within either large or small-scale networks. Most notably and recently, Bensaid and colleagues (*Bensaid et al., 2019*) proposed a whole-brain model of TMS-EEG TEP waveforms, with a focus on the sleep/wake differences in TMS-EEG responses studied by *Casali et al., 2013*, *Massimini et al., 2005*, and others. Bensaid et al's., model (*Bensaid et al., 2019*) includes extensive 'horizontal' corticothalamic connectivity, which we elected not to replicate in the present model for reasons of tractability, but may add in future iterations. None of the above studies, or indeed any published work to date to our knowledge, achieve the level of accuracy for single-subject TEP waveform fits that we show here. Our model's success on this front is owed in large part to our decision to formulate and implement the Jansen-Rit connectome network differential equations in the widely-used machine learning library PyTorch (*Paszke et al., 2019*). We have recently discussed and demonstrated the advantages of deep learning-based computational architectures for neurophysiological model simulation and parameter estimation (*Griffiths et al., 2022*). In the present study this precision was critical for addressing our research questions, which centred on the timing and amplitudes of well-defined TEP waveform components. These components can be found in most or all subjects, but vary considerably in their shapes and exact timings.

One example of the utility of this new model-fitting framework can be seen in our results in *Figure 5*, where we identified trends over subjects in the relationship of estimated model parameters to individual variation in TEP waveform features. Through these analyses we found, in an entirely data-driven fashion, that the synaptic time constant of the inhibitory Jansen-Rit population is a strong predictor of the amplitude of early (P60) and late (N100) TEP components. This is consistent with the finding of increased TEP amplitudes following the application of paired-pulse TMS protocols known to affect inhibition (or reduced excitability) (*Rogasch et al., 2013*). Similarly, pharmacological intervention studies have shown that GABA$_B$ receptor agonists (benzodiazepine) decrease N100 component amplitude, suggesting that this component is driven by GABA$_B$ receptor-mediated inhibition. Whilst further research will be needed to explore and verify this hypothesis, its generation via the combination of data-driven model fitting and theoretically-informed brain network simulations offers a promising new approach for the interpretation of TMS-EEG experiments, and neurophysiological research more broadly.

## Materials and methods
### Overview of approach

The analyses conducted in the present study consist of four main components: (i) TMS-EEG evoked response source reconstructions, (ii) construction of anatomical connectivity priors for our computational model using diffusion-weighted MRI (DW-MRI) tractography, (iii) simulation of whole-brain dynamics and stimulation-evoked responses with a connectome-based neural mass model, and (iv) fitting of the model to individual-subject TMS-EEG data. A schematic overview of the overall approach is given in *Figure 6*.

### TMS-EEG data and source reconstruction

The TMS-EEG data used in this study were taken from an open dataset collected and provided to the community by the Rogasch group (figshare.com/articles/dataset/TEPs-_SEPs/7440713), where high-density EEG was recorded following a stimulation of primary motor cortex (M1) in 20 healthy young individuals (24.50±4.86 years; 14 females), and in which state-of-the-art preprocessing had already been applied. For details regarding the data acquisition and the preprocessing steps please refer to the original paper of *Biabani et al., 2019*. All TMS-evoked EEG source reconstruction was performed

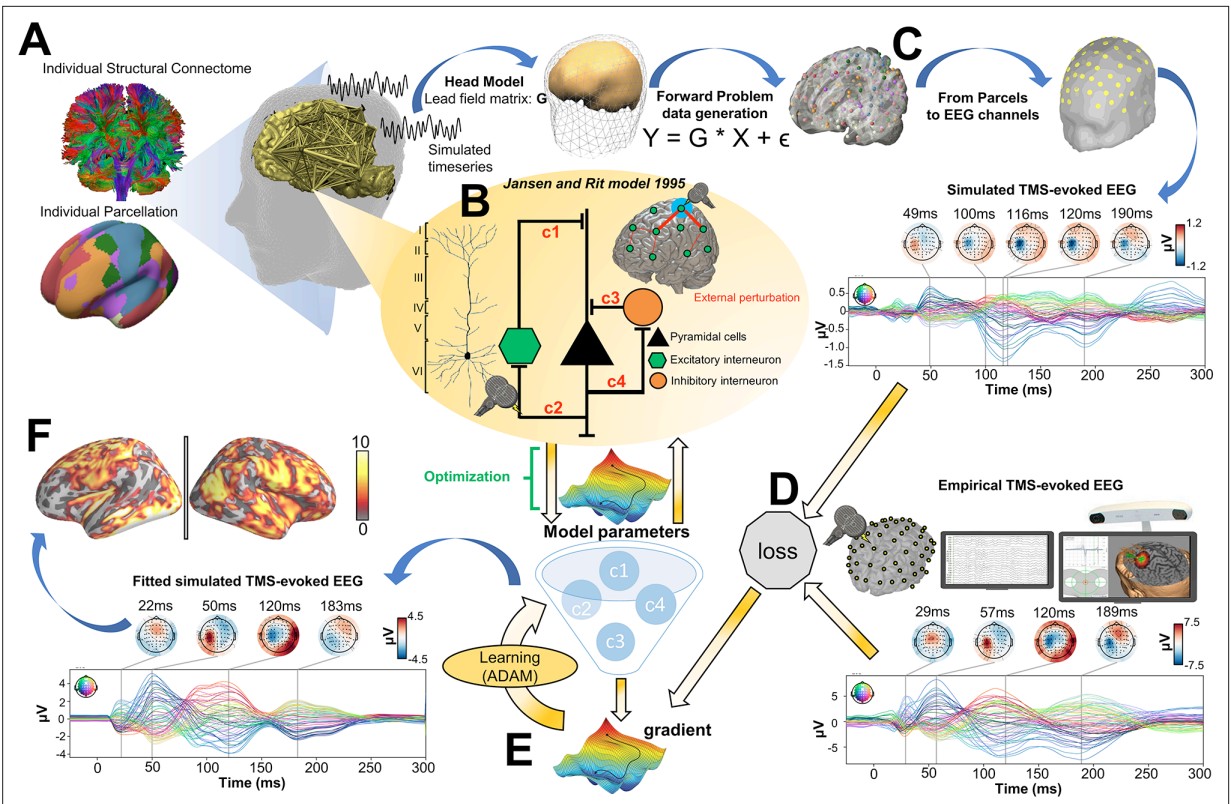

**Figure 6.** Methodological workflow for subject-specific connectome-based neurophysiological modelling of TMS-EEG TEPs. (**A**) Diffusion-weighted MRI (DW-MRI) tractography was computed from a sample of healthy young individuals from the Human Connectome Project (HCP) Dataset (*Van Essen et al., 2012*), and then averaged to give a grand-mean anatomical connectome. The 200-parcel Schaefer atlas (*Schaefer et al., 2018*) was used, which usefully aggregates its 200 brain regions into seven canonical functional networks (Visual network: VISN, Somatomotor network: SMN, Dorsal attention network: DAN, Anterior salience network: ASN, Limbic network: LIMN, Fronto-parietal network: FPN, Default mode network: DMN). These parcels were mapped to the individual's FreeSurfer parcellation using spherical registration (*Fischl et al., 1999*). Once this brain parcellation covering cortical structures was extrapolated, it was then used to extract individual anatomical connectomes. (**B**) The Jansen-Rit model (*Jansen and Rit, 1995*), a neural mass model comprising pyramidal, excitatory interneuron, and inhibitory interneuron populations was embedded in every parcel for simulating and fitting neural activity time series. The TMS-induced depolarization of the resting membrane potential was modeled by a perturbing voltage offset to the mean membrane potential of the excitatory interneuron population. (**C**) A lead field matrix was then used for moving the parcels' time series into channel space and generating simulated electroencephalography (EEG) measurements. (**D**) The goodness-of-fit (loss) was calculated as the cosine similarity between simulated and empirical TMS-EEG time series. (**E**) Utilizing the autodiff-computed gradient (*Rall, 1981*) between the objective function and model parameters, model parameters were optimized using the ADAM algorithm (*Kingma and Ba, 2017*). (**F**) Finally, the optimized model parameters were used to generate the fitted, simulated TMS-EEG activity, for which we report comparisons with the empirical data at both the channel and source level using conventional statistical techniques.

using the MNE software library (*Gramfort et al., 2014*) (mne.tools/stable/index.html) running in Python 3.6. First, the watershed algorithm was used to generate the inner skull, the outer skull, and the outer skin surface triangulations for the 'fsaverage' template. Then the EEG forward solution was calculated using a three-compartment boundary-element model (*Gramfort et al., 2010*). Noise covariance was estimated from individual trials using the pre-TMS (from −1000 ms to −100 ms) time window as a baseline. The inverse model solution of the cortical sources was performed using the dSPM method with current density (*Dale et al., 2000*) and constraining source dipoles to the cortical surface. The resulting output of EEG source reconstruction was the dSPM current density time series for each cortical surface location.

## Neuroimaging data and definition of connectome weight priors

The whole-brain model we fit to each of the 20 subjects' TMS-EEG consists of 200 brain regions, connected by weights of the anatomical connectome. We set strong priors on the connection weights, such that individual fits allow for small adjustments of these values.

Specifically, a prior variance of 1/50 was set for every connection, which was determined empirically to provide a strong constraint but allows for some flexibility during the fitting process. Importantly, after fitting every subject, a visual inspection was conducted to confirm that the (posterior mean) connection weight matrices retained the key topological features present in empirical neuroimaging measurements. For a comprehensive overview of the posterior mean structural connectivity weights please refer to *Appendix 2—figure 4*.

To obtain population-representative values for these connectivity priors, we ran diffusion-weighted MRI tractography reconstructions across a large number of healthy young subjects and averaged the results.

For these analyses we used structural neuroimaging data of 400 healthy young individuals (170 males; age range 21–35 years), taken from the Human Connectome Project (HCP) Dataset ( humanconnectome.org/study/hcp-young-adult) (*Van Essen et al., 2012*). DW-MRI preprocessing was run in Ubuntu 18.04 LTS, using tools from the FMRIB Software Library (FSL 5.0.3; https://www. fmrib.ox.ac.uk/fsl) (*Jenkinson et al., 2012*), MRtrix3 (https://www.MRtrix.readthedocs.io) (*Tournier et al., 2012*) and FreeSurfer 6.0 (*Fischl, 2012*). All images used were already corrected for motion via FSL's EDDY (*Andersson and Sotiropoulos, 2016*) as part of the HCP minimally-preprocessed diffusion pipeline (*Glasser et al., 2013*). The multi-shell multi-tissue response function (*Christiaens et al., 2015*) was estimated using constrained spherical deconvolution (*Jeurissen et al., 2014*). T1-weighted (T1w) images, which were already coregistered to the b0 volume, were segmented using the FAST algorithm (*Zhang et al., 2001*). Anatomically constrained tractography was employed to generate the initial tractogram with 10 million streamlines using second-order integration over fiber orientation distributions (*Tournier et al., 2010*). Then, the spherical-deconvolution informed filtering of tractograms (SIFT2) methodology was applied (*Smith et al., 2015*), in order to provide more biologically accurate measures of fiber connectivity. Brain regions or network nodes were defined using the 200-region atlas of *Schaefer et al., 2018*, which was mapped to each individual's FreeSurfer surfaces using spherical registration (*Fischl et al., 1999*). This atlas additionally provides categorical assignments of regions into 7 canonical functional brain networks (Visual network: VISN, Somatomotor network: SMN, Dorsal attention network: DAN, Anterior salience network: ASN, Limbic network: LIMN, Fronto-parietal network: FPN, Default mode network: DMN). Using this atlas in combination with the filtered streamlines, 200 × 200 two anatomical connectivity matrices were extracted, with matrix elements representing the number of streamlines and the fiber length connecting each pair of regions, respectively. These connectomes for the 400 HCP subjects were then averaged, yielding a healthy subject population-representative connectome matrix. Finally, this matrix was prepared numerically for physiological network modelling by rescaling values by first taking the matrix Laplacian, and second by scalar division of all entries by the matrix norm, which ensures that the matrix is linearly stable (i.e. all eigenvalues have negative real part except one eigenvalue which was zero). The Laplacian sets each row sum (i.e. each node's weighted in-degree) to zero by subtracting row sums from diagonal, and has been used often in previous whole-brain modelling research (*Abdelnour et al., 2018*; *Atasoy et al., 2016*; *Raj et al., 2022*).

## Large-scale connectome-based neurophysiological brain network model

As previously described, a brain network model comprising 200 cortical areas was used to model TMS-evoked activity patterns, where each network node represents the population-averaged activity of a single brain region according to the rationale of mean-field theory (*Deco et al., 2008*). We used the Jansen-Rit (JR) equations to describe activity at each node, which is one of the most widely used neurophysiological models for both stimulus-evoked and resting-state EEG activity measurements (*David et al., 2005*; *Jansen and Rit, 1995*; *Spiegler et al., 2010*). JR is a relatively coarse-grained neural mass model of the cortical microcircuit, composed of three interconnected neural populations: pyramidal projection neurons, excitatory interneurons, and inhibitory interneurons. The excitatory and the inhibitory populations both receive input from and feedback to the pyramidal population but not to each other, and so the overall circuit motif (*Figure 6B*) contains one positive and one negative feedback loop. For each of the three neural populations, the post-synaptic somatic and dendritic membrane response to an incoming pulse of action potentials is described by the second-order differential equation

$$\ddot{v}(t) + \frac{2}{\tau_{e,i}}\dot{v}(t) + \frac{1}{\tau_{e,i}^2}v(t) = \frac{H_{e,i}}{\tau_{e,i}}m(t) \tag{1}$$

which is equivalent to a convolution of incoming activity with a synaptic impulse response function

$$v(t) = \int_0^\infty d\tau\, m(\tau) \cdot h_{e,i}(t - \tau) \tag{2}$$

whose kernel $h_{e,i}(t)$ is given by

$$h_{e,i} = \frac{H_{e,i}}{\tau_{e,i}} \cdot t \cdot exp(-\frac{t}{\tau_{e,i}}) \tag{3}$$

where $m(t)$ is the (population-average) presynaptic input, $v(t)$ is the postsynaptic membrane potential, $H_{e,i}$ is the maximum postsynaptic potential, and $\tau_{e,i}$ a lumped representation of delays occurring during the synaptic transmission.

This synaptic response function, also known as a pulse-to-wave operator (**Freeman, 1975**) determines the excitability of the population, as parameterized by the time constants $\tau_e$ and $\tau_i$, which are of particular interest in the present study. Complementing the pulse-to-wave operator for the synaptic response, each neural population also has a wave-to-pulse operator (**Freeman, 1975**) that determines the output - the (population-average) firing rate - which is an instantaneous function of the somatic membrane potential that takes the sigmoidal form

$$S(v) = \frac{e_0}{1 - exp(r(v_0 - v))} \tag{4}$$

where $e_0$ is the maximum firing rate, $r$ is the steepness of the sigmoid function, and $v_0$ is the postsynaptic potential for which half of the maximum firing rate is achieved.

In practice, we re-write the three sets of second-order differential equations that follow the form in Equation 1 (one for each population in the JR circuit) as three interconnected pairs of coupled first-order differential equations, and so the full JR system for each individual cortical area $j \in \{i,..., N\}$ in our network of N=200 regions is given by the following six equations:

$$\dot{v}_{j1} = x_{j1} \tag{5}$$

$$\dot{x}_{j1} = \frac{H_e}{\tau_e}(p + conn_j) + C_1 S(C_2 v_{j3}) - \frac{2}{\tau_e}x_{j1} - \frac{1}{\tau_e^2}v_{j1} \tag{6}$$

$$\dot{v}_{j2} = x_{j2} \tag{7}$$

$$\dot{x}_{j2} = \frac{H_i}{\tau_i}(C_4 S(C_3 v_{3j})) - \frac{2}{\tau_i}x_{j2} - \frac{1}{\tau_i^2}v_{j2} \tag{8}$$

$$\dot{v}_{j3} = x_{j3} \tag{9}$$

$$\dot{x}_{j3} = \frac{H_e}{\tau_e}(S(v_{j1} - v_{j_2})) - \frac{2}{\tau_e}x_{j3} - \frac{1}{\tau_e^2}v_{j3} \tag{10}$$

where $v_{1,2,3}$ is the average postsynaptic membrane potential of the excitatory interneuron, inhibitory interneuron, and pyramidal cell populations, respectively. The input from other nodes in the whole-brain network

$$conn_j(t) = S(\sum_{k \neq j} a_{jk}x_{k1}(t - m_{jk})) \tag{11}$$

where $a_{jk}$ is the $j_{th}$ row and the $k_{th}$ column in the connectivity matrix **A** (which in our case is the rescaled connectivity Laplacian as described above). $conn_j$ thus enters into the excitatory population only and collects excitatory population activity from other network nodes. Due to the finite velocity of long-range axonal conduction, these inputs appear after delays of around 5–50 ms, which vary on a per-connection basis. This is specified by $m_{jk}$, the $j,k_{th}$ entry of the delays matrix **M=T/s**, which is a function of the inter-regional fiber tract length matrix **T** and the global axonal conduction velocity $s$. Especially important here, the TMS-induced depolarization of the resting membrane potential was modelled by an external perturbing voltage offset p applied to the excitatory interneuron population.

To establish which parcels in the model the TMS stimulation is injected into, and with what strength, the TMS-induced electric field was modelled with SimNIBS (*Saturnino et al., 2019*) in the MNI152 standard space. The normalized electric field or E-field distribution was thresholded at 83% of its maximal value, following recent estimates of the E-field thresholds above which tissue is activated by TMS (*Romero et al., 2019*). This thresholded E-field map was then used to inject a weighted stimulus into the target regions in the model. Finally, channel-level EEG signals were computed in the model by first taking the difference r(t)=v1(t) − v2(t) between the excitatory and inhibitory interneuron activity at each cortical parcel (*Schaefer et al., 2018*), and projected to the EEG channels space using a lead field matrix Y=G * X + ε. We based this decision on previous studies (*David et al., 2006*) which consider that the post-synaptic potential in the Jansen-Rit model is defined by the inputs to that population, namely the excitatory and inhibitory interneuron populations, which are differenced due to their opposite polarities. For details on the TMS biophysical modelling please see Appendix 1.

## Individual-subject Jansen-Rit connectome model parameter estimation from TMS-EEG data

We used a novel brain network model parameter optimization technique (*Griffiths et al., 2022*) for fitting individual-subject TEP waveforms and identifying subject-level physiological parameters from empirical data. Notably, the model is implemented in PyTorch (*Paszke et al., 2019*), a software library that has in recent years been widely adopted by the machine learning community in both academic and commercial sectors. Moving to this framework from more conventional numerical simulation libraries involves some minor modifications to accommodate tensor data structures, but brings the substantial advantage of naturally accommodating gradient-based parameter optimization via automatic differentiation-based algorithms, for relatively complex sets of equations that do not admit tractably computable Jacobians. This is one of a growing number of cases (e.g. *Richards et al., 2019*; *Suárez et al., 2020*) where the natural parallel between our physiologically-based large-scale brain network models and deep recurrent neural networks used in machine learning is proving technically and conceptually fruitful.

The general mathematical framework for this approach has been described by us in a recent technical paper (*Griffiths et al., 2022*), where it was applied in the context of connectome-based neurophysiological modelling of resting-state fMRI data. In the present work we are extending this technique's domain of application to fast-timescale evoked responses, but the overall approach in the two cases is the same with minor modifications. The algorithm proceeds by dividing a subject's multi (in this case 64) -channel, 400 ms long (–100 ms to +300 ms post-stimulus), trial-averaged TMS-EEG TEP waveform into short (20 ms) non-overlapping windows, termed batches. The total length of the simulation was 400 ms which included –100 ms of baseline (pre-TMS) and +300 ms after stimulation injection. Prior to the start of the 100 ms baseline, a 20 ms burn-in was included, which allowed for the system to settle after the usual transient due to randomly selected initial conditions for the state variables.

Rolling through each batch in the time series sequentially, the JR model-simulated TEP $\hat{y}$ was generated with the current set of parameter values, and its match to the empirical TEP y was calculated with the following mean-squared error (MSE) loss function

$$L = \frac{1}{N_t} \sum_{t=1}^{N_t} \left( \frac{1}{N_{ch}} \sum_{i=1}^{N_{ch}} (y_i(t) - \hat{y}_i(t))^2 \right) \tag{12}$$

where $N_t$ is the number of the time points and $N_{ch}$ is the number of EEG channels. It is assumed that the model parameters are Gaussian. Together with a complexity-penalizing regularization term on each model parameter $\theta$,

$$C = \ln \sigma + \frac{1}{\sigma^2} (\theta - \mu)^2, \tag{13}$$

where the mean μ and standard variation σ of the model parameter θ are hyper-parameters to be fitted. The model parameters' complexity defined in Equation (13) is included as a regularization term to avoid over-fitting and help achieve a robust model. The loss function L and the complexity term C are combined into a final objective function that is provided to PyTorch's native ADAM algorithm

(*Kingma and Ba, 2017*), which selects the candidate parameter set for the next batch with a stochastic gradient descent-based scheme that utilizes automatic-differentiation-based gradients (efficient computation of which is a primary design objective of the (Py)Torch C++backend). When the batch window reaches the end of the TEP time series, it returns to the start and repeats until convergence. When the optimization was completed, the average value for every parameter was computed using the last 100 batches and then used for running the simulations. For an overview of all parameters used in the model, please refer to *Appendix 2—figure 3* and *Supplementary file 1*. For a complete description of the parameter estimation algorithm, please see *Griffiths et al., 2022*.

## Assessing the similarity between simulated and empirical TEPs

To further assess the goodness-of-fit of the simulated TEP waveforms arrived at after convergence of the ADAM algorithm, we conducted additional analyses in both EEG sensor and source space. At the channel level, Pearson correlation coefficients and corresponding p-values between empirical and model-generated TEP waveforms were computed for each subject. In order to control for type I error, this result was compared with a null distribution constructed from 1000 time-wise random permutations, with a significance threshold set at $p < 0.05$. As a complement to these TEP comparisons that emphasize matching of waveform shape and component timing, we also examined more holistic time series variability characteristics using the PCI (*Casali et al., 2013*), which was extracted from the simulated and the empirical TMS-EEG data, and Pearson correlations between the two computed. PCI gauges the amount of information contained in the integrated response to a direct perturbation which uses the Lempel-Ziv measure of algorithmic complexity to approximate the amount of non-redundant information contained in a binary sequence by estimating the minimal number of different patterns necessary to describe the sequence (*Lempel and Ziv, 1976*). For a detailed description of the PCI calculation and extraction please refer to the original paper (*Casali et al., 2013*).

Assessment of the goodness-of-fit at the source level proceeded in a similar fashion: Individual subjects' empirical and model-generated TMS-EEG time series were first computed for every source-space surface vertex, as described above. Pearson correlation coefficients and corresponding p-values, indicating empirical-simulated data similarities, were computed. Again, in order to control for type I error, time-wise permutation testing was done by comparison against 1000 surrogate, shuffled TEP differences, with a significance threshold set at $p < 0.05$. Finally, and unlike the channel-level data, network-level comparisons of simulated vs. empirical activity patterns were made by averaging current densities over surface vertices at each point in time within each of the seven Freesurfer surface-projected canonical Yeo network maps (*Thomas Yeo et al., 2011*), and Pearson correlation coefficients and p-values between empirical and simulated network-level time series were again computed.

## Dissecting the propagation dynamics of TMS-evoked responses

A key aim of the present study is to ascertain whether the TMS-evoked activity in a certain region at a certain time point is primarily attributable to a localized response to TMS at the primary stimulation site, or to re-entrant activity feeding back from other nodes in the connectome network. In order to explore this, activity of each network node at a given time point was extracted as the sum of the absolute value of the simulated pyramidal cell population activity within a narrow temporal window (0–300 ms). Maximally activated nodes were defined as the top 1% of nodes exceeding two standard deviations above the mean over regions. This approach was used to identify, for each subject individually, the most important nodes at three key time points: 20 ms, 50 ms, and 100 ms after the TMS pulse, where we wanted to identify the contribution of re-entrant activity. With these key brain regions identified for each time window of interest, simulations were re-run for each subject using their optimal parameters estimated from the original TEP fitting step - but this time with the selected nodes' incoming and the outgoing connection weights set to zero for the duration of the window. These new 'virtually lesioned' TEP time series was again projected to the EEG channel space and back to the source level, where they were compared against the original model-generated TEP time series. Finally, as above, the model-generated dSPM values were extracted from the seven canonical network surface maps for each individual and for each condition, and analyzed statistically using the Statistical Package for the Social Sciences (SPSS) Version 25 (IBM Corp). A 4x7 repeated measures ANOVA with within-subjects factors 'TIME OF DAMAGE" (four levels: 20 ms; 50 ms; 100 ms; no damage) and 'NETWORK' (seven levels: VISN; SMN; DAN; ASN; LIMN; FPN; DMN) was run. Post-hoc paired t-tests

were used to detect dSPM value changes for different networks and lesion times, testing on a per-network basis whether and at what times the virtual lesions impacted on network-level activations.

In order to control for possible confounding factors such as the residual sensory artifacts (auditory or somatosensory) that can completely bias the interpretation of the re-entrant activity, we have run the same analysis in another independent open dataset (https://figshare.com/articles/dataset/Data_for_Fecchio_Pigorini_et_al_/4970900/1) (*Fecchio et al., 2017*). For a complete overview of this control analysis and the corresponding results, please refer to Appendix 2 and *Appendix 2—figures 6–9*. For a comprehensive representation of the maximally activated nodes please refer to *Appendix 2—figure 2*.

### Identifying clusters of different TMS-evoked responses

We aimed at predicting the spatiotemporal propagation of the TMS-evoked signal using the optimized physiological parameters of the model. First, SVDs were run on the grand mean of both the empirical and the model-generated TMS-EEG data, in order to identify prototypical TMS-evoked responses. Following this, the group-level SVD spatial eigenmodes were identified within each subject's time series corresponding to the time point with the highest cosine similarity between the individual's TEP and the prototypical response. Latencies and amplitudes of the SVD left singular vector time series peaks were extracted for every subject and related to the individuals' JR model parameters, with Pearson correlation coefficients and corresponding p-values computed accordingly. The critical p-value was then adjusted using Bonferroni correction to account for multiple comparisons (corrected p-value = 0.007).

## Acknowledgements

This research was generously supported by grants from the Labatt Family Innovation Fund in Brain Health, the CAMH Discovery Fund, and the Tri-Council (SSHRC-NSERC-CIHR) UK-Canada AI-Initiative. The funding sources had no involvement in the study design; nor in the collection, analysis, and interpretation of data; nor in the writing of the manuscript; nor in the decision to submit the manuscript for publication.

## Additional information

### Funding

| Funder | Grant reference number | Author |
| --- | --- | --- |
| Labatt Family Innovation Fund in Brain Health | | John D Griffiths |
| Centre for Addiction and Mental Health | Discovery Fund | John D Griffiths |
| Tri-Council (SSHRC-NSERC-CIHR) UK-Canada AI Initiative | | John D Griffiths |

The funders had no role in study design, data collection and interpretation, or the decision to submit the work for publication.

### Author contributions

Davide Momi, Conceptualization, Data curation, Formal analysis, Validation, Investigation, Visualization, Methodology, Writing – original draft, Writing – review and editing; Zheng Wang, Formal analysis, Supervision, Investigation, Methodology; John D Griffiths, Conceptualization, Supervision, Funding acquisition, Project administration, Writing – review and editing

### Author ORCIDs

Davide Momi (ID) http://orcid.org/0000-0001-6048-8296
Zheng Wang (ID) http://orcid.org/0009-0003-6539-2543

John D Griffiths http://orcid.org/0000-0002-1764-2179

### Ethics

The computational modelling work undertaken in this study made use of the publicly available TMS-EEG datasets of Rogasch group (figshare.com/articles/dataset/TEPs-_SEPs/7440713; Biabani et al., 2019) and Massimini & Rosanova group (https://figshare.com/articles/dataset/Data_for_Fecchio_Pigorini_et_al_/4970900/1; Fecchio et al. 2017), links to which are provided above. No new human subjects data was collected. The study was reviewed and approved by the Toronto Centre For Addiction and Mental Health Research Ethics Board (REB numbers #022/2020 & #035/2021; PI Dr. John D Griffiths), which operates in compliance with the Canadian Tri-Council Policy Statement: Ethical Conduct for Research Involving Humans (TCPS 2), the International Conference on Harmonisation Good Clinical Practice Consolidated Guideline (ICH GCP), Part C Division 5 of the Food and Drug Regulations, Part 4 of the Natural Health Products Regulations, Part 3 of the Medical Devices Regulations, and the provisions of the Ontario Personal Health Information Protection Act (PHIPA 2004) and its applicable regulations.

### Decision letter and Author response

Decision letter https://doi.org/10.7554/eLife.83232.sa1
Author response https://doi.org/10.7554/eLife.83232.sa2

---

## Additional files

### Supplementary files

• Supplementary file 1. Used fitted parameters for each Jansen-Rit element in the large-scale brain network.

• MDAR checklist

### Data availability

Full code for reproduction of the data analysis and model fitting described in this paper is freely available online at https://github.com/GriffithsLab/PyTepFit (copy archived at swh:1:rev:4222cd27f-c3a451a9b43eb788d5f5e50312aed41). As noted above, TMS-EEG data were taken from an open dataset (figshare.com/articles/dataset/TEPs-_SEPs/7440713). Structural MRI data used in the study are available from the original Human Connectome Project dataset (humanconnectome.orghuman-connectome.org).

The following previously published datasets were used:

| Author(s) | Year | Dataset title | Dataset URL | Database and Identifier |
|---|---|---|---|---|
| Biabani M, Fornito A, Mutanen TP, Morrow J, Rogasch NC | 2019 | TEPs-PEPs | https://figshare.com/articles/dataset/TEPs-_SEPs/7440713 | figshare, TEPs-_SEPs/7440713 |
| Fecchio M, Pigorini A, Comanducci A | 2017 | TEPs | https://figshare.com/articles/dataset/Data_for_Fecchio_Pigorini_et_al_/4970900/1 | figshare, Data_for_Fecchio_Pigorini_et_al_/4970900/1 |

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

# Appendix 1

## Modelling of the TMS-induced electric field

In order to identify the brain regions engaged by M1-targeted TMS, and therefore which nodes in our simulation to inject an external input, and with what magnitude, the TMS-induced electric field was modelled with SimNIBS (*Thielscher et al., 2015*) (simnibs.github.io/simnibs). A tetrahedral head model (mesh file) was created, consisting of five tissue types: white matter (WM), gray matter (GM), cerebro-spinal fluid (CSF), skull, and scalp. The assigned conductivity values were fixed: 0.126 S/m (WM), 0.275 S/m (GM), 1.654 S/m (CSF), 0.01 S/m (skull), 0.465 S/m (scalp). The distance between the coil and the cortex was set to 10 mm, as measured in our MRI images, and the coil handle was oriented following the M1 coordinates following the methods and materials of the original paper of *Biabani et al., 2019*. The rate of change of the coil current *(dI/dt)* was calculated assuming a quasi-static regime (*Opitz et al., 2015*), according to the following equation:

$$E = \frac{\vartheta A}{\vartheta t} - \Delta \varphi$$

where *E* is the electric field vector, φ denotes the electric potential, and *A* is the magnetic vector potential of the TMS coil, which depends on the coil's shape, position, and the current flow in the coil wires.

## Appendix 2

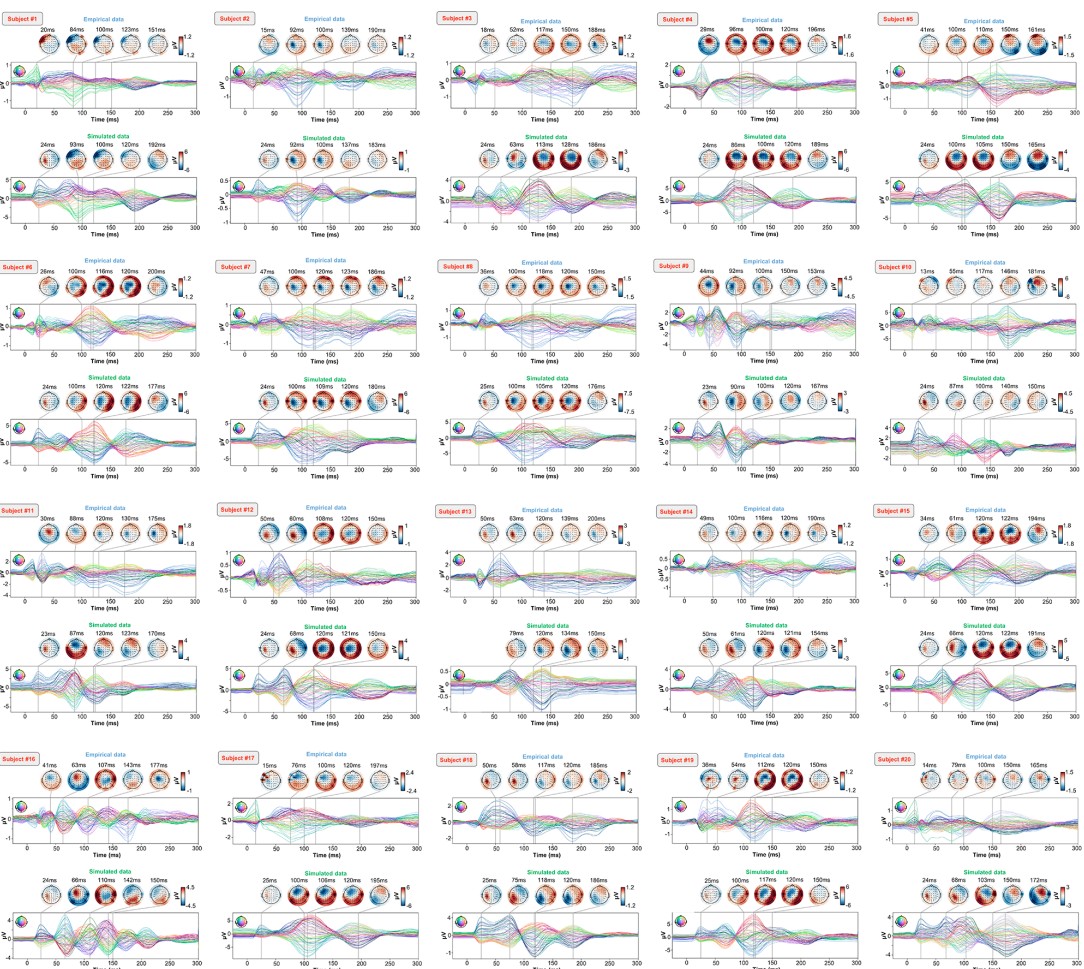

**Appendix 2—figure 1.** Optimized TMS-EEG evoked potential (TEP) models for all subjects. For every pair of rows, empirical (upper) and simulated (lower) TMS-EEG responses are shown for every study subject, extending main text *Figure 2* where a selected subset of subjects' data are shown. These data reiterate and reinforce the demonstrations in *Figure 2* that the model-generated electroencephalography (EEG) activity time series achieve robust recovery of individual subjects' empirical TEP propagation patterns.

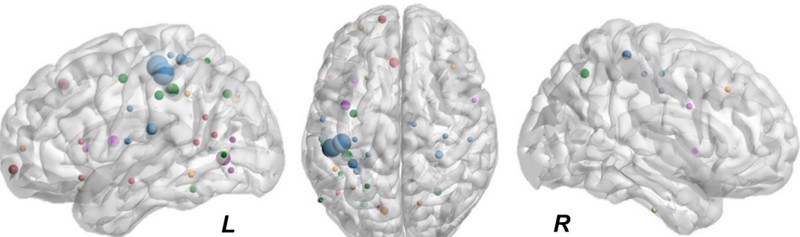

**Appendix 2—figure 2.** Representation of the maximally activated nodes. Sizes indicate for how many subjects that node was identified as maximally activated by the external perturbation. A clear pattern is evident, where the more important nodes are the ones more affected by the stimulation (e.g. motor, parietal, and frontal cortex). Once those nodes were identified, their incoming and the outgoing connection weights were set to zero for the duration of the simulation.

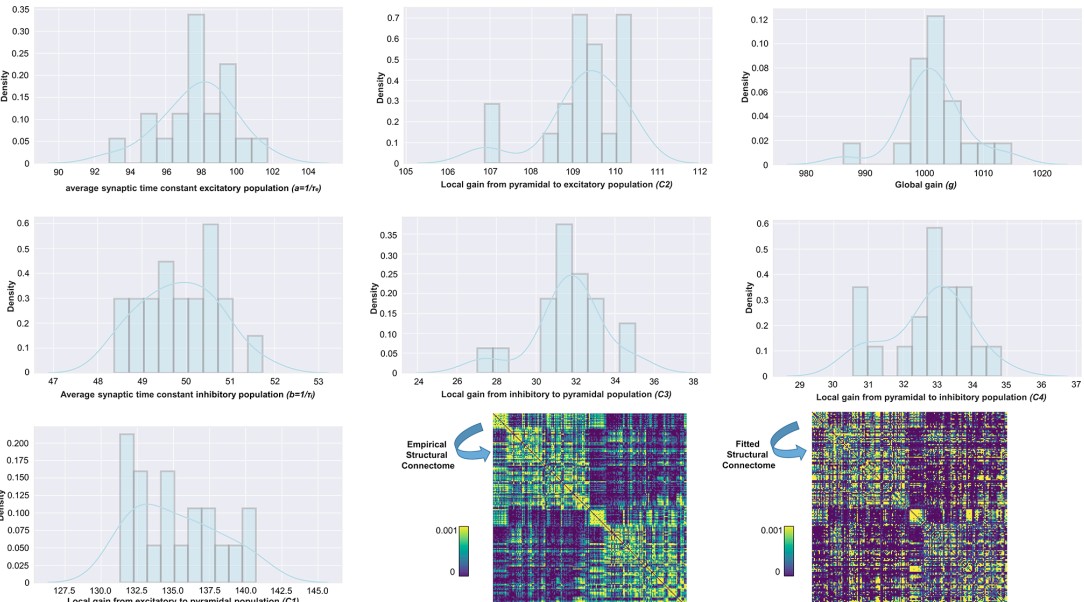

**Appendix 2—figure 3.** Distributions of physiological parameter estimates over subjects. Histograms and kernel density estimates of the estimated values for the Jansen-Rit model physiological parameters over all subjects. Also shown are prior and posterior parameter values for anatomical connectome weights for a single example subject (bottom right). Parameter estimation was performed using our novel automatic differentiation and gradient-based approach inspired by current techniques in deep learning (***Griffiths et al., 2022***).

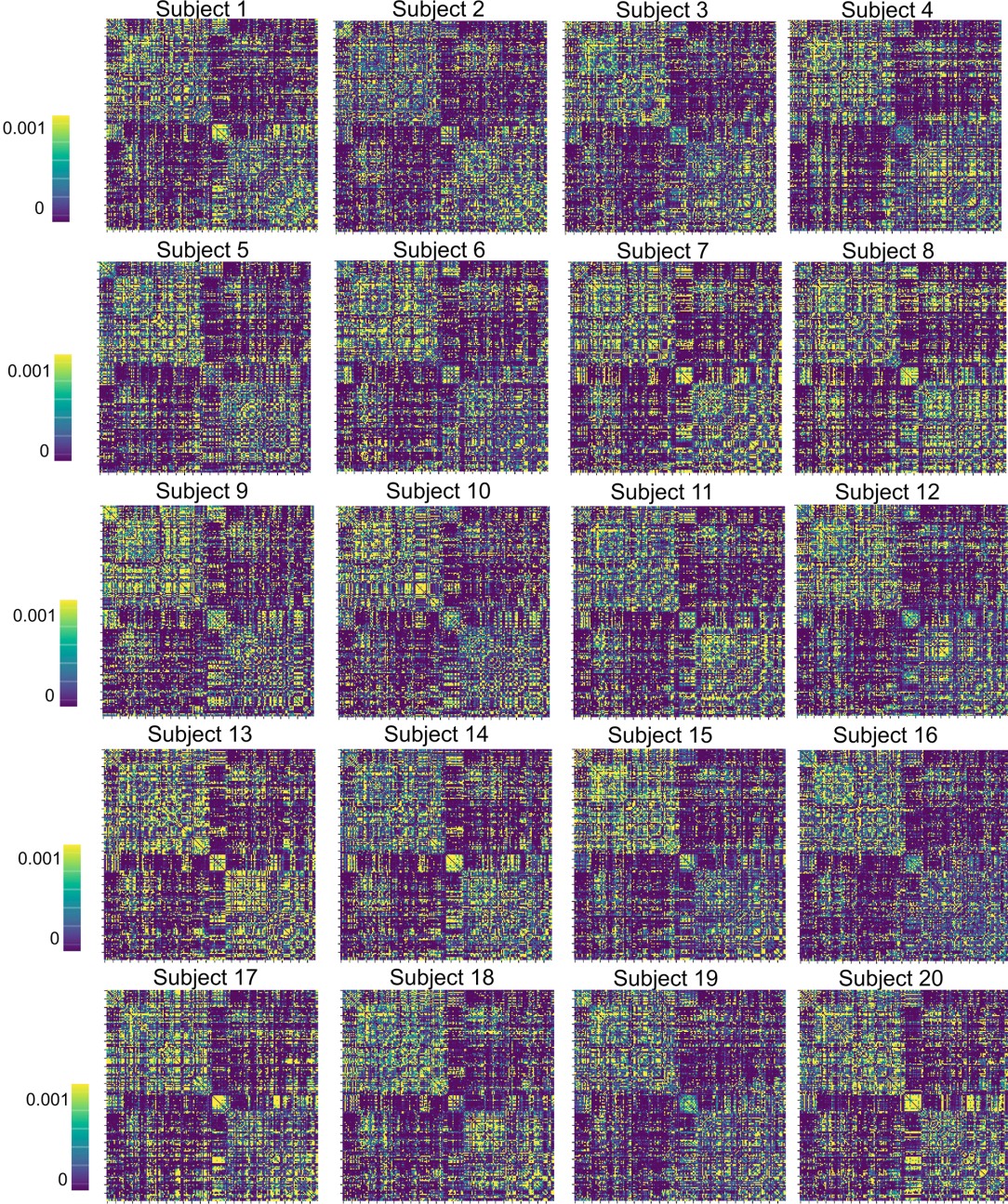

**Appendix 2—figure 4.** Matrices of the posterior mean structural connectivity weights for all subjects.

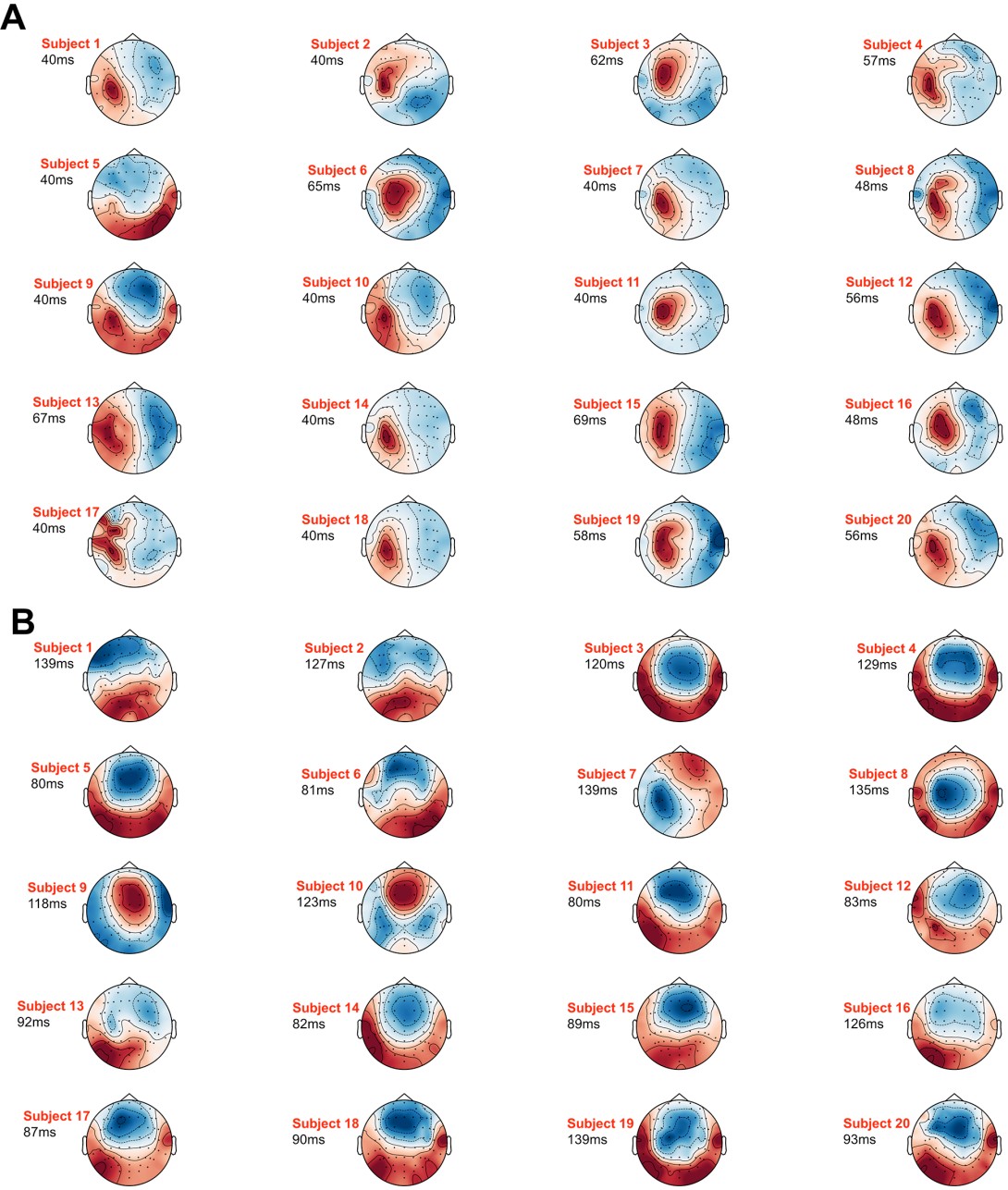

**Appendix 2—figure 5.** Timing and topographies of the prototypical TMS-EEG evoked potential (TEP) response pattern in each subject. These figures extend the single-subject examples from TEP channel data singular value decompositions (SVD) decompositions in *Figure 5*. (**A**) First right singular vectors from TEP SVDs for all subjects, with corresponding time location indicating the time point of maximum expression for the corresponding left singular vector (temporal eigenmode). (**B**) Second right singular vectors and corresponding time points of maximum expression.

## Statistical analyses of TMS-evoked activity propagation and maximum activation-based virtual lesions

Continuing the results reported in main text Section 2.2 from ANOVA on network ROI-averaged source activity (dSPM) values, we conducted the following post-hoc t-tests to compare simulated network activation by TMS for different networks and virtual lesion timings: Taking into account the factor TIME OF DAMAGE, post-hoc comparisons showed significant differences in dSPM values for: 20 ms>50 ms: mean difference = −0.52, p<0.0001; 20ms>100 ms: mean difference = −2.16, p<0.0001; 20 ms >no damage: mean difference = 1.53, p<0.0001; 50 ms>100 ms: mean difference =

−1.63, p<0.0001; 50 ms>no damage: mean difference = 1.01, p<0.0001; 100 ms>no damage: mean difference = 0.62, p<0.0001. Conversely, considering the factor NETWORK, post-hoc comparisons showed significant differences in dSPM values for: VIS >SMN: mean difference = −1.91, p<0.0001; VIS >DAN: mean difference = −0.90, p<0.0001; VIS >ASN: mean difference = 1.93, p<0.0001; VIS >LIM: mean difference = −0.21, p=0.002; VIS >FPN: mean difference = −0.79, p<0.0001; VIS >DMN: mean difference = −0.72, p<0.0001; SMN >DAN: mean difference = 1.01, p<0.0001; SMN >ASN: mean difference = 1.72, p<0.0001; SMN >LIM: mean difference = 2.12, p<0.0001; SMN >FPN: mean difference = 1.11, p<0.0001; SMN >DMN: mean difference = 1.19, p<0.0001; DAN >ASN: mean difference = 0.70, p<0.0001; DAN >LIM: mean difference = 1.11, P<0.0001; DAN >DMN: mean difference = 0.18, p=0.006; ASN >LIM: mean difference = −0.40, p<0.0001; AS >FPN: mean difference = 0.60, p<0.0001; ASN >DMN: mean difference = −0.52, p<0.0001; LIM >FPN: mean difference = −1.01, p<0.0001; LIM >DMN: mean difference = −0.93, p<0.0001; FPN >DMN: mean difference = −0.07, p=0.002. Conversely no significant differences were found for DAN >FPN: mean difference = 0.11, p=0.1. These post-hoc comparisons indicate that first, early damages of the structural connectome (e.g. 20 ms and 50 ms) compromise the propagation of the TMS-evoked signal compare to the other conditions (e.g. 100 ms and no damage). Second, the network where this result is more pronounced is the stimulated SMN.

## Control analyses

*Appendix 2—figures 6 and 7* shows empirical and fitted TEP waveforms and selected topography maps for the entire group of the Fecchio, Pigorini et al., dataset (*Fecchio et al., 2017*) using both high (*Appendix 2—figure 6*) and low (*Appendix 2—figure 7*) voltage motor-evoked potentials data. Similarly to the analyses for the Rogasch's dataset, it is visually evident in these figures that the model accurately captures several individually-varying features of these time series, such as the timing of the 50 ms and 100–120 ms TEP components, and the extent to which they are dominated by left/right and temporal/parietal/frontal channels.

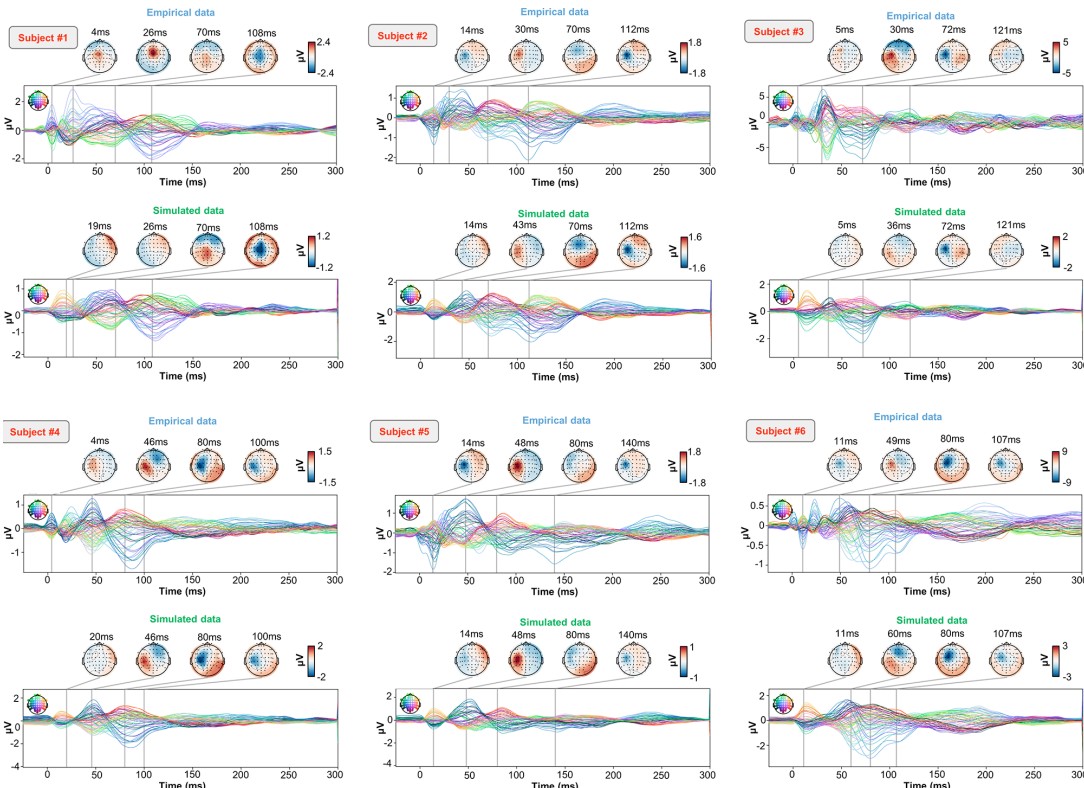

**Appendix 2—figure 6.** Optimized TMS-EEG evoked potential (TEP) models for all subjects of the Fecchio, Pigorini et al., high voltage motor evoked potential dataset. For every pair of rows, empirical (upper) and simulated (lower) TMS-EEG responses are shown for every study subject, validating the results using the Rogasch's data. These data reiterate and reinforce the demonstrations of the main paper that the model-generated electroencephalography (EEG) activity time series achieve robust recovery of individual subjects' empirical TEP propagation patterns.

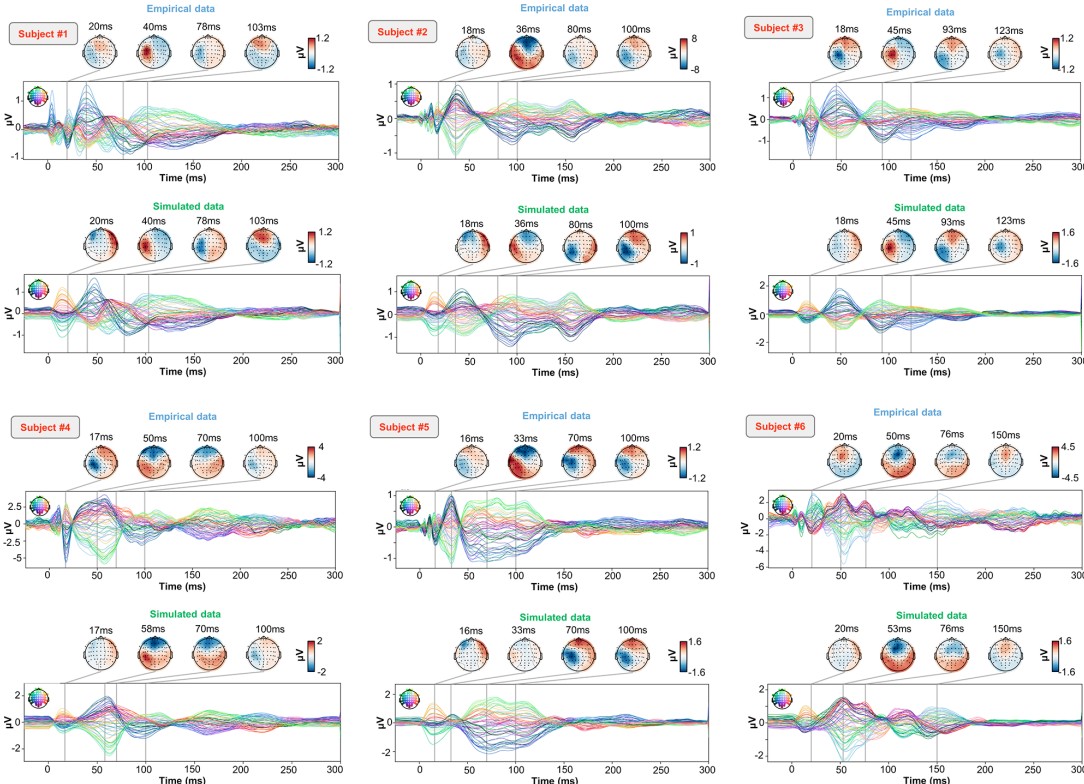

**Appendix 2—figure 7.** Optimized TMS-EEG evoked potential (TEP) models for all subjects of the Fecchio, Pigorini et al., low voltage motor evoked potential dataset. For every pair of rows, empirical (upper) and simulated (lower) TMS-EEG responses are shown for every study subject, validating the results using the Rogasch's data. These data reiterate and reinforce the demonstrations of the main paper that the model-generated electroencephalography (EEG) activity time series achieve robust recovery of individual subjects' empirical TEP propagation patterns.

Similarly to the single-subject fits, an accurate recovery of the grand mean empirical TEP waveform was reported when the fitted TEPs were averaged over subjects (*Appendix 2—figure 8A*). These grand mean channel-level waveforms were further used to assess model fit in brain source space. As shown in Figure *Appendix 2—figure 8B*, the same spatiotemporal activation pattern is observed both for empirical (top row) and model-generated (bottom row) time series. M1 stimulation begins with activation in left motor area at ~20–30 ms, then propagates to temporal, frontal and homologous contralateral brain regions, resulting in a waveform peak at ~100–120 ms. This correspondence between empirical and simulated TEP data is clearly visible both the surface plot and in the bar charts of *Appendix 2—figure 8C and D*.

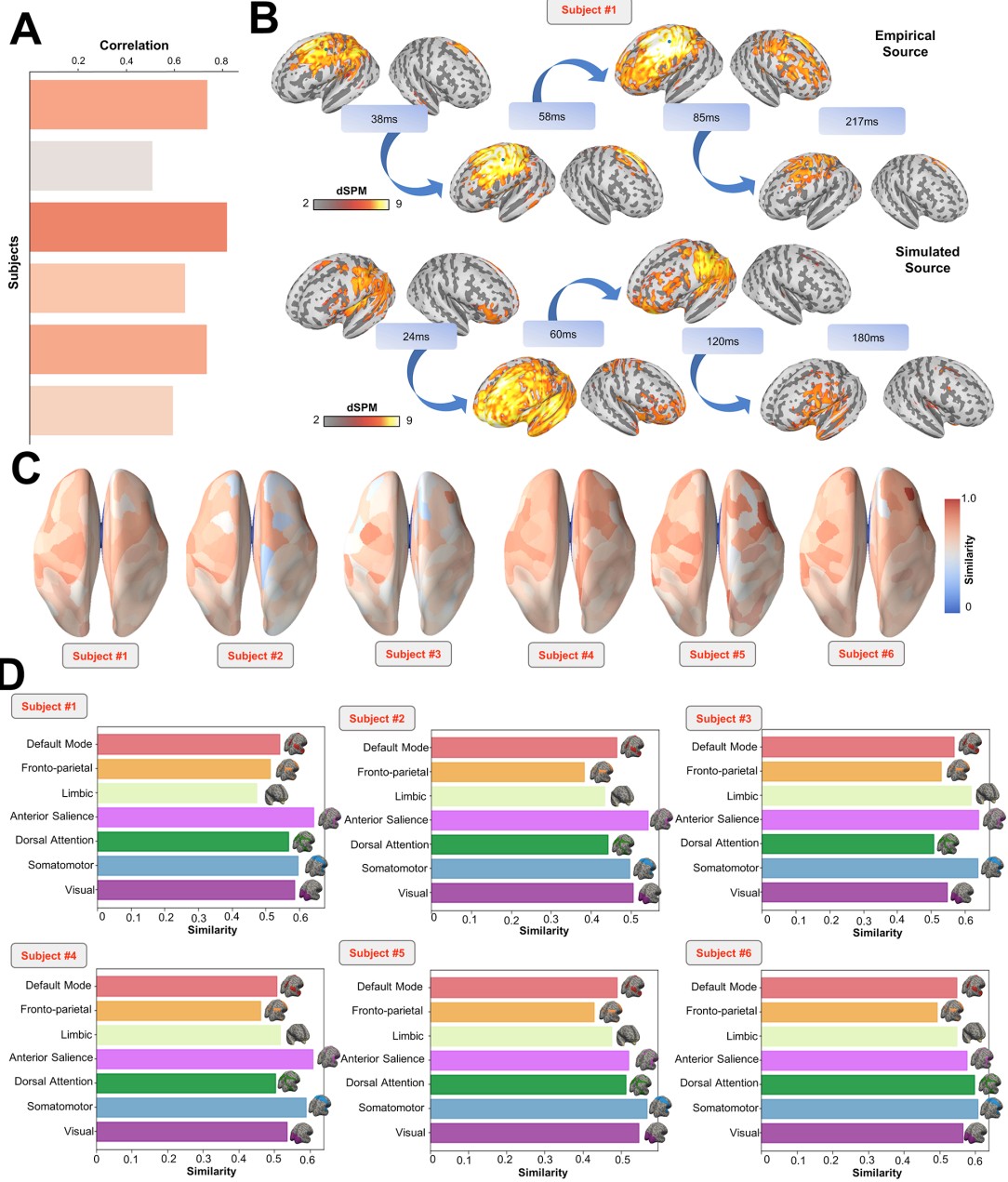

**Appendix 2—figure 8.** Comparison between simulated and empirical TMS-EEG data in source space for the Fecchio, Pigorini et al., dataset. (**A**) Bar plot showing high vertex-wise cosine similarity between empirical and simulated sources for all the subjects. (**B**) Source reconstructed TMS-evoked propagation pattern dynamics for empirical (top) and simulated (bottom) data for one representative subject. (**C**) Parcels-wise cosine similarities plotted on the surface for the entire sample of the Fecchio, Pigorini et al., dataset (*Fecchio et al., 2017*), showing a robust recovery of empirical TMS-evoked patterns in model-generated activity electroencephalography (EEG) time series. (**D**) Network-based cosine similarity for all the subjects. This results validate the findings using the Rogasch's dataset in a completely independent dataset.

Shown in *Appendix 2—figure 9* are effects for one representative subject of the Fecchio, Pigorini et al., dataset (*Fecchio et al., 2017*) on the simulated TEP of virtual lesions to recurrent incoming connections of the main activated regions at 20 ms, 50 ms, and 100 ms after single-pulse TMS stimulation of left M1.

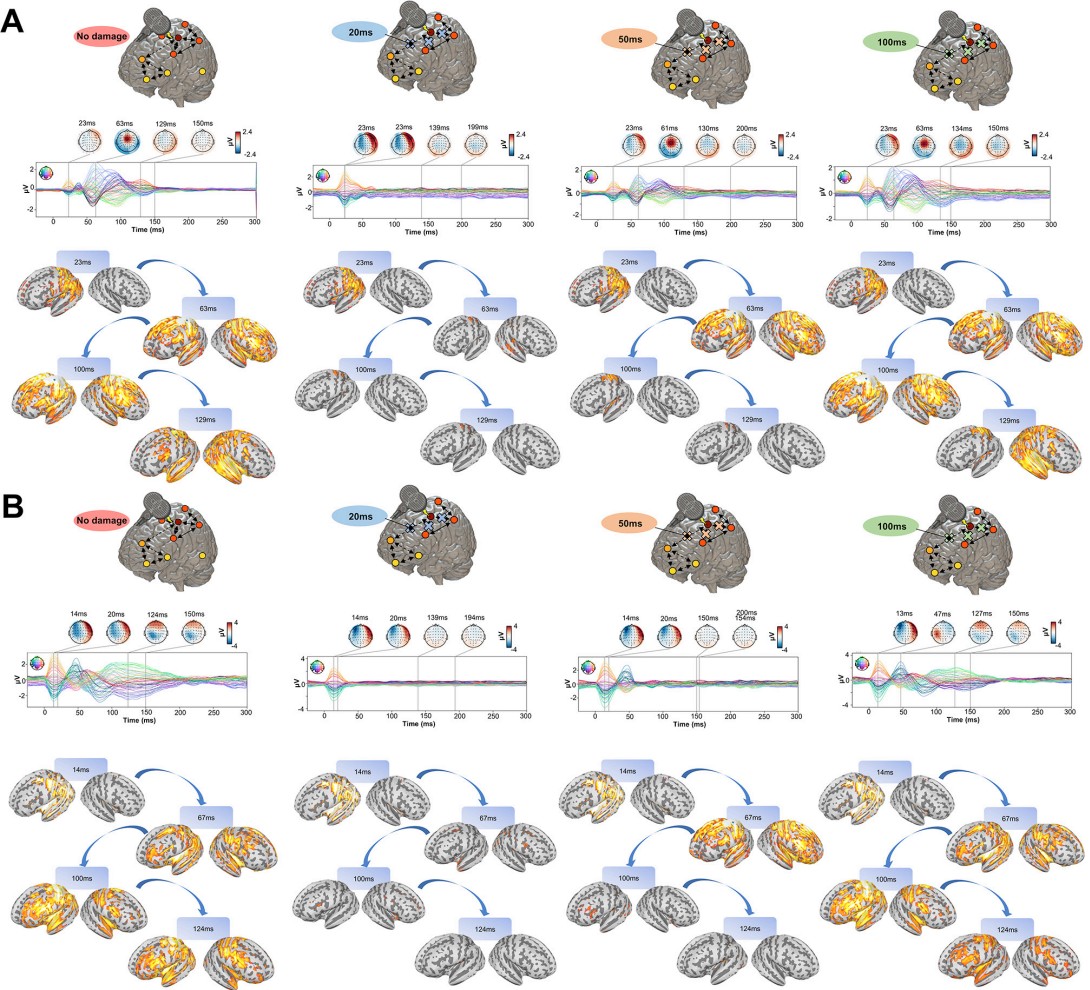

**Appendix 2—figure 9.** Removing recurrent connections from stimulated target nodes suppresses their late TMS-EEG evoked potential (TEP) activity using the dataset of Fecchio, Pigorini et al. TMS-evoked propagation dynamics in the model change significantly depending on the specific time that a virtual lesion is applied. Specifically, early significant reductions in the TMS-evoked activity (50 ms-100 ms time window) were found when important connections were removed at 20 ms (blue) and 50 ms (orange) after the TMS pulse, as compared to both a later virtual lesion (100 ms green) and no damage (red) conditions. This findings was found using both high (Panel A) and low (Panel B) voltage data and replicate the results reported in the main paper and in **Figure 4** using the Rogasch's dataset.

Similarly to the analyses for the Rogasch's dataset, the network-level effects were evaluated statistically by extracting dSPM loadings from the source activity maps for each of the seven canonical Yeo networks (**Thomas Yeo et al., 2011**) and entering them into an ANOVA with factors of NETWORK and TIME OF DAMAGE.

For the high voltage motor evoked potentials data (**Appendix 2—figure 9A**), a significant main effects were found for both TIME OF DAMAGE ($F_{(6,30)} = 229.87$, p<0.0001, $\eta 2P=0.97$) and NETWORK ($F_{(6,18)} = 12.08$, p<0.0001, $\eta 2P=0.71$) - indicating that the effects of virtual lesions vary depending on both the time administered and the site administered to, as well as the combination of these factors (significant interaction NETWORK*TIME OF DAMAGE ($F_{(3,18)} = 10.26$, p<0.0001, $\eta 2P=0.67$)).

For the low voltage motor evoked potentials data (**Appendix 2—figure 9B**), a significant main effects were found for both TIME OF DAMAGE ($F_{(6,30)} = 204.07$, $P<0.0001$, $\eta 2P=0.95$) and NETWORK ($F_{(6,18)} = 11.25$, $P<0.0001$, $\eta 2P=0.68$) - indicating that the effects of virtual lesions vary depending on both the time administered and the site administered to, as well as the combination of these factors (significant interaction NETWORK*TIME OF DAMAGE ($F_{(3,18)} = 9.58$, $P<0.0001$, $\eta 2P=0.64$)).

The fact that the same results were found for both the high and the low voltage motor evoked potentials data indicates that the results observed are not affected by proprioceptive sensory feedback from the periphery.

## No significant relationship between empirical data features and model's parameters

No significant correlation were found between the amplitude of the first and second eigenmode at its peaks and the average synaptic time constant of excitatory population (first eigenmode: $R^2$=0.5%, p=0.75; second eigenmode: $R^2$=0.1%, p=0.99), the local gain from the excitatory interneuron to the pyramidal cell (first eigenmode: $R^2$=9.17%, p=0.19; second eigenmode: $R^2$=4.16%, p=0.38), the local gain from the pyramidal cell to the excitatory interneuron (first eigenmode: $R^2$=0.6%, p=0.74; second eigenmode: $R^2$=4.17%, p=0.38), the local gain from inhibitory interneuron to the pyramidal cell (first eigenmode: $R^2$=4.71%, p=0.35; second eigenmode: $R^2$=0.9%, p=0.67), the local gain from the pyramidal cell to the inhibitory interneuron (first eigenmode: $R^2$=2.2%, p=0.53; second eigenmode: $R^2$=1.4%, p=0.6), the global gain (first eigenmode: $R^2$=4.78%, p=0.35; second eigenmode: $R^2$=0.02%, p=0.91), and the mean input firing rate (first eigenmode: $R^2$=16%, p=0.08; second eigenmode: $R^2$=1.74%, p=0.57).

## 2.4 Relationship between fitted structural connectome and TMS-evoked engagement

In order to replicate the results (*Momi et al., 2021b*), we have extracted brain modularity (*Rubinov and Sporns, 2010*) from the individual fitted connectome This index was computed using the Louvain algorithm (*Blondel et al., 2008*), implemented in the Brain Connectivity Toolbox (*Rubinov and Sporns, 2010*) as follows:

$$Q = \frac{1}{2m} \sum_{ij} \left[ A_{ij} - \frac{k_i k_j}{2m} \right] \delta(c_i, c_j)$$

where m is the sum of all the nodes in the network; $A_{ij}$ is the adjacency matrix representing the edge weight between node i and node j; $k_i$ and $k_j$ are the sum of the weights of the edges attached to nodes i and j, respectively; $\delta(c_i, c_j)$ are the communities of the nodes and is 1 if nodes i and j belong to the same subset of the maximized partition of brain nodes, and 0 otherwise.

As for the TMS-evoked engagement, we extracted the Global Mean Field Power (GMFP) and computed the area under the curve (AUC) for significant post-TMS time points (200 ms after the stimulation).

In accordance with our previous work (*Momi et al., 2021b*), a significant positive correlation ($R^2$=52%, p=0.02) was found between the individual AUC and the modularity of the fitted structural connectome (*Appendix 2—figure 10*).

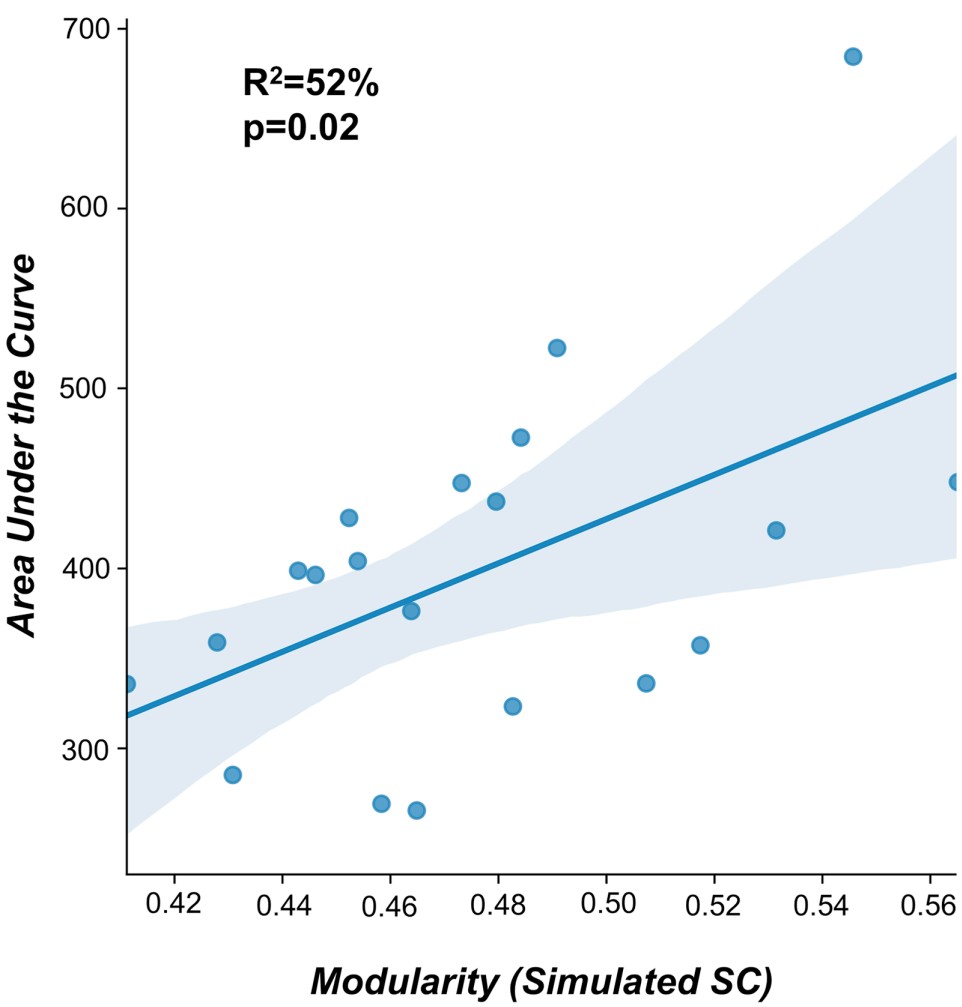

**Appendix 2—figure 10.** Structural connectivity predictor of TMS-EEG propagation. A significant positive correlation ($R^2$=52%, p=0.02) was found between the modularity of the fitted structural connectomes and the area under the curve (AUC) extracted for significant post-TMS time points. This findings replicate the results reported in **Momi et al., 2021b**.

