## [Editor Report]

This important work advances our understanding of the effects of focal perturbations with transcranial magnetic stimulation (TMS) on brain activity. By combining TMS, electroencephalography (EEG), and computational modelling, the authors provide solid evidence to indicate that early EEG signal changes result from local dynamics in the stimulated region whereas later signal changes are influenced by reverberating activity within more broadly connected networks. The work will be of interest to people researching the physiological effects of brain stimulation and biophysical models of large-scale neuronal activity.

---

## [Decision Letter]

**Decision letter after peer review:**

Thank you for submitting your article "TMS-evoked responses are driven by recurrent large-scale network dynamics" for consideration by *eLife*. Your article has been reviewed by 3 peer reviewers, and the evaluation has been overseen by a Reviewing Editor and Michael Frank as the Senior Editor. The following individuals involved in the review of your submission have agreed to reveal their identity: Jil Meier (Reviewer #1); Andrea Pigorini (Reviewer #3).

Essential revisions:

1. The authors state: "Our central question is whether these late responses represent evoked oscillatory 'echoes' of the initial stimulation that are entirely locally-driven and independent of the rest of the network, or whether they rather reflect a chain of recurrent activations dispersing from and then propagating back to the initial target site via the connectome." Since the network is connected, it seems unlikely that any effect can be "independent of the rest of the network", in which case Figure 4B would have been sufficient to show that the later responses to TMS are due to recurrent network dynamics. The authors should provide stronger motivation for their temporal virtual lesion analysis.

2. Regarding structural connectivity: "We set strong priors on the connection weights, such that individual fits allow for a small adjustment of these values." How much variance did you allow? How did the structural matrices look afterwards? There is one example of one subject in Figure S4 but could you also please quantify the differences between the original and the individually fitted structural matrices?

3. "This matrix was prepared numerically for physiological network modelling by rescaling values by first taking the matrix Laplacian, and second by scalar division of all entries by the matrix norm." Which matrix norm is referred to here? Please clarify why the Laplacian was taken as input as this seems like an unconventional normalization.

4. "This thresholded E-field map was then used to inject a weighted stimulus into the target regions in the model." How was this weighting calculated? How were the SimNIBS results transferred into the parcellation?

5. "JR model-simulated TEP ŷ was generated" To which state variable of the model does ŷ refer?

6. How were the Gaussian prior distributions chosen for each of the JR model parameters?

7. Why were the source-space waveforms not assessed on an individual subject level? It seems like the waveforms and Figure 4 should be validated on an individual basis.

8. At what time was the stimulus onset? Was this onset time varied and tested for different correlation outcomes with the empirical time series? It seems that the responses are slightly shifted between empirical and simulated waveforms. Related to this, what was the transient time of the simulation, and was this cut-off? What was the total length of the simulation?

9. Can the authors please comment on the differences in the y-axis ranges between simulated and empirical responses? Can these not also be optimized?

10. It could be quite interesting to also compare the TEPs with a resting-state simulation using the same optimized parameters. This could provide a useful benchmark to compare the stimulus ON periods with stimulus OFF periods and distinguish the stimulus propagation from normal activity spreading in the individual brain networks. If the authors compare their model outcomes also with other whole-brain dynamics benchmarks, e.g. resting-state networks or markers of healthy EEG activity, then this could give increased credibility to the model. Since the authors make claims about whole-brain dynamics, it would be important to validate their model beyond the TEP signature.

11. How much does the result of the optimization algorithm depend on the window length? In my opinion, a 40 ms window seems quite long. Could the authors also test shorter time windows?

12. One of the Reviewers reran the code which is provided by the authors (compliments for sharing!) and noticed that the optimization algorithm has quite varying results, e.g. the first run returned an optimal value of 98.53 and the second one an optimal value for the same parameter of 100.34. Thus, did the authors run the newly developed algorithm multiple times and checked the robustness of the fitting results?

13. The HCP data also offers good quality to extract subcortical regions and their connectivity. These regions could play a major role in TEP stimulus propagation. Why did the authors choose to leave out subcortical regions?

14. The main results hinge on the idea of performing virtual lesions at different times and comparing the activity against the no-lesion case, to see whether activity continues in the absence of recurrent feedback from distant regions. This is an elegant and logical approach. But the implementation requires clarification. From the stated premise and the schematics in Figures 1B and 4A, it seems that the lesions are to isolate a stimulated node. But Figures 4B and 4C appear to use a fully-disconnected network, and the Methods text refers to disconnecting the connections of "maximally activated nodes" at different time points. It is thus difficult to understand what type of lesion was used for what analysis. Please clarify.

15. For the central question of teasing apart recurrent vs local activity, the activation map results in Figure 4A appear to focus on spatial spread rather than the precise question of how much activity at the stimulation site is local vs recurrent. It seems potentially trivial that if you lesion around the stimulation site at 20 ms or 50 ms you don't see spread beyond that site, whereas for a later lesion the delayed stimulus-evoked activity has already reached the other areas (i.e., closing the door after the horse has bolted). Please provide a quantitative calculation of the relative contributions of recurrent vs local activity at the stimulation site. Figures 4B and C may speak to this but the text does not address these.

16. Relatedly, it is unclear how the lesions interact with the propagation delays between regions, which could be important for interpreting the post-lesion activity. In particular, if a signal is "in transit" at the time of lesioning, does that delayed activity still make it to the destination? This likely depends on whether the lesions are modelled as being physically close to the sender or close to the receiver (or both). In any case, one would expect that the delays between regions would determine whether the initial pulse is able to drive activity there before the lesion cuts them off.

17. The fitting seems to do impressively well overall, but not so well at the start (within the first ~50 ms post-stimulus). This is interesting because one might think that the earliest part of the pulse is "simpler" than the more complex patterns possible after receiving multiple variously-delayed feedbacks from the rest of the network (this is of course central to the paper's main topic). Do the authors have insight into why the initial pulse is apparently more difficult to fit?

18. Regarding the role of inhibitory time constants, it is stated that this result is "entirely data-driven" but it is not clear whether relationships were sought with the various other model parameters too. Some correlations are given (Figure 5B) and some additional modelling is used to verify this (Figures 5C and D), but those panels are not addressed in the text so it's difficult to know what to take from these results. The authors give some justification along the lines that TMS protocols have inhibitory effects, but it's not clear why this should be specifically the time constant vs a synaptic strength or some other property of the inhibitory population.

19. As is evident from most of the figures and as mentioned by the authors of the database (see below), the data contains residual sensory artefacts (auditory or somatosensory) that can bias the authors' interpretation of the re-entrant activity. One key limitation is due to the stimulation target. Only the primary motor cortex (M1) is considered here. TMS of M1 – especially at Resting Motor Threshold, as in Biabani et al. – leads to peculiar features in the time and frequency domain of the EEG response, possibly due to feedback from the periphery following target muscle activation – see for example Fecchio, Pigorini et al., Plos One 2017. Are the authors able to address this potential contamination of the signal?

20. On a related note, the data used here are collected by delivering white noise to cover TMS-click instead of a customized one (see Russo et al. JNeurosMethods, 2022). This could introduce the residual auditory activity in the EEG response to TMS that might substantially bias the results and their interpretation in terms of network long-range interactions. For example, residual auditory effects (i.e. n100-p200 component) seem to be present in most of the figures. Can the authors rule out a major contribution of such effects to their findings?

21. Besides the influence of late sensory components, in the same subjects the initial response of the real data seems to be of rather low amplitude and not lateralized, suggestive of a non-effective TMS delivery (see Belardinelli et al., Brain Stim, 2019, for the relevance of this topic). Importantly, in the related simulated data it seems that the initial response is instead present and much higher and lateralized with respect to the real data, possibly reflective of a capacity of the model to replicate the typical TEPs obtained by "effective" TMS. The authors should comment on this potential limitation.

22. The work by Biabani et al. reports that "TEPs elicited by motor cortex TMS reflect a combination of transcranially and peripherally evoked brain responses despite adopting sensory attenuation methods during experiments". This spurious effect is particularly relevant after 60 ms following TMS, and thus the present findings, as well as their interpretation, could be in principle biased by somatosensory re-entrant activity. As such, for the purpose of the present manuscript – i.e. evaluating the late effect of TMS, a dataset in which the residual sensory activity is maximally reduced during data collection is recommended (see Casarotto et al., JNeurosc Methods, 2022 for a full discussion on this topic).

23. The Perturbational Complexity Index (PCI, Casali et al., Science Tr Med, 2013) should be used to evaluate changes in complexity induced by cortical lesions (as in Rosanova, Fecchio, Nat comms 2018 or in Sarasso, Brain, 2020) and not for evaluating the similarity between real and simulated data. Indeed, two completely different wave shapes could in principle have exactly the same PCI value. Instead, for evaluating the quality of the simulated signal (i.e. its similarity with real data), the "natural frequency" may be preferable (Rosanova et al. 2009, JNeurosc).

24. Please clarify the rationale for performing virtual lesions only at 20, 60, and 100 ms. A time-resolved analysis would be preferable.

*Reviewer #1 (Recommendations for the authors):*

Key points:

(i) The authors themselves state that structural connectivity plays an important role in the waveform shapes: "More recently we obtained a similar result with anatomical connectivity (Momi et al., 2021b), namely that network-level anatomical connectivity is more relevant than local and global brain properties in shaping TMS signal propagation after the stimulation of two resting-state networks (again DMN and DAN)." Thus, it would be very interesting to also compare the optimized structural networks among individuals and analyze the relationship between strong structural connections and their individual variability in waveform shapes.

(ii) 26. "Latencies and amplitudes of the SVD left singular vector time series peaks were extracted for every subject and related with the individuals' JR model parameters, with Pearson correlation coefficients and corresponding p-values were computed accordingly." Does this mean that the authors tested correlations of the amplitudes with several of the JR parameters? If the authors ran more tests, one would need to correct for multiple comparisons. Also, it would be great to report the results of these other comparisons in the supplementary material.

*Reviewer #2 (Recommendations for the authors):*

1. I appreciate the authors including Supplementary Figure S4 showing the parameter distributions across the cohort. It's interesting that there is apparently not a great deal of spread in any of the local region parameters – are these fitted values in line with those found by others using the Jansen-Rit model? What dynamical regime does this place the model in? This would give confidence that the model is indeed in a state consistent with the pre-stimulus resting state.

2. The fitting algorithm apparently allows a great many parameters to vary because the connectivity matrix is allowed to vary from the tractography-derived priors. Can the authors give a sense of how much of the fit is due to this huge number of degrees of freedom in the network, versus the local region parameters? And can they defend against the classic charge that a model with so many free parameters can fit anything?

3. Figure 3, it appears that the model fits better in channel space than in source space. Can the authors give an intuition for why this is so?

4. How focal is the stimulation? It is mentioned that SimNIBS is used, but how many regions are stimulated by the TMS pulse?

5. Can the authors justify why TMS is modeled as perturbing specifically the excitatory interneuron population?

6. What's the reasoning for using the matrix laplacian of the connectivity matrix?

7. Equation (4), it appears that Su(t) should be S(v) or S(v(t)). And why the t=0 boundary? I suspect a typo in the denominator too (I think it should be 1+exp instead of 1-exp).

8. Equation (11), bracket missing.

9. Here the modeled EEG signal is estimated from the difference between the excitatory and inhibitory interneuron activity in each region, which at first sounds at odds with the generally accepted view that EEG is most related to pyramidal neurons. I suspect this choice is because in the Jansen Rit model it is those outputs of the interneurons that drive the pyramidal neurons. This should be clarified.

10. lines 647-649, it is unclear how Equation (13) is a measure of complexity; also a typo referring to it as Equation (14).

11. Figure 4C, what's the inset scalp picture?

12. Figure 4C, if this is Local Mean Field Power how can it go negative? And why does it have units of microvolts?

13. Supp. Figure S4 uses different symbols for the parameters vs the main text.

14. Many of the figures pack in so much detail that it is difficult to see what is going on without excessive zoom to read labels etc. Some labeling should be larger, and some figures should potentially be split into multiple if there's no space to increase the size of details within.

15. There should be a space between a number and its units.

*Reviewer #3 (Recommendations for the authors):*

I really enjoyed reading this manuscript. I found it interesting and well-conceived. Below the authors can find a list of issues (along with possible suggestions). In case the authors will properly address them, I would really recommend the publication of this otherwise excellent study on *eLife*.

1) As clearly stated in my public review, most of my issues derive from the data used in the manuscript. My suggestion is to use the open dataset that came with Fecchio, Pigorini et al. (Plos One, 2017), including parietal, premotor, prefrontal, and M1 stimulation. The dataset is available at :

https://figshare.com/articles/dataset/Data_for_Fecchio_Pigorini_et_al_/4970900/1

Using this dataset:

i) You could consider cortices other than M1;

ii) You use data collected with customized noise masking, that minimizes residual auditory evoked potentials (i.e. n100-p200 component, as in Figure S1, Subjects 5, 6, 7, 8, 15… etc);

iii) You use data that maximize the impact of TMS on the cortex;

iv) You can be sure that residual sensory artefacts are maximally reduced during data collection.

For all these reasons, I would really recommend using the above-mentioned dataset (or a similar one). This will be enormously important also for the TMS-EEG community. Indeed, it will allow demonstration, thanks to your modelling approach, that TMS-EEG data can contain late components even in the complete absence of residual sensory artefacts. I understand that re-running the entire analysis over another dataset is an enormous amount of work, but I strongly believe that it is fundamental for a proper interpretation of the results, and to make your study useful for the entire TMS-EEG community.

2) By applying your modelling approach to such a dataset you should also be able to replicate the results reported in Rosanova et al. (JNeurosc, 2009) and Ferrarelli et al. (Arch Gen Psychiatry, 2012) showing that the stimulation of different areas leads to TEPs characterized by different "natural frequencies". Reproducing this feature in simulated data is fundamental to check the quality of simulated data – as a note I would indeed recommend using the natural frequency instead of the Perturbational Complexity Index to evaluate the quality of the simulated data (see also point 5 of the public review), and perhaps PCI to evaluate changes in complexity upon virtual lesion (as in Rosanova, Fecchio et al., Nat Comms, 2020). As a note, I am not 100% sure that a connectome-based approach that accounts for only a few local properties (Jensen and Rit equations) can be able to reproduce the Natural frequency. If it were the case, it would be an incredibly valuable readout of the efficacy of your method; if not, it means that this is just a first step that needs to be perfected (and this can be easily discussed).

3) Related to my point (6) of the public review; if possible and not computationally challenging, I would perform more than 3 lesions over time. Instead of 20, 60, and 100ms, I would try to perform a more time-resolved analysis with one lesion every 5/10 ms, up to 200-300 ms.

4) It is not very clear how the authors performed the statistical comparison between real and simulated data. In addition, in figure 1A, empirical and simulated TEPs exhibit extremely different voltages, so simulated TEPs appear to be remarkably larger than empirical TEPs. How do the authors explain this discrepancy? Since the metric employed by the authors (correlation) is independent of the absolute value of the voltage, it would be important to quantify and report these differences.

5) This is more of a curiosity: what happens to high frequencies (>20Hz) after the lesion? Do you see any suppression as in Rosanova, Fecchio (Nat Comms, 2018)?

[Editors' note: further revisions were suggested prior to acceptance, as described below.]

Thank you for resubmitting your work entitled "TMS-evoked responses are driven by recurrent large-scale network dynamics" for further consideration by *eLife*. Your revised article has been evaluated by Michael Frank (Senior Editor) and a Reviewing Editor.

The manuscript has been improved but there are some remaining issues that need to be addressed. These issues are outlined in the Reviewer comments provided below. These need to be addressed comprehensively before we can consider publication of the manuscript.

*Reviewer #1 (Recommendations for the authors):*

The authors provided a significantly improved revised manuscript and with most of their responses to my comments. I am very happy. However, there are still a few major and minor points that I suggest the authors take a closer look at.

General point:

Often, the authors nicely provide detailed explanations in the rebuttal but fail to also add those explanations to the manuscript. If these refinements were also added to the improved manuscript, then future readers could benefit from the improved clarity directly without additionally having to consult the rebuttal letter. An example is essential revision 3, where it would be great if the authors could also clarify in the text that they used the Frobenius norm.

Essential revision 2:

– I fail to recognize the distance-dependence in the subject-specific SCs. There are many strongly red-colored entries, not just the ones around the diagonal. It seems similar in the original SC, thus I would not take this as a key feature here to judge the fitted SCs.

– In the visualization of the original SC matrix (Appendix 2- Figure 4), one can see that the range of values is [0:0.003], thus allowing a variance of 1/50=0.02 seems very extreme in my opinion.

– Also for better comparison of the original SC with the subject-specific SCs, could the authors please use the same color bar limits for all SCs?

– If the authors allowed such a 1/50 variance, did they only allow this variance in the positive direction or did their prior distribution also allow for negative entries in the SC?

– "We selected a prior variance for each of these connections of 1/50, which was determined empirically to provide a strong constraint but allow for some flexibility during the fitting process." What exactly is meant with „determined empirically" here?

– "It can be seen here that the distribution of the norm distances scores (see image below) acceptable, indicating a relatively small deviation from the prior connectivity weights." It would be much better if the authors describe in words what they calculated here instead of posting the code as a justification of acceptability of the norm distance scores.

Essential revision 4: The x axis ‚brain regions' should probably be relabeled and also add numbers to the ticks please.

Essential revision 6: I applaud the authors for providing such a table with individually optimized parameters, which is an important step towards full reproducibility of their study.

– In the table, there is only one column of variances added for each variable, was this the variance of the prior or posterior distribution or were they the same?

– How can there be a negative variance for the global gain?

Essential revision 12:

The 100 repetition times of their fitting algorithm show a large and to me alarming variability of optimized parameters and warrants further analyses. One solution to deal with this alarming variability would be to please always run the algorithm 100 times (at least 10 I would suggest) and take the mean of its distribution as the final optimized parameter in order to increase robustness of the presented fitting results.

– "For completeness, the prior means for each parameter are also shown as vertical red lines. Shown below are parameter estimate distribution plots for a single subject, repeated 100 times. The vertical red line indicates the original parameters' value (the one in the manuscript) for the same subject." What do the red lines in the plots represent, the prior means or the parameter value chosen for this subject in the manuscript results?

– "Reassuringly, these histograms are approximately normally distributed, and have a tight range of variation that spans only a few percent of the magnitude of each parameter" Even though the variation spans only a few percent of the magnitude of each parameter, the variation for this single subject (e.g. for parameter a: [94:102]) spans nearly over the whole range of optimal parameter values that were fitted over all subjects (e.g. for parameter a: [92:102], see Appendix 2 – Table 1) – or even a larger range than the one fitted over all subjects as in the case of c1. Thus, all results regarding individual variability should be interpreted with caution. The plot for c4 is probably mislabeled as no subject has optimal values in this range according to Table 1.

Essential comment 18:

Results lack significance after Bonferroni correction, therefore they should in my opinion be moved to the Supplementary Material and not be mentioned in the main text as being significant.

*Reviewer #2 (Recommendations for the authors):*

The authors have done a good job in responding to the reviewer comments. In particular it is impressive that they successfully replicated the analysis on an independent dataset. Some residual comments:

1. In general it is preferable that if a clarification is made to the reviewers, it is also made in some way in the text. I won't list them all, but for example the matrix laplacian clarifications don't appear to have made it to the text.

2. Re notation difference, figures and the appendix table refer to parameters a, b, c1, c2, c3, c4, g, but I do not see these in Equations 1-11.

3. Re Equation 4, Su(t) appears to be a typo since the other equations refer to the function S(v).

4. Re Equation 4, why the t=0 boundary?

*Reviewer #3 (Recommendations for the authors):*

First, I would like to congratulate the authors on the incredible amount of work done in order to address all the issues posed by myself and the other reviewers. In particular, I really appreciate the new set of analyses performed on a completely different dataset in order to control for possible confounding factors, such as residual sensory input. In general, the authors have addressed most of the issues I formerly raised, and significantly improved the manuscript with the revised version.

Yet, the first fundamental issue mentioned in my previous review has not been addressed (see point 19 of previous review). The authors mentioned in their response that they "believe that [testing different areas] is beyond the scope of the present manuscript…". I respectfully disagree. In my opinion, the interpretation of the results could be severely biased by the stimulation of one single area (primary motor cortex – M1), having strong efferent connections with periphery. To better clarify my point, by stimulating only M1 it is not possible to conclude that late TEP components are "driven by the recurrent activity within the rest of network" (as stated in the introduction); or at least this is not the only possible interpretation. Indeed, TMS of M1 – especially at Resting Motor Threshold (RMT), as in Biabani et al., or above RMT, as in Fecchio, Pigorini et al. (Experiment_1) – most often leads to muscle twitch, which in turn, induces proprioceptive (sensory) feedback from the periphery. So, TEP components under these experimental conditions are inherently confounded by the peripheral twitch induced by the stimulation. How can the authors address this problem?

In my view, the most straightforward solution implies including other areas in the main analysis (as suggested in my previous review); however, I understand that this implies a huge amount of work, encompassing "diffusion-weighted MRI analysis, TMS-EEG analysis, e-field modeling, deep learning for parameter optimization and neural mass model implementations…". I also understand that "this question is among the scientific priorities" of the authors, which "are currently working on expanding this model further to other stimulation sites".

So in my view, other two – less time consuming – possible solutions to disentangle this problem are:

1) The authors could control whether TEP associated with high voltage and low voltage Motor Evoked Potentials are associated with different late components in the simulated data (data available in Fecchio, Pigorini 2017 – Experiment_2).

2) The authors need to significantly rework the main message of the manuscript by incorporating this confound in the authors' interpretation of their main finding (e.g. the title should be changed because late components could be recurrent large scale network dynamics, but could be also feedback from the periphery…).

I find this second option overly detrimental to the overall significance of the study and I strongly suggest the authors to implement the first in a revision.

Apart of this main issue, I also found a couple of issues the authors may want to address before publication.

1) Perhaps my previous comment about PCI was not clear enough. I did not say that "PCI should never be used for evaluating model fit"; and I fully understand that fitting "could be done with any metric at all that can be computed from empirical and simulated time series". However, while it is true that "the model may be considered a good one to the extent that it captures multiple data features of physiological interest that are reliably quantifiable", it is also true that some metrics are better than others to capture similarities among different time series. As mentioned in my previous review, "two completely different wave shapes could in principle have exactly the same PCI value". The authors can find an example for this in Author response image 1. Top and bottom panels report the butterfly plot (black) and the channel under the coil obtained by stimulating (in the same subject) occipital and prefrontal cortices, respectively. In both cases PCI is 0.47, but the wave shapes highlight major differences both in time and space which are not captured by the single PCI scalar. I hope this example clarifies my former concern and suggest the authors to report metrics other than PCI in the main figures (Pearson correlation coefficients are more than enough).

**Decision letter image 1. sa1fig1:** 

2) Something must be wrong with the axes in Figure S5. See Author response image 2 the authors can find a plot with the correct time and voltage scales for each subject. For example, in Subj_1 the first peak is a 4 ms and not at 38 ms (as in Figure S5).

**Decision letter image 2. sa1fig2:** 

3) Related to my former comment on the original manuscript: "Figure 2B shows a correlation that can be partially driven by the spectral content of the waveforms. It could be relevant to verify whether these correlations are significant with respect to a comparable null hypothesis (e.g. IAAFT, or correlating different empirical and simulated sessions)." I mean that correlation analysis could be strongly affected by low frequencies (characterized by higher power). Thus, a statistical test accounting for different frequencies (e.g., IAAFT) could be preferable (yet not mandatory at this stage).

---

## [Author Response]

Essential revisions:1. The authors state: "Our central question is whether these late responses represent evoked oscillatory 'echoes' of the initial stimulation that are entirely locally-driven and independent of the rest of the network, or whether they rather reflect a chain of recurrent activations dispersing from and then propagating back to the initial target site via the connectome." Since the network is connected, it seems unlikely that any effect can be "independent of the rest of the network", in which case Figure 4B would have been sufficient to show that the later responses to TMS are due to recurrent network dynamics. The authors should provide stronger motivation for their temporal virtual lesion analysis.

We thank the Editor for this considered evaluation of the rationale of our analyses and results interpretation. The recommendation to strengthen the stated motivation for the temporal lesion analysis is a good one, and we are happy to do so. Corresponding text edits are listed below.

In response to the questions:

Since the network is connected, it seems unlikely that any effect can be "independent of the rest of the network"

– We are rather less confident than the Editor that this possibility should be completely dismissed without evaluation (Barabási 2009)

– …but we do nevertheless broadly agree with the Editor’s intuition. To phrase it another way: the question of most interest is not a binary case of recurrence vs. no recurrence, but rather how important is the amount of recurrent feedback, and when are the critical windows for it. This would be a fuller description of the motivation for our temporal virtual lesion analysis.

in which case Figure 4B would have been sufficient to show that the later responses to TMS are due to recurrent network dynamics.

– This is correct, Figure 4C (previous 4B since we added a new panel B) shows the dynamics of a disconnected network. We realize however that this example may be something of a distraction from our main results (which comes up again in Editor Question #14), since the aim of this sub-figure was simply to show visually the effect of the lesion on TEP time series in one example case. We have therefore replaced this figure with the corresponding waveforms for the 20ms lesion condition (see response to Question #14 for full description of this edit).

2. Regarding structural connectivity: "We set strong priors on the connection weights, such that individual fits allow for a small adjustment of these values." How much variance did you allow? How did the structural matrices look afterwards? There is one example of one subject in Figure S4 but could you also please quantify the differences between the original and the individually fitted structural matrices?

We thank the Editor for raising this important point regarding the structural connectome we used in our analysis. Our prior connectivity weights matrix was constructed as the average of 400 healthy young individuals taken from the Human Connectome Project (HCP) (humanconnectome.org/study/hcp-young-adult; Van Essen et al. 2012). We selected a prior variance for each of these connections of 1/50, which was determined empirically to provide a strong constraint but allow for some flexibility during the fitting process. Importantly, after fitting every subject we also conducted visual inspections to confirm that the (posterior mean) connection weight matrices retained the key topological features present in empirical neuroimaging measurements – namely: distance-dependence (strongest values around the main diagonal), within-hemisphere and within-network modular block structure, and (relatively) strong(er) inter-hemispheric connections between homologous brain regions.

Shown in Author response image 1 are the matrices of the posterior mean structural connectivity weights for all subjects from our analysis. We have included this information in the revised supplementary materials section.

**Author response image 1. sa2fig1:** 

As is visually evident here, these parameter estimates differ only very slightly across subjects in their overall spatial structure, indicating that overfitting of subject-specific features is not a major concern.As requested, we have also calculated the norm distances between the original (prior) and the individually fitted structural matrices. Author response image 2 shows the distribution of this metric over study subjects:

It can be seen here that the distribution of the norm distances scores (see Author response image 3) acceptable, indicating a relatively small deviation from the prior connectivity weights.

**Author response image 3. sa2fig3:** 

3. "This matrix was prepared numerically for physiological network modelling by rescaling values by first taking the matrix Laplacian, and second by scalar division of all entries by the matrix norm." Which matrix norm is referred to here? Please clarify why the Laplacian was taken as input as this seems like an unconventional normalization.

This is an important question that we have examined extensively in our present and previous work (see eg pages 337-338 of (John D. Griffiths et al. 2020) for a review of connectivity matrix normalization approaches used in the whole-brain modelling literature). The matrix norm we use here is the Frobenius norm (numpy.linalg.norm(x); see e.g.(Mehta-Pandejee et al. 2017; Robinson et al. 2014)). Other commonly used approaches are scalar division by the maximum weight (placing all entries in the range {0,1} ). Division by the matrix norm is the preferable option here as it ensures that the matrix is linearly stable (i.e. all eigenvalues have negative real part). As to the second question: the matrix Laplacian is not particularly unconventional, it has also been used often in previous whole-brain modelling research (Abdelnour et al. 2018; Atasoy et al. 2016; Raj et al. 2022). The Laplacian sets each row sum (i.e. each node’s weighted in-degree) to be equal (by subtracting row sums from diagonal). The effect of this is that the difference between nodes now becomes a question of the pattern of connections, but not their magnitude. This is related to the rationale followed in eg (Deco et al. 2013, 2014; Schirner et al. 2018), where connection weights as defined by streamline counts are divided by the connecting surface area of a given brain region.

4. "This thresholded E-field map was then used to inject a weighted stimulus into the target regions in the model." How was this weighting calculated? How were the SimNIBS results transferred into the parcellation?

We thank the Editor for giving us the opportunity to expand this important point regarding the computation of the e-field weights.

Please see Author response image 4 for the code cell we used for this part of our analysis:

**Author response image 4. sa2fig4:** 

This code is doing the following:We use volumetric 200-parcel Schaefer atlas (Schaefer et al. 2018) nifti image (https://github.com/ThomasYeoLab/CBIG/tree/master/stable_projects/brain_parcellat ion/Schaefer2018_LocalGlobal/Parcellations/MNI/Schaefer2018_200Parcels_7Netw orks_order_FSLMNI152_2mm.nii.gz) as a mask to extract average values per parcel from E-field maps computed for M1-targeted TMS stimulation.

We have modelled the TMS-induced electric field with SimNIBS (Saturnino et al. 2019) in the MNI152 standard-space. We have imported the 3D arrays of the Schaefer atlas and the E-field into memory in Python. The former has integer values for every entry in the array, corresponding to the parcels of the atlas, whilst the latter has the E-field values. We then find the x,y,z location for each parcel, and use them for extracting the E-field values corresponding to that parcel. We have repeated this process for every parcel, extrapolating the average E-field. The final output (the variable “stim_weights” in the code) is a vector containing the average E-field value for every parcel.

Regarding the E-field map, please see Author response image 5 for a distribution plot with intensity of the electric field (in V/m) over all 200 parcels. The raw average intensity of the electric field was 0.24 V/m with a maximum of 1.95 V/m.

**Author response image 5. sa2fig5:** 

As reported in Line 607, we used 83% of the maximum value to threshold the original E-field distribution (Romero et al. 2019). This thresholded E-field map was then used to inject a weighted stimulus into the target regions in the model which correspond to variable p in equation number 5.

5. "JR model-simulated TEP ŷ was generated" To which state variable of the model does ŷ refer?

We wish to express our gratitude to the Editor for pointing this out. We also wish to apologize for accidentally using the label **y** to refer to two different but related variables: neural mass activity (Lines 611-614) and channel-level TEPs (Lines 639-646). Despite those two entities being highly connected, there is a slight but significant difference between them. For this reason, we have edited the main text, and now **y** only refers to channel-level TEPs (and the corresponding objective function in equation 12).

Line 611: “Finally, channel-level EEG signals were computed in the model by first taking the difference r(t) = v1(t) − v2(t) between the excitatory and inhibitory interneuron activity at each cortical parcel (Schaefer et al. 2018), and projected to the EEG channel space using the leadfield matrix Y = G * X + ϵ.”

We hope that this point is clearer now and thanks again for giving us the opportunity to clarify it.

6. How were the Gaussian prior distributions chosen for each of the JR model parameters?

We wish to thank the Editor for giving us the opportunity to explain this important point regarding the model.

The prior mean values for each of the Jansen-Rit model parameters were taken directly from the original Jansen-Rit paper, where the authors simulated α oscillations in the occipital cortex (Jansen and Rit 1995). Prior variance values for each of these parameters are listed below. As with the connectivity weights prior variances described above, these variances were chosen empirically based on our experience with this and other Bayesian parameter estimation applications.

In the manuscript we have already Figure S4 for an overview of all parameters used in the model. However in order to more fully explain this point to the reader we have decided to add a corresponding Table S1 to the supplementary section, with all the final posterior mean and variances values for every subject:

For convenience, please find below a description of the above model parameters:

a = Reciprocal of the time constant of passive membrane and all other spatially distributed delays in the dendritic network. Also called average synaptic time constant b = Reciprocal of the time constant of passive membrane and all other spatially distributed delays in the dendritic network. Also called average synaptic time constant

c1 = Average probability of synaptic contacts in the feedback excitatory loop. Local

gain from E to P

c2 = Average probability of synaptic contacts in the slow feedback excitatory loop. Local gain from P to E

c3 = Average probability of synaptic contacts in the feedback inhibitory loop. Local

gain from I to P

c4 = Average probability of synaptic contacts in the slow feedback inhibitory loop. Local gain from P to I

g = global gain

k = adjusts the magnitude of EEG signal

mu = Mean input firing rate

We have also edited the corresponding section:

Line 658: “For an overview of all parameters used in the model, please refer to Supplementary Figure S4 and Table S1. For a complete description of the parameter estimation algorithm, please see (John David Griffiths et al. 2022).”

7. Why were the source-space waveforms not assessed on an individual subject level? It seems like the waveforms and Figure 4 should be validated on an individual basis.

We thank the Editor for their insightful suggestion.

We have run the recommended analyses for evaluating goodness of fit in source space at the single subject level.

Please see the new Figure 3 which now includes 3 panels (E, F and G).

Specifically, panel E includes a bar plot showing the cosine similarity between empirical and simulated sources at the level of vertices for all the subjects. In panel F there are the same 3 representative subjects as Figure 2, but with similar scores at the level of the 200 parcels. Finally, in panel G we have reported the cosine similarity at the level of the 7 canonical Yeo networks. As can be seen, despite their differences, all 3 of these analyses show cosine similarity scores > 0.7. This is particularly notable given the level of individual variability in the fitted TEP waveform data.

8. At what time was the stimulus onset? Was this onset time varied and tested for different correlation outcomes with the empirical time series? It seems that the responses are slightly shifted between empirical and simulated waveforms. Related to this, what was the transient time of the simulation, and was this cut-off? What was the total length of the simulation?

This is an important question that we are happy to elaborate on further.

Regarding transient times: The total length of the simulation was 400ms which included -100ms of baseline (pre-TMS) and +300ms after stimulation injection. Prior to the start of the 100ms baseline there was a 20ms burn-in, which allowed for the system to settle after the usual transient due to randomly selected initial conditions. The sufficiency of this burn-in period is evidenced clearly in all of our simulated TEP waveform figures by the lack of any residual large inflection at the start of the prestimulus baseline.

Regarding stimulus onsets and durations: The external input was injected at t=0ms (i.e. exactly the onset of the TMS pulse) for a duration of 10ms, with a simple squarewave on/off pattern. We do not claim that this 10ms square-wave is a particularly faithful replication of the physical field patterns and physiological response of individual neuronal cell membranes to a ~200μs biphasic TMS-pulse, that level of physical detail being largely beyond the scope of the neural mass modelling framework. Rather, the 10ms stimulus duration is best understood as a ‘charging time’ during which reciprocal excitation across multiple stimulated cells reaches the level of population-level activation. We have explored different stimulation durations between ~3-15ms, however the overall model behaviour was little affected by this parameter.

The lag of ~20ms that the reviewer correctly observes is therefore not the result of a delayed stimulation onset, but rather to the rise time of the evoked response at the first few stimulated nodes. The speed of this rise, and peak time of that first inflection, is in part determined by some of the fitted JR parameters, which allow the model to account for variation across subjects in those early TEP component times and waveforms.

As the reviewer may be aware, other neural mass modelling approaches for sensory-evoked responses (e.g. (David et al. 2006)) have elected to inject stimuli into the cortex at large lag times, such as up to ~80ms, so as to account for thalamic relay of sensory inputs. In the case of electromagnetic fields administered directly to the cortex, however, there is (apart from the 10ms charging period indicated above) no physiological rationale for inserting long delay times before stimulus onset, and so we have not done so here.

9. Can the authors please comment on the differences in the y-axis ranges between simulated and empirical responses? Can these not also be optimized?

We thank the Editor for pointing this out.

Whilst we did not explicitly attempt to optimize the exact μV channel-level TEP amplitudes, this should still have been optimized implicitly by the fact that the main term in the objective function of our model is the mean-squared error between empirical and simulated EEG time series.

We certainly do agree that more accurately matching these quantities is a desirable property for future iterations of the model, and will investigate further customization of the objective function to promote this in future work. Relaxing the level of regularization (prior precision) on the lead-field matrix entries, for example, would likely help here, although this comes at the cost of increasing potential overfitting.

10. It could be quite interesting to also compare the TEPs with a resting-state simulation using the same optimized parameters. This could provide a useful benchmark to compare the stimulus ON periods with stimulus OFF periods and distinguish the stimulus propagation from normal activity spreading in the individual brain networks. If the authors compare their model outcomes also with other whole-brain dynamics benchmarks, e.g. resting-state networks or markers of healthy EEG activity, then this could give increased credibility to the model. Since the authors make claims about whole-brain dynamics, it would be important to validate their model beyond the TEP signature.

We agree with the Editor’s suggestion and we strongly agree with their vision. Indeed, this is exactly our goal: building a model that is able to resemble both resting state and evoked activity. Originally, the Jansen and Rit model was employed for simulating α oscillations in the occipital cortex and evoked responses to visual stimuli. Following the request suggested by the Editor’s comment we have tested this by re-running the simulation using the same parameters we have fitted but without injecting the external stimulus. This can be easily achieved by the implementation of the model we have changing the following code line.

u[:,:,110:120]=0

As the Editor can see this line of code simply set the external stimulus to zero.

Please see Author response image 6 for the results for 1 representative subject. As shown in the joint plots and time frequency plots in Author response image 6 showing simulated (left) and empirical (right) prestimulus activity.

**Author response image 6. sa2fig6:** 

As can be seen in Author response image 6, the resting state of our model, shows some similarity to the empirical resting state spectrum on the right, at least in terms of its slope and the presence of some oscillatory peaks. The frequency of the peaks does not match the 10Hz α rhythm evident in the empirical data however. Our feeling is that this result is encouraging (seeing as the parameters used here were obtained from fitting ERP data and not resting-state data), and achieving accurate recovery of individual subject TEP waveforms as well as resting-state oscillations will be a major focus of our future work with this model.

11. How much does the result of the optimization algorithm depend on the window length? In my opinion, a 40 ms window seems quite long. Could the authors also test shorter time windows?

This is an interesting question. We apologize for this typo and indeed the window length was 20ms (not 40ms). We would like to specify that when we have explored this parameter we have invariably found that changing the window size across a sensible range of 15ms-60ms does not dramatically affect the fit results. In our time-windowed model-fitting and epoching approach, the only real relevance of the window length is to maximize the smoothness of the gradient with respect to the model parameters that generate simulation trajectories. Smaller/Larger windows may perform poorly for this reason, and this is an empirical question, but since our model fits are invariably excellent, we do not feel it is one of paramount importance.

It is true that potentially if we used smaller/larger windows we could be more sensitive to some short-time features in the TEP. But there aren’t really very many things in the TEP that are shorter than 20ms in temporal width. Moreover, we do not (yet, at least) wish to claim that the current model accurately reproduces the multiple fine-grained fast and early TEP components in the first few dozen milliseconds post-stimulation. We promptly corrected this missing info in the manuscript.

12. One of the Reviewers reran the code which is provided by the authors (compliments for sharing!) and noticed that the optimization algorithm has quite varying results, e.g. the first run returned an optimal value of 98.53 and the second one an optimal value for the same parameter of 100.34. Thus, did the authors run the newly developed algorithm multiple times and checked the robustness of the fitting results?

We wish to express our gratitude with the Editor for raising this important point. We firmly believe that sharing code and data should be the default rather the exception in science, and particularly so in computationally-focused fields such as ours. It is very validating to see that early sharing of this information can also enhance and enrich the review process, and we greatly appreciate that the Reviewers took the time to download and run the code and repository that we have shared.

Regarding the question of parameter stability and robustness of fitting results: This is an important question that we have studied closely within our model, although we did not discuss at length in the manuscript. Our model is non-deterministic in two key ways – its use of stochastic differential equations at the level of the neural dynamics, and its use of Bayesian inference + stochastic gradient descent at the level of parameter estimation. It also does not rely on consistently specified initial conditions for the states. For these reasons we do indeed expect some variability in parameter estimates upon re-runs. The important question is whether the magnitude of this variability is acceptable. From our testing we feel that this is the case. Shown in Author response image 7 is an example plot of the distributions of posterior mean values for the 7 main Jansen-Rit model parameters, for 100 repeat runs for an exemplary subject. Reassuringly, these histograms are approximately normally distributed, and have a tight range of variation that spans only a few percent of the magnitude of each parameter. For completeness, the prior means for each parameter are also shown as vertical red lines.

Shown in Author response image 7 are parameter estimate distribution plots for a single subject, repeated 100 times. The vertical red line indicates the original parameters’ value (the one in the manuscript) for the same subject.

**Author response image 7. sa2fig7:** 

Finally, although we do not explore it here, we note that it is highly likely that the level of variability shown here could be further reduced by reducing / removing the stochastic terms in the neural model and by fixing initial conditions.Thanks again for allowing us to elaborate this key point!

13. The HCP data also offers good quality to extract subcortical regions and their connectivity. These regions could play a major role in TEP stimulus propagation. Why did the authors choose to leave out subcortical regions?

We wish to thank the Editor for again asking a very important and pertinent question regarding our model.

We agree that it is possible that subcortical regions, particularly the thalamus, may play an important role in TMS-evoked stimulus propagation in the human brain. Indeed, we attempted to include subcortical structures in an earlier version of our model. Specifically, we created an ad-hoc brain parcellation of 231 regions of interest (ROIs) covering cortical, subcortical and cerebellar structures. As stated the first 200 parcels were cortical ROIs from the Schaefer Atlas (Schaefer et al. 2018).

Then, a parcellation into the same networks was also applied for 14 cerebellar ROIs, which were extracted from the Buckner Atlas (Buckner et al. 2011), and a parcellation of the thalamic nuclei was used to derived 7 different ROIs corresponding to Yeo 7 networks (Yeo et al. 2011).

https://www.dropbox.com/s/b0cz132fxva7yox/ThalamusParcellation.zip?dl=0.

Finally, a Freesurfer parcellation was added to this set, giving an additional 12 subcortical structures, including the bilateral basal ganglia (divided into the putamen, caudate and pallidum nuclei), the amygdalae and thalami. This resulted in a 231x231 structural connectivity matrix.

**Author response image 8. sa2fig8:** 

Ultimately however we elected not to include this subcortically-equipped version in the final manuscript, for the following reasons:

1. We did not observe any qualitatively differences in model behaviour or model fit performance in the cortico-subcortical vs. the cortical-only model.

2. Whilst we believe that the above-described approach followed the current stateof-the-art in neuroimaging parcellation and connectome estimation methods, it still suffers from various issues, whose satisfactory resolution was beyond the scope of the present work. For example, anatomical connectivity estimation for subcortical regions from MRI data is well known to be highly problematic. On the one hand, low physical distance between subcortical parcels leads to an over-representation of their inter-connectivity. Conversely, estimating connection strengths between subcortical and cortical regions from tractography is highly challenging due to the much smaller size of the former.

3. Unlike most cortical nodes, the source-localized TMS-EEG signal that is our primary object of study here is not well understood or readily measurable in subcortical regions. Having a large proportion of the (hidden) state variables in our model not contributing to any observed measurements is certainly possible, but would pose additional challenges for our model fitting approach that would not be suitable in the present work, this being the first time we have applied it to the use case of TMS-EEG.

Taking these points together, we have opted in the present work to not attempt to address the necessarily large question of how best to integrate cortico-subcortical contributions to connectome-based models of TEP dynamics. As regards to the neurobiological question itself, an important contribution of the results in our paper is however to offer a ‘computational existence proof’ that subcortical contributions are not *necessary* for an effective model of TMS-EEG dynamics; the observed waveform patterns can indeed be generated by a connectome-based model without complex subcortical wiring.

Strong examples of whole-brain modelling work that incorporates (in different ways) cortico-subcortical connectivity are (Meier et al. 2022; Spiegler et al. 2016, 2020). In the latter case a comprehensive treatment of basal ganglia regions was essential, given the focus of investigation on deep brain stimulation (DBS) and Parkinson’s. We look forward to exploring the extent to which these regions dictate the TMS-EEG response in our future modelling work.

14. The main results hinge on the idea of performing virtual lesions at different times and comparing the activity against the no-lesion case, to see whether activity continues in the absence of recurrent feedback from distant regions. This is an elegant and logical approach. But the implementation requires clarification. From the stated premise and the schematics in Figures 1B and 4A, it seems that the lesions are to isolate a stimulated node. But Figures 4B and 4C appear to use a fully-disconnected network, and the Methods text refers to disconnecting the connections of "maximally activated nodes" at different time points. It is thus difficult to understand what type of lesion was used for what analysis. Please clarify.

We apologize if Figure 1 and Figure 4 were confusing and we are grateful to the Editor for reporting this.

Regarding maximally activated nodes: As stated in the methods, the lesion was applied for disconnecting the connections of maximally activated nodes. In order to do that we identified the nodes that were most affected by the stimulation and virtually lesion their connection at different time points. These maximally activated nodes overlapped 92% with the stimulated nodes as defined by the thresholded E-field maps, but importantly also showed some variability over subjects. We attribute this variability to a combination of heterogeneities in the empirical EEG dataset that we were not able to control for post-hoc, including on the stimulation side (precise TMS coil location relative to the brain; brain size and shape differences), and on the EEG measurement side (brain size and shape differences again, as well as differential location of EEG channels relative to the brain). Extending the lesioned node set to the maximally activated nodes provides, we believe, an efficient and data-driven way to accommodate inter-subject differences in TEP patterns given limited information.

**Author response image 9. sa2fig9:** 

Regarding fully disconnected networks: We included the example of a simulation where the network is fully disconnected in Figure 4C and 4D to help convey the idea behind our methodology, although these model types per se are not a key feature of our results or our argument. We do see now however from the editor’s question (as well as a similar comment in Question #1), that this additional example may be causing more confusion than it is resolving. We have therefore added another panel B where we display a zoom of the source localized EEG activities for the four different conditions showing how early lesions (e.g 20ms and 50ms) compromised the propagation of the TMS-induced signal compared to control (no damage) and late lesion (100ms) conditions (left). Moreover, we also added a bar plot showing a reduced TMS-evoked activity (dSPM area under the curve) when the lesion was applied 20ms and 50ms compared to the others conditions for all the subjects. Importantly we have replaced panels C and D (previous panel B and C) in Figure 4 with the analogous plot from an example subject in our main results set, where the lesion was applied at t=50ms. This replacement figure achieves the original goal (showing channel-level time series in the same subject before and after the virtual lesion), without adding confusion around fully disconnected networks.We thank the Editor again for helping us to sharpen the messaging in this section of the paper.

15. For the central question of teasing apart recurrent vs local activity, the activation map results in Figure 4A appear to focus on spatial spread rather than the precise question of how much activity at the stimulation site is local vs recurrent. It seems potentially trivial that if you lesion around the stimulation site at 20 ms or 50 ms you don't see spread beyond that site, whereas for a later lesion the delayed stimulus-evoked activity has already reached the other areas (i.e., closing the door after the horse has bolted). Please provide a quantitative calculation of the relative contributions of recurrent vs local activity at the stimulation site. Figures 4B and C may speak to this but the text does not address these.

We thank the editor for the perceptive point. The key sub-panels in Figure 4A are those highlighted with an orange background, and within those surface maps the key part is indeed the differential level of left motor cortex activation in the lesioned vs. non-lesioned case. The observation that the latter is not emphasized enough, given its importance for our argument, is a fair one with which we concur.

To address this, we have therefore added a further panel B to Figure 4, which (a) represent the surface maps from Figure 4A with clear emphasis on presence/lack of recurrent activity at M1, and (b) quantify the magnitude of this effect over subjects.

16. Relatedly, it is unclear how the lesions interact with the propagation delays between regions, which could be important for interpreting the post-lesion activity. In particular, if a signal is "in transit" at the time of lesioning, does that delayed activity still make it to the destination? This likely depends on whether the lesions are modelled as being physically close to the sender or close to the receiver (or both). In any case, one would expect that the delays between regions would determine whether the initial pulse is able to drive activity there before the lesion cuts them off.

Again, an important point, which we thank the Editor for raising.

This is likely well-understood by the Editor, but to be clear: as is standard with this type of network model, there is no explicit concept of physical space in relation to signal propagation, such as is used when modelling the transmission of action potentials along axons. Inputs from connected regions ‘appear at’ their pre-synaptic destination after some delay, but do not travel along a spatial pathway to reach that destination per se. More importantly with respect to the Editor’s question: our implementation of the model’s delay-differential equations does not (as is sometimes done) make use of a moving buffer that saves and aggregates delayed inputs. Rather, the history of the state variables themselves are directly indexed according to the time delay matrix (discretized into number of integration steps). As a result, when a lesion is applied, it is applied at the point where network inputs to a region are collated and summed, as opposed to when they are sent from upstream regions. Neurobiologically, this could be understood as a lesion occurring at the location of synapses, or synaptic boutons at the end of axons, as opposed to e.g. at the mid-point of the axon or at the axon hillock of the pre-synaptic cell(s).

17. The fitting seems to do impressively well overall, but not so well at the start (within the first ~50 ms post-stimulus). This is interesting because one might think that the earliest part of the pulse is "simpler" than the more complex patterns possible after receiving multiple variously-delayed feedbacks from the rest of the network (this is of course central to the paper's main topic). Do the authors have insight into why the initial pulse is apparently more difficult to fit?

This is an excellent question. The observation is correct, our model does not perform as well in the initial 50ms post-stimulation. There are likely a few reasons why this is captured less well in our fits is that the later responses. First, the later response components (100-200ms) dominate the signal, in terms of both amplitude and number of channels contributing, and so the fitting algorithm is able to account for more variance in the data by prioritizing those features. An alternative way of stating this is that the characteristic time scale of TEPs (as well as, for that matter, most canonical ERP waveforms) is more on the order of 100-200ms than 25-50ms – with most channels going ‘up/down/up’ over a 300ms period. This is visually apparent in both the empirical and simulated channel-level time series shown below for each subject, and in most TMS-EEG TEP waveform figures in the literature. Related to this, the default (prior mean) Jansen-Rit model parameter set we are using here has an inbuilt ‘preference’ for these slightly longer time-scale ranges – although interesting not because those parameters were selected for ERP models, but rather because they produce rhythmic oscillations in the α frequency range (~10Hz), the period of which is also 100ms.

It would nevertheless be interesting to explore whether this model can successfully reproduce some of the canonical early TEP components in the <50ms window, which could be done by restricting the fitting range to only this first portion of the response. We have not yet attempted this modelling experiment.

18. Regarding the role of inhibitory time constants, it is stated that this result is "entirely data-driven" but it is not clear whether relationships were sought with the various other model parameters too. Some correlations are given (Figure 5B) and some additional modelling is used to verify this (Figures 5C and D), but those panels are not addressed in the text so it's difficult to know what to take from these results. The authors give some justification along the lines that TMS protocols have inhibitory effects, but it's not clear why this should be specifically the time constant vs a synaptic strength or some other property of the inhibitory population.

We thank the editor for this feedback on parameter correlation analyses. Regarding the first point: yes, we tested for correlations between TEP component amplitude and all 7 Jansen-Rit physiological parameters. The strongest effect, and only one that is significant, is that of the inhibitory time constant. Thus it is data-driven in the sense that other patterns *could* have been observed in the fitted parameters’ relationship to TEP components, but they were not. The observation that panels 5C and 5D are not mentioned in the main text is helpful, and we have corrected it.

Line 305: “As shown in Figure 5B, a significant negative correlation was found between the synaptic time constant of the Jansen-Rit inhibitory population and the amplitude of the first eigenmode at its peak (R2 = 27%, p = 0.02). Interestingly, we also observed a significant positive correlation between this parameter and the second eigenmode at its peak (R2 = 28%, p = 0.02). Indeed, as shown in Figure 5C, the topoplots for 2 representative subjects with high (top) and low (bottom) estimated values for the synaptic time constant of the inhibitory population show that the magnitude of the synaptic time constant is closely coupled to the amplitude of the individual first and second SVD modes. Interestingly, as shown in Figure 5D, by varying the optimal value of the inhibitory synaptic time constant, an increase in the amplitude of the first, early, and local TEP component; and a decrease of the second, late, and global TEP component was found.

However, these significant results were not corrected for the Bonferroni corrected pvalue of 0.007. For this reason the reader should critically interpret such results. For a comprehensive overview of individual timing and topographies of the first two eigenmodes, please refer to Supplementary Figures S3”.

As to why this parameter specifically: in the Jansen-Rit model there are two chief parameters of interest that determine the dynamics of the inhibitory population – the inhibitory time constant and, as the editor also notes, the excitatory input strength from the pyramidal cell population. The latter would implicate synaptic weights and glutamatergic neurotransmission more directly than the former. The inhibitory time constant should be understood as controlling the overall level of inhibitory activity in the circuit. Note that it does not (as is implied by the technical term ‘time constant’, and is also the case for simpler systems such as harmonic oscillators) simply continuously shift the (damped, in our case) oscillator frequency – this would require a concurrent shift in both the excitatory and inhibitory time constants (c.f. e.g.(David and Friston 2003), Figure 4; (Spiegler et al. 2010), Figure 9; also discussed in (Stefanovski et al. 2019), page 7); although it can alter the frequency by discretely shifting some nodes in the network to a different point in the bistable regime as defined by its bifurcation diagram (c.f. e.g. Figures 6, 7, 13 in(Stefanovski et al. 2019)). Thus our interpretation of the inhibitory time constant parameter estimates in our analyses is that they may index a natural variation across this population of healthy subjects in the intrinsic level of cortical inhibition, and that this leads to variation in TEP component amplitudes, in a manner broadly consistent with conclusions of previous pharmacological TMS-EEG studies (Premoli et al. 2014).

19. As is evident from most of the figures and as mentioned by the authors of the database (see below), the data contains residual sensory artefacts (auditory or somatosensory) that can bias the authors' interpretation of the re-entrant activity. One key limitation is due to the stimulation target. Only the primary motor cortex (M1) is considered here. TMS of M1 – especially at Resting Motor Threshold, as in Biabani et al. – leads to peculiar features in the time and frequency domain of the EEG response, possibly due to feedback from the periphery following target muscle activation – see for example Fecchio, Pigorini et al., Plos One 2017. Are the authors able to address this potential contamination of the signal?

Please see response to item 22 below.

20. On a related note, the data used here are collected by delivering white noise to cover TMS-click instead of a customized one (see Russo et al. JNeurosMethods, 2022). This could introduce the residual auditory activity in the EEG response to TMS that might substantially bias the results and their interpretation in terms of network long-range interactions. For example, residual auditory effects (i.e. n100-p200 component) seem to be present in most of the figures. Can the authors rule out a major contribution of such effects to their findings?

Please see response to item 22 below.

21. Besides the influence of late sensory components, in the same subjects the initial response of the real data seems to be of rather low amplitude and not lateralized, suggestive of a non-effective TMS delivery (see Belardinelli et al., Brain Stim, 2019, for the relevance of this topic). Importantly, in the related simulated data it seems that the initial response is instead present and much higher and lateralized with respect to the real data, possibly reflective of a capacity of the model to replicate the typical TEPs obtained by "effective" TMS. The authors should comment on this potential limitation.

Please see response to item 22 below.

22. The work by Biabani et al. reports that "TEPs elicited by motor cortex TMS reflect a combination of transcranially and peripherally evoked brain responses despite adopting sensory attenuation methods during experiments". This spurious effect is particularly relevant after 60 ms following TMS, and thus the present findings, as well as their interpretation, could be in principle biased by somatosensory re-entrant activity. As such, for the purpose of the present manuscript – i.e. evaluating the late effect of TMS, a dataset in which the residual sensory activity is maximally reduced during data collection is recommended (see Casarotto et al., JNeurosc Methods, 2022 for a full discussion on this topic).

We wish to express our gratitude to the Editor (along with the other reviewers) for raising these important points regarding the dataset we have used in our study. We do concur that it is possible there are some residual auditory/somatosensory evoked potential sources in the data used here. This was unfortunately largely out of our control, given that we neither collected nor preprocessed the data. We do note however that the dataset in question has (at least until very recently) represented the state-of-the-art in terms of TMS-EEG preprocessing and analysis (the publication source is indeed a data preprocessing methodology paper) – having been produced by one of the strongest groups in the field, and used for multiple publications in leading journals (Biabani et al. 2019; Rogasch et al. 2020).

Nevertheless, being active in the TMS-EEG community, we do recognize that the questions raised are important ones. We have therefore run the re-run the analysis using the dataset suggested by Reviewer#3 ((Fecchio et al. 2017)- https://figshare.com/articles/dataset/Data_for_Fecchio_Pigorini_et_al_/4970900/1 ).

Our goal was to replicate the results we obtained with the Rogash group dataset, rejecting in this way the null hypothesis that our previous results were due to sensory artifact. In this way, our interpretation of the re-entrant activity should not be biased by the residual sensory artifacts (auditory or somatosensory).

Please see the new Appendix 2-figure 6 showing the Empirical (upper row) and simulated (lower row) TMS-EEG butterfly plots with scalp topographies for all 6 subjects of the Fecchio and Pigorini’s dataset. Results show a robust recovery of individual empirical TEP patterns in model-generated activity EEG time series.

Next, please see the new Appendix 2-figure 8. Similarly to Figure 2B of the manuscript, in Panel A there are the Pearson correlation coefficients between simulated and empirical TMS-EEG time series for each subject. Panel B shows the source reconstructed TMS-evoked propagation pattern dynamics for empirical (top) and simulated (bottom) data for one representative subject of Fecchio and Pigorini’s dataset. Panel C represents the similarity between simulated and empirical time series in source space for every subject of the Fecchio-Pigorini dataset, showing a robust recovery of TEP patterns in model-generated EEG time series. Panel D shows the network-level Pearson correlation coefficients between empirical and simulated source-reconstructed time series extracted for every subject.

Finally, we have also re-run the analysis for the virtual lesion approach using the Fecchio-Pigorini dataset. Please see the new Appendix 2-figure 9 showing how removing recurrent connections from stimulated target nodes suppresses their late TEP activity in the same way we found for the Rogash group dataset.

In the same way as our analyses of the Rogash dataset in the manuscript, the network level effects were evaluated statistically by extracting dSPM loadings from the source activity maps for each of the 7 canonical Yeo networks (Yeo et al., 2011) and entering them into a 4x7 repeated measures ANOVA with within-subjects factors “TIME OF DAMAGE (4 levels: 20ms; 50ms; 100ms; no damage) and “NETWORK” (7 levels: VISN; SMN; DAN; ASN; LIMN; FPN; DMN) Significant main effects were found for both TIME OF DAMAGE (F(6,30) = 229.87, p<0.0001, η2p = 0.97) and NETWORK (F(6,18) = 12.08, p < 0.0001, η2p = 0.71) – indicating that the effects of virtual lesions vary depending on both the time administered and the site administered to, as well as the combination of these factors (significant interaction NETWORK*TIME OF DAMAGE (F(3,18) = 10.26, p < 0.0001, η2p = 0.67)).

The fact that the results we have reported for the Rogash group dataset were replicated and extended in the Fecchio-Pigorini dataset indicates that, despite the suggestion that the first dataset does contain residual sensory artifacts (auditory or somatosensory), our interpretation of the re-entrant activity is unaffected by those artifacts.

We edited the main manuscript accordingly to reflect the above points:

Line 700: “Finally, as above, the model-generated dSPM values were extracted from the 7 canonical network surface maps for each individual and for each condition, and analyzed statistically using the Statistical Package for the Social Sciences (SPSS) Version 25 (IBM Corp). A 4x7 repeated measures ANOVA with within-subjects factors “TIME OF DAMAGE (4 levels: 20ms; 50ms; 100ms; no damage) and “NETWORK” (7 levels: VISN; SMN; DAN; ASN; LIMN; FPN; DMN) was run. Post-hoc paired t-tests were used to detect dSPM value changes for different networks and lesion times, testing on a per-network basis whether and at what times the virtual lesions impacted on network-level activations.

In order to control for possible confounding factors such as the residual sensory artifacts (auditory or somatosensory) that can completely bias the interpretation of the re-entrant activity, we have run the same analysis in another independent open dataset

(https://figshare.com/articles/dataset/Data_for_Fecchio_Pigorini_et_al_/4970900/1)( Fecchio et al., 2017). For a complete overview of this control analyses and the corresponding results, please refer to Supplementary Results 2.2 and Supplementary Figures S5, S6, S7.”

We have also added a new section as Supplementary Results (2.2 Control analyses). Figure S5 shows empirical and fitted TEP waveforms and selected topography maps for the entire group of the Fecchio, Pigorini et al. dataset (Fecchio et al. 2017). Similarly to the analyses for the Rogasch’s dataset, it is visually evident in these figures that the model accurately captures several individually-varying features of these time series, such as the timing of the 50 ms and 100-120 ms TEP components, and the extent to which they are dominated by left/right and temporal/parietal/frontal channels.

Similarly to the single-subject fits, an accurate recovery of the grand mean empirical TEP waveform was reported when the fitted TEPs were averaged over subjects (Figure S6A). These grand mean channel-level waveforms were further used to assess model fit in brain source space. As shown in Figure S6B, the same spatiotemporal activation pattern is observed both for empirical (top row) and model generated (bottom row) time series. M1 stimulation begins with activation in left motor area at <inline-graphic mimetype="image" mime-subtype="png" xlink:href="media/image1.png" />20-30 ms, then propagates to temporal, frontal and homologous contralateral brain regions, resulting in a waveform peak at <inline-graphic mimetype="image" mime-subtype="png" xlink:href="media/image2.png" />100-120 ms. This correspondence between empirical and simulated TEP data is clearly visible both the surface plot and in the bar charts of Figure S6C, D.

Shown in Figure S7 are effects for one representative subject of the Fecchio, Pigorini et al. dataset (Fecchio et al. 2017) on the simulated TEP of virtual lesions to recurrent incoming connections of the main activated regions at 20 ms, 50 ms, and 100 ms after single-pulse TMS stimulation of left M1.

Similarly to the analyses for the Rogasch’s dataset, the network-level effects were evaluated statistically by extracting dSPM loadings from the source activity maps for each of the 7 canonical Yeo networks (Yeo et al. 2011) and entering them into an ANOVA with factors of NETWORK and TIME OF DAMAGE. Significant main effects were found for both TIME OF DAMAGE (F_(6,30)_ = 229.87, p<0.0001, η^2^_p_ = 0.97) and NETWORK (F_(6,18)_ = 12.08, p < 0.0001, η^2^_p_ = 0.71) – indicating that the effects of virtual lesions vary depending on both the time administered and the site administered to, as well as the combination of these factors (significant interaction NETWORK*TIME OF DAMAGE (F(3,18) = 10.26, p < 0.0001, η^2^p = 0.67)).

As for the Editor’s point regarding testing different cortical sites, we absolutely embrace his view. However, although we agree that this is an important consideration, we believe that is beyond the scope of the present manuscript, given the various complexities involved that include diffusion-weighted MRI analysis, TMS-EEG analysis, e-field modelling, deep learning for parameter optimization and neural mass model implementations. Despite this, we would like to further note that this question is in fact our primary scientific priority at the moment; and indeed we are currently working on expanding this model further to other stimulation sites.

23. The Perturbational Complexity Index (PCI, Casali et al., Science Tr Med, 2013) should be used to evaluate changes in complexity induced by cortical lesions (as in Rosanova, Fecchio, Nat comms 2018 or in Sarasso, Brain, 2020) and not for evaluating the similarity between real and simulated data. Indeed, two completely different wave shapes could in principle have exactly the same PCI value. Instead, for evaluating the quality of the simulated signal (i.e. its similarity with real data), the "natural frequency" may be preferable (Rosanova et al. 2009, JNeurosc).

We thank the Editor for raising up this important point and for giving us the opportunity to elaborate around choosing Perturbational Complexity Index (PCI,(Casali et al. 2013)) as a metric for evaluating the similarity between real and simulated data.

In regards to the Editor’s comments:

1. Respectfully, we disagree with this blanket injunction that PCI should never be used for evaluating model fit. In principle this could be done with any metric at all that can be computed from empirical and simulated time series, and the model may be considered a good one to the extent that it captures multiple data features of physiological interest that are reliably quantifiable.

2. The multiple realizability issue that the Editor raises also applies just as equally to other standard metrics such as variance, covariance, correlations, for which various pattern combinations could produce similar values.

3. Indeed, for similar reasons to these concerns, we have not limited ourselves to only reporting PCI in comparing the similarity between empirical and modelgenerated time series, but instead we have used multiple metrics. None of our arguments in the paper are dependent on the simulated time series PCI values. In the manuscript we report PCI scores, Pearson correlation coefficients for comparing the simulated and empirical TMS-EEG time series for each subject (Figure 2B), cosine similarities, as well as run a time-wise permutation tests yielding with statistically significant fits for every EEG electrode (Figure 3C).

4. Quantification aside, the simulated TEP waveforms we present are quite clearly very visually similar to the modelled empirical data. To assist with this, we have added additional information to the manuscript. First of all, in Figure 2A we show the individual empirical and simulated TEPs patterns for 3 representative subjects. Moreover, in Supplementary Figure S1 we extended Figure 2A and we reported all the subjects empirical and simulated TMS-EEG activity. Additionally, as requested by the Editor and Reviewer #3’s later comment, we have validated our model in a completely independent open dataset Fecchio, Pigorini et al. (Fecchio et al. 2017). The results for this new dataset for every subject are now available as Appendix 2-figure 1.

5. Regarding the Editor’s final comment on natural frequency – this is indeed an interesting experimental observation that we are keen to investigate further and replicate in our model. It is not however (unlike the TMS-EEG TEP waveform itself, which we have focused on here) a conventional metric studied in the field, and so is not (yet) a major priority in our model fitting evaluations.

24. Please clarify the rationale for performing virtual lesions only at 20, 60, and 100 ms. A time-resolved analysis would be preferable

We wish to thank the Editor for pointing out this important doubt regarding the choice of the time point for lesioning the structural connectome. The rationale behind our selection was based on previous literature, which is robust in terms of the TEP waveforms following M1 stimulation. Indeed, each TEP elicited by a single TMS pulse delivered to M1 is actually composed of a sequence of negative and positive deflections (mainly N15, P30, P60, N120). Despite there being little understood and/or definitely established in terms of the physiological mechanisms underlying each component of the TEP, several studies have suggested that the early components N15 and P30 reflect fast excitatory mechanisms (Mäki and Ilmoniemi 2010; Rogasch et al. 2013). The P60 component has also been proposed to reflect somatosensory return (Paus et al. 2001), and the N120 has been associated with the activation of GABA-B receptors (Bonnard et al. 2009; Nikulin et al. 2003; Premoli et al. 2014).

One of our main goals in the present work was to disentangle the physiological origin of these waveforms, and for this reason we have decided to apply a virtual lesion slightly before the key time point. We have deliberately omitted the N15 because applying a lesion before that time point would be completely infeasible due its proximity to the TMS pulse. Moreover, the first 14ms after the TMS pulse were zero padded and then interpolated, making any inference on the N15 difficult to interpret.

As for investigating the P30, P60 and N120 we have applied the virtual lesion 20ms, 50ms or 100ms after the pulse. Our results show that, as compared to both a later virtual lesion (100ms) and no damage (control) conditions, when important connections were removed at 20ms and 50ms after the TMS pulse, there was an early significant reductions in the TMS-evoked activity of the P60 and N120 signal components, but not in P30. This indicates that P30 is mainly due to a local echo of the stimulus, such that even an early lesion (20ms) doesn’t affect the overall generation and shape of this waveform. Conversely at 80ms (P60 has some degree of freedom, +/-20ms) we have found a complete essence of propagation in the condition where the lesion was applied before that time frame (20ms and 50ms after the pulse). That means that P60 might be due to a network reverberation, and not a local node echo. Finally, as for the N120: we have found both an essence of propagation to ipsilateral and homologous brain regions (like in the control condition) and a local residual of the stimulus. This indicates that the N120 can be due to both local and global effects.

Overall, we found that TMS-evoked propagation dynamics in the model change significantly depending on the specific time that a virtual lesion is applied, providing insight on the physiological origin of the TEPs and their underlying dynamics.

Despite this, as requested by Reviewer #3, we have run the lesions analysis again for one subject with one lesion every 10ms, up to 200ms after the TMS pulse. Please see in Author response image 10 an overview of the results for this analysis:

**Author response image 10. sa2fig10:** 

As can be seen, early lesions (e.g. 20ms and 50ms) compromised the propagation of the TMS-evoked activity; whereas for a later lesion the delayed stimulus-evoked activity has already reached the other areas (i.e., ‘closing the door after the horse has bolted’, to borrow the Editor’s expression). Despite the necessary loss of some information due to ‘downsampling’, we believe that our choice of 20, 50, and 100ms as a window for showing the propagation changes due to virtual lesions does summarize this complexity well.We hope that the Editor would appreciate our efforts in showing the result for the analysis asked for and we hope he would agree regarding the time windows we have used for showing the results in the main manuscript.

Reviewer #1 (Recommendations for the authors):Key points:(i) The authors themselves state that structural connectivity plays an important role in the waveform shapes: "More recently we obtained a similar result with anatomical connectivity (Momi et al., 2021b), namely that network-level anatomical connectivity is more relevant than local and global brain properties in shaping TMS signal propagation after the stimulation of two resting-state networks (again DMN and DAN)." Thus, it would be very interesting to also compare the optimized structural networks among individuals and analyze the relationship between strong structural connections and their individual variability in waveform shapes.

We thank the Reviewer for this compelling comment and interesting idea. Indeed, the relationship between the structural connectome and how its topology shapes the propagation of the TMS-induced signal is of great interest and we are definitely motivated in investigating this point further.

In accordance with the flow suggested by the Reviewer’s comment, we have undertaken the following additional analyses:

We first extracted the fitted structural connectome and then computed the Louvain modularity by using the following line of code.

**Author response image 11. sa2fig11:** 

With this cell, we have simply looped over all subjects and collected the modularity for the fitted structural connectome using the Louvain algorithm (Blondel et al. 2008), implemented in the Brain Connectivity Toolbox (Rubinov and Sporns 2010), which is defined as follows:Q=12m∑ij[Aij−kikj2m]δ(cicj) where m is the sum of all the nodes in the network; A_ij_ is the adjacency matrix representing the edge weight between node i and node j; k_i_ and k_j_ are the sum of the weights of the edges attached to nodes i and j, respectively; δ(c^i^,c_j_) are the communities of the nodes and is 1 if nodes i and j belong to the same subset of the maximized partition of brain nodes, and 0 otherwise.

Following this we have collected the fitted eeg using the cell in Author response image 12.

**Author response image 12. sa2fig12:** 

We then have computed the Global Mean Field Average (GMFA) and extracted the area under the curve (AUC) from the TMS period.

**Author response image 13. sa2fig13:** 

Finally we have correlated the individual AUC and the modularity of the fitted structural connectome. See Appendix 2—figure 10.As can be seen, this analysis returned a correlation between the fitted connectome modularity and TMS AUC with an R2=52% which was also significant p=0.02.

The fact that we were able to replicate our previous result (Momi et al. 2021) in this model-generated data is of great interest to us and deserves further investigation, which we shall pursue in future work. What is particularly notable here is that the connectome modularity values in the above figure, which are fitted on an individual subject basis using their TMS-EEG TEP waveforms and the same group-average connectivity priors (i.e. no individual-subject DWI tractography information) show the same relationship that we have previously reported when comparing individual-subject DWI tractography and TMS-EEG data. This result supports the claim that our parameter estimation correctly recovers inter-subject variation in structural connectivity. New prospective studies including a predefined assignment to a low and high modularity group will be needed to further verify this relationship between TMS responses and connectome modularity.

We have added a Supplementary Results section (2.4 Relationship between fitted structural Connectome and TMS-evoked engagement) when these results are reported. We wish to thank the Reviewer again for this comment and idea!

(ii) 26. "Latencies and amplitudes of the SVD left singular vector time series peaks were extracted for every subject and related with the individuals' JR model parameters, with Pearson correlation coefficients and corresponding p-values were computed accordingly." Does this mean that the authors tested correlations of the amplitudes with several of the JR parameters?

Thanks for raising this point. We confirm that we have tested the correlation using several JR params. We have edited the manuscript accordingly for clarifying this point.

Line 727: “Latencies and amplitudes of the SVD left singular vector time series peaks were extracted for every subject and related with the individuals’ JR model parameters, with Pearson correlation coefficients and corresponding p-values were computed accordingly. The critical p-value was then adjusted using Bonferroni correction to account for multiple comparisons (corrected p-value=0.007).”

If the authors ran more tests, one would need to correct for multiple comparisons.

We wish to thank the Reviewer for pointing this out. Indeed we have calculated the relationship between the individual’s latencies and amplitudes of the SVD left singular vector time series and the model parameters by running multiple tests. For this reason we should have corrected for multiple comparisons as suggested by the Reviewer’s comment. Unfortunately the significant correlations we found between the synaptic time constant of the Jansen-Rit inhibitory population and the amplitude of the first (R^2^ = 27%, p = 0.02) and the second (R^2^ = 28%, p = 0.02) eigenmode at their peaks did not survive the multiple comparisons correction (Bonferroni corrected critical p-value=0.007). We have now explicitly stated in the manuscript that the results were not corrected for multiple comparisons and for this reason the reader should critically interpret such results:

Line 299: “As shown in Figure 5B, a significant negative correlation was found between the synaptic time constant of the Jansen-Rit inhibitory population and the amplitude of the first eigenmode at its peak (R2 = 27%, p = 0.02). Interestingly, we also observed a significant positive correlation between this parameter and the second eigenmode at its peak (R2 = 28%, p = 0.02). Note however these planned comparison results did not exceed the (N=7) Bonferroni-corrected critical p-value of 0.007. For this reason it is important that the reader critically interpret our reported results. For a comprehensive description of individual timing and topographies of the first two eigenmodes, please refer to Supplementary Figures S3.”

Line 727: “Latencies and amplitudes of the SVD left singular vector time series peaks were extracted for every subject and related with the individuals’ JR model parameters, with Pearson correlation coefficients and corresponding p-values were computed accordingly. The critical p-value was then adjusted using Bonferroni correction to account for multiple comparisons (corrected p-value=0.007).”

Also, it would be great to report the results of these other comparisons in the supplementary material.

We wish to thank the Reviewer for their insightful comment, and we agree that a detailed overview of the non-significant relationship between empirical data features and model’s parameters would be important and will contribute to making the paper more comprehensive.

For this reason we added a Supplementary Results section (2.3) where we have reported all the no significant results of the other comparisons.

Moreover we have edited the manuscript accordingly:

Line 304: “For a comprehensive overview of individual timing and topographies of the first two eigenmodes, please refer to Supplementary Figures S3. For a detailed description of the no significant relationship between empirical data features and model’s parameters, please refer to Supplementary Results 2.3.”

Reviewer #2 (Recommendations for the authors):1. I appreciate the authors including Supplementary Figure S4 showing the parameter distributions across the cohort. It's interesting that there is apparently not a great deal of spread in any of the local region parameters – are these fitted values in line with those found by others using the Jansen-Rit model? What dynamical regime does this place the model in? This would give confidence that the model is indeed in a state consistent with the pre-stimulus resting state.

We thank the reviewer for the interesting question. Please see Author response image 1 for reference listing Jansen-Rit model parameters reported in previous studies.

**Author response table 1. sa2table1:** 

	a	b	C1	C2	C3	C4	g
Our paper	97.78305	49.85237	1352205	109.2454	31.65449	32.72027	1001.478
Stefanovski 2019	100	50	135	108	33.75	33.75	0<g>600
Jansen and Rit 1995	100	50	135	108	33.75	33.75	n/a
Ableidinger et al., 2017	100	50	135	108	33.75	33.75	1000
Ahmadizadeh et al., 2018	100	50	135	108	33.75	33.75	1000

We have decided to add a corresponding Supplementary file 1 to the supplementary section, with all the final posterior mean and variances values for every subject:

For convenience, please find below a description of the above model parameters:

= Reciprocal of the time constant of passive membrane and all other spatially distributed delays in the dendritic network. Also called average synaptic time constant= Reciprocal of the time constant of passive membrane and all other spatially distributed delays in the dendritic network. Also called average synaptic time constant

c1 = Average probability of synaptic contacts in the feedback excitatory loop. Local

gain from E to P

c2 = Average probability of synaptic contacts in the slow feedback excitatory loop. Local gain from P to E

c3 = Average probability of synaptic contacts in the feedback inhibitory loop. Local

gain from I to P

c4 = Average probability of synaptic contacts in the slow feedback inhibitory loop. Local gain from P to I

g = global gain

k = adjusts the magnitude of EEG signal

mu = Mean input firing rate

Regarding the question of whether our fitted model parameters place the system in a similar dynamical regime to resting state activity – please see our response to Editor’s question 10.

2. The fitting algorithm apparently allows a great many parameters to vary because the connectivity matrix is allowed to vary from the tractography-derived priors. Can the authors give a sense of how much of the fit is due to this huge number of degrees of freedom in the network, versus the local region parameters? And can they defend against the classic charge that a model with so many free parameters can fit anything?

This is indeed an important question, and one we have been very mindful of in conducting this work.

We would like to first highlight that we have examined this question of the contribution of connectivity weights vs. neurophysiological model parameters to fitting performance in our other recent work (John David Griffiths et al. 2022), (doi.org/10.1101/2022.05.19.492664, also under review). As can be seen in that paper (compare e.g. Figure 3C vs. Supplementary Figure 2A), we showed that including strongly constrained connectivity matrix parameter estimation does not substantially alter estimation of physiological neural mass model parameters, or the behaviour of the model when weights are fixed and those parameters are used alone.

Unlike Griffiths et al. 2022, which is a technical methodology paper introducing the parameter estimation approach used in the present paper, here we are principally concerned with the neurobiological question of recurrent activity. Our approach to connectivity matrix fitting was identical however; namely to avoid this issue we have strongly constrained the model by placing relatively tight priors on the connectivity matrix weights. As can be seen in our response to the Editor’s Question #2, the structure of the resultant fitted connectivity matrices does not vary hugely across subjects, even though the TEP waveforms themselves are highly variable across subjects. There is some individual variation however, and as noted in the above response to Review #1 Question (i), variation in the modularity structure of the fitted weights does in fact replicate previously reported results on the relationship between empirical structural connectivity and TMS-EEG measurements. We are therefore quite confident that the primary results of interest here are not heavily determined by the estimation of connectivity weight parameters, but that the patterns in these recovered parameters do nevertheless express biologically meaningful non-arbitrary (i.e. not over-fitted) structure.

3. Figure 3, it appears that the model fits better in channel space than in source space. Can the authors give an intuition for why this is so?

We thank the Reviewer for this compelling question. We note that in our model fitting framework, we are fitting TMS-EEG times series at the level of the channel. The source EEG is then derived for the simulated EEG time series. For this reason, it is expected that the model fits better in channel space than in source space, also considering the nature of the inverse problem which is by definition “illposed” (e.g. different source eeg configuration can give rise to the same channel level topoplots).

We hope that we clarify this important point better and thanks for the opportunity to discuss this further.

4. How focal is the stimulation? It is mentioned that SimNIBS is used, but how many regions are stimulated by the TMS pulse?

We thank the Reviewer for pointing out this question which gives us the opportunity to elaborate and discuss this further.

Regarding the focality of the stimulation, TMS offers the possibility of stimulating a small portion of cortical grey matter (i.e., ~ 1 cm3). As mentioned in the paper, the TMS-induced electric field was modelled with SimNIBS (Thielscher et al., 2015). In the following cell of code we simply import the E-field map (generated via SimNIBS) and the Schefer parcellation.

**Author response image 14. sa2fig14:** 

We then loop over the 200 parcels and extract the mean E-field (expressed in volts per meter) for every brain region. Author response image 15 shows a histogram of this raw normalized Efield map for every brain regions

**Author response image 15. sa2fig15:** 

Then, as reported in the manuscript, the normalized E-field distribution was thresholded at 83% of the maximal E-field value (Romero et al. 2019). This was done as shown in Author response image 16.

**Author response image 16. sa2fig16:** 

This procedure resulted in 5 nodes where the stimulus was injected, specifically the following Schaefer’s parcels:

LH_SomMot_6LH_SomMot_9LH_SomMot_10LH_DorsAttn_Post_4LH_SalVentAttn_ParOper_2

We thank the Reviewer again for pointing this out and for giving us the opportunity to provide more details.

5. Can the authors justify why TMS is modeled as perturbing specifically the excitatory interneuron population?

Our implementation, where input is inserted into the excitatory interneuron population, follows the convention set by the original Jansen-Rit model (see e.g. variable *p(t)* in Equation 6 of Jansen and Rit, 1995; variable Cu^U^ Figure 1 and Equation 2 of David et al. 2006). Given the number of novel aspects to the present work, we felt that following conventions such as this were particularly important to ground our analyses in the extant body of work. We do however agree that whether excitatory interneurons should be the sole TMS target population open is very much an open question for the field moving forward.

It is also a question that we have further evaluated in our next (currently ongoing) study using this model, by comparing alternative options (e.g. input to excitatory interneurons, to pyramidal cells, to all populations). Thus far we have found relatively little difference in the overall modelled TMS-EEG response for stimuli injected into different target populations within the JR circuit. What this indicates is that at the local level, stimulation spreads to all three populations within a given node fairly quickly, and then the transmission to other nodes in the network (which always proceeds via the pyramidal cells) proceeds similarly regardless of the specific subpopulation that was targeted.

6. What's the reasoning for using the matrix laplacian of the connectivity matrix?

We have addressed this point in our response to Editor point #3, above.

7. Equation (4), it appears that Su(t) should be S(v) or S(v(t)). And why the t=0 boundary? I suspect a typo in the denominator too (I think it should be 1+exp instead of 1-exp).

Thanks, it was indeed a typo. We have corrected that.

8. Equation (11), bracket missing.

Thanks we have corrected that.

9. Here the modeled EEG signal is estimated from the difference between the excitatory and inhibitory interneuron activity in each region, which at first sounds at odds with the generally accepted view that EEG is most related to pyramidal neurons. I suspect this choice is because in the Jansen Rit model it is those outputs of the interneurons that drive the pyramidal neurons. This should be clarified.

The Reviewer is correct here, we take the post-synaptic potential on to the Pyramidal cell population (i.e. the Pyramidal cell somato-dendritic activity) as the EEG source, consistent with the generally accepted biophysical basis of the EEG signal, as well as common usage of the Jansen-Rit model (e.g. (David et al. 2006)). In the Jansen-Rit model, this PSP is defined by the inputs to that population, namely the excitatory and inhibitory interneuron populations, which are differenced due to their opposite polarities. We have clarified this in the text at Line 636.

10. lines 647-649, it is unclear how Equation (13) is a measure of complexity; also a typo referring to it as Equation (14).

Thanks for pointing this out. Equation (13) is a version of L2 regularization, which is also called regularization for simplicity, and it is used to deal with overfitting. It considers the model complexity as a function of weights, and therefore as proportional to the absolute value of its weight. The L2 regularization works by forcing the weights toward zero (without making them exactly zero), removing a small percentage of weights at each iteration.

As for the Line 647-649, it was indeed a typo. First of all, we apologize for that, and secondly we promptly corrected in the manuscript.

11. Figure 4C, what's the inset scalp picture?

We thank the Reviewer for giving us the opportunity to clarify this. The red dots indicate the electrodes from where the Local mean field potentials (LMFP) was extracted from. This is a standard representation, but we have now added a clear statement of this in the Figure’s caption.

12. Figure 4C, if this is Local Mean Field Power how can it go negative? And why does it have units of microvolts?

Thanks for pointing this out. We wish first to specify that the Local Mean Field Power (LMFP) was not used in any of the statistics that we ran. Indeed, we used the LMFP in Figure 4D (previous 4C) only for addressing and reinforcing the concept that signal propagation is highly affected by significant lesions of the structural connectome. We have corrected the information as power should be amplitude^2.

13. Supp. Figure S4 uses different symbols for the parameters vs the main text.

We apologize but unfortunately we were not able to find the inconsistencies the Reviewer mentioned. We have decided to add a corresponding Table S1 to the supplementary section, with all the final posterior mean and variances values for every subject where we used the same symbols of Figure S4.

14. Many of the figures pack in so much detail that it is difficult to see what is going on without excessive zoom to read labels etc. Some labeling should be larger, and some figures should potentially be split into multiple if there's no space to increase the size of details within.

We thank the Reviewer for pointing this out. Following the flow suggested by his/her comment we have done our best to increase the readability of the figure labels.

We would like to point out that it was hard to summarize all the complexity of this paper through the figures. Indeed. conversely from most of the publications when only 1 (maximum 2) different techniques are combined, here we have used a lot of expertise/data coming from multimodal neuroimaging, brain stimulation, machine learning, whole brain models, TMS E-field modelling, parameters optimization etc.

We would like to avoid splitting the images into multiple Figures (if possible) because we personally believe that the figure structure fits well with manuscript narrative. Breaking down some of the figures in our opinion would make the overall story scatter and in some sense disconnected with the risk to confound the reader further.

However, we do agree with the Reviewer, and for this reason we made an effort to increase the size of the label (of at least 2 pnts and where possible 5 or 6 pnts), and reorganize the figure most of the time.

15. There should be a space between a number and its units.

We have edited this in the manuscript as well as in every figure.

Reviewer #3 (Recommendations for the authors):I really enjoyed reading this manuscript. I found it interesting and well-conceived. Below the authors can find a list of major and minor issues (along with possible suggestions). In case the authors will properly address them, I would really recommend the publication of this otherwise excellent study on eLife.1) As clearly stated in my public review, most of my issues derive from the data used in the manuscript. My suggestion is to use the open dataset that came with Fecchio, Pigorini et al. (Plos One, 2017), including parietal, premotor, prefrontal, and M1 stimulation. The dataset is available at:https://figshare.com/articles/dataset/Data_for_Fecchio_Pigorini_et_al_/4970900/1Using this dataset:i) You could consider cortices other than M1;ii) You use data collected with customized noise masking, that minimizes residual auditory evoked potentials (i.e. n100-p200 component, as in Figure S1, Subjects 5, 6, 7, 8, 15… etc);iii) You use data that maximize the impact of TMS on the cortex;iv) You can be sure that residual sensory artefacts are maximally reduced during data collection.For all these reasons, I would really recommend using the above-mentioned dataset (or a similar one). This will be enormously important also for the TMS-EEG community. Indeed, it will allow demonstration, thanks to your modelling approach, that TMS-EEG data can contain late components even in the complete absence of residual sensory artefacts. I understand that re-running the entire analysis over another dataset is an enormous amount of work, but I strongly believe that it is fundamental for a proper interpretation of the results, and to make your study useful for the entire TMS-EEG community.

We are grateful to the reviewer for these constructive recommendations. We do agree with the observation that the dataset we have may suffer from some contamination by residual auditory evoked potentials. We refer the Reviewer to our response to the Editor’s re-statement of these concerns above, where we have conducted all of the requested new analyses, replicating and verifying our primary results.

2) By applying your modelling approach to such a dataset you should also be able to replicate the results reported in Rosanova et al. (JNeurosc, 2009) and Ferrarelli et al. (Arch Gen Psychiatry, 2012) showing that the stimulation of different areas leads to TEPs characterized by different "natural frequencies". Reproducing this feature in simulated data is fundamental to check the quality of simulated data – as a note I would indeed recommend using the natural frequency instead of the Perturbational Complexity Index to evaluate the quality of the simulated data (see also point 5 of the public review), and perhaps PCI to evaluate changes in complexity upon virtual lesion (as in Rosanova, Fecchio et al., Nat Comms, 2020). As a note, I am not 100% sure that a connectome-based approach that accounts for only a few local properties (Jensen and Rit equations) can be able to reproduce the Natural frequency. If it were the case, it would be an incredibly valuable readout of the efficacy of your method; if not, it means that this is just a first step that needs to be perfected (and this can be easily discussed).

We agree with the Reviewer that the ‘natural frequencies’ observations of Rosanova et al. are an interesting and potentially important feature of TMS-evoked EEG responses. As noted above in response to the Editor however, we disagree that natural frequency responses are a critical feature for novel models of TEPs to reproduce, for the simple reason that (as opposed to eg the time series waveform and component structure trial-averaged TEP) these are not a widely studied and widely replicated experimental result. We do look forward to investigating questions around natural frequency responses in future work.

3) Related to my point (6) of the public review; if possible and not computationally challenging, I would perform more than 3 lesions over time. Instead of 20, 60, and 100ms, I would try to perform a more time-resolved analysis with one lesion every 5/10 ms, up to 200-300 ms.

We thank the Reviewer for his suggestion and for proposing this interesting analysis to better disentangle the TMS-induced propagation pattern. For this reason, even though it was indeed quite computationally challenging, we have followed the flow suggested by the Reviewer’s comment, and have run the lesion analysis again for one subject with one lesion every 10ms, up to 200ms after the TMS pulse. The results of this analysis are detailed above in our response to the Editor’s question #24.

4) It is not very clear how the authors performed the statistical comparison between real and simulated data.

We thank the Reviewer for his question and for giving us a chance to clarify how we performed the statistical comparison between real and simulated data.First of all, we have performed this comparison both in channel and source space. As for channel-level we have computed the Pearson correlation coefficients and corresponding p-values between empirical and model-generated TEP waveforms. We have used Pearson correlation coefficients also as the objective functions during the fitting process for computing the loss between empirical and simulated times series.

**Author response image 17. sa2fig17:** 

Then, in order to control for type I error, this result was compared to a null distribution constructed from 1000 time-wise random permutations, with a significance threshold set at p < 0.05. Please see Author response image 18:

**Author response image 18. sa2fig18:** 

Importantly in the cell, as the Reviewer can see, we have permuted both across subjects and across channels. We have then collected these “fake/surrogate” Pearson correlation coefficients and corresponding p-values, and created a distribution. Finally, we have used this distribution for determining the cluster threshold as the 95th percentile of the cluster's surrogate distribution.Furthermore, in channel space, we also examined more holistic time series variability characteristics using the PCI (Casali et al. 2013), which was extracted from the simulated and the empirical TMS-EEG data, and Pearson correlations between the two computed. This was a complement to these TEP comparisons that emphasize matching of waveform shape and component timing.

All this information is reported in the manuscript as follows:

Line 664: “At the channel level, Pearson correlation coefficients and corresponding pvalues between empirical and model-generated TEP waveforms were computed for each subject. In order to control for type I error, this result was compared with a null distribution constructed from 1000 time-wise random permutations, with a significance threshold set at p < 0.05. As a complement to these TEP comparisons that emphasize matching of waveform shape and component timing, we also examined more holistic time series variability characteristics using the PCI (Casali et al. 2013), which was extracted from the simulated and the empirical TMS-EEG data, and Pearson correlations between the two computed.”

As for source space comparison, we have used the same statistical framework (e.g. permutations and cluster correction) but this time in vertex space. Moreover we have also compared network-level time series from the simulated and empirical data extracted using the 7 Freesurfer surface-projected canonical Yeo network maps (Yeo et al. 2011).

Please find below the corresponding line in the manuscript:

Line 680: “Finally, and unlike the channel-level data, network-level comparisons of simulated vs. empirical activity patterns were made by averaging current densities over surface vertices at each point in time within each of the 7 Freesurfer surface projected canonical Yeo network maps (Yeo et al. 2011), and Pearson correlation coefficients and p-values between empirical and simulated network-level time series were again computed.”

We thank the Reviewer again for giving us the opportunity to clarify this point and we hope that we succeeded in this.

In addition, in figure 1A, empirical and simulated TEPs exhibit extremely different voltages, so simulated TEPs appear to be remarkably larger than empirical TEPs. How do the authors explain this discrepancy? Since the metric employed by the authors (correlation) is independent of the absolute value of the voltage, it would be important to quantify and report these differences.

Please see response to Editor’s Question #10 above.

5) This is more of a curiosity: what happens to high frequencies (>20Hz) after the lesion? Do you see any suppression as in Rosanova, Fecchio (Nat Comms, 2018)?

We thank the Reviewer for their perceptive comment and for raising this interesting question. In accordance with their suggestion, please see the plot in Author response image 19:

**Author response image 19. sa2fig19:** 

As can be seen, we have mainly found a suppression for the α frequency compared to the condition of no damage. Although this topic certainly falls within our general areas of interest, we do however believe that adding this as a main result to the present manuscript would be out of the scope. We will definitely return to this question in future work with the improved model version that we are currently working on. It would also be interesting to use the dataset the Reviewer suggested for implementing this analysis in another prospective publication. For this suggestion and input we wish to thank the Reviewer, as this comment will come in handy for other analyses we are currently running in our lab.

[Editors' note: further revisions were suggested prior to acceptance, as described below.]

The manuscript has been improved but there are some remaining issues that need to be addressed. These issues are outlined in the Reviewer comments provided below. These need to be addressed comprehensively before we can consider publication of the manuscript.Reviewer #1 (Recommendations for the authors):The authors provided a significantly improved revised manuscript and with most of their responses to my comments. I am very happy. However, there are still a few major and minor points that I suggest the authors take a closer look at.

We appreciate the Reviewer’s positive comment and for expressing appreciation on the changes and the new analyses we have performed. We would also like to thank the Reviewer for his/her thoughtful comments and efforts in further improving our manuscript. As she/he will see, we have followed his/her suggestions closely, and carefully revised our manuscript to address the few major and minor points raised. We believe that our revised version is further improved in accordance with these suggestions.

General point:Often, the authors nicely provide detailed explanations in the rebuttal but fail to also add those explanations to the manuscript. If these refinements were also added to the improved manuscript, then future readers could benefit from the improved clarity directly without additionally having to consult the rebuttal letter. An example is essential revision 3, where it would be great if the authors could also clarify in the text that they used the Frobenius norm.

We wish to express our gratitude to the Reviewer for his/her positive evaluation of our responses. We have added the requested responses, plus some additional points (please see below):

Line 559: “Finally, this matrix was prepared numerically for physiological network modelling by rescaling values by first taking the matrix Laplacian, and second by scalar division of all entries by the matrix norm, which ensures that the matrix is linearly stable (i.e. all eigenvalues have negative real part except one eigenvalue which was zero). The Laplacian sets each row sum (i.e. each node’s weighted in-degree) to zero by subtracting row sums from diagonal, and has been used often in previous whole-brain modelling research (Abdelnour et al., 2018; Atasoy et al., 2016; Raj et al., 2022).”

Line 525: “The whole-brain model we fit to each of the 20 subjects’ TMS-EEG consists of 200 brain regions, connected by weights of the anatomical connectome. We set strong priors on the connection weights, such that individual fits allow for small adjustment of these values.

Specifically, a prior variance of 1/50 was set for every connection, which was determined empirically to provide a strong constraint but allow for some flexibility during the fitting process. Importantly, after fitting every subject, visual inspection was conducted to confirm that the (posterior mean) connection weight matrices retained the key topological features present in empirical neuroimaging measurements.

To obtain population-representative values for these connectivity priors, we ran diffusion weighted MRI tractography reconstructions across a large number of healthy young subjects and averaged the results.”

Line 664: “The algorithm proceeds by dividing a subject’s multi (in this case 64) -channel, 400ms long (100ms to +300ms post-stimulus), trial-averaged TMS-EEG TEP waveform into short (20 ms) nonoverlapping windows, termed batches. The total length of the simulation was 400ms which included -100ms of baseline (pre-TMS) and +300ms after stimulation injection. Prior to the start of the 100ms baseline, a 20ms burn-in was included, which allowed for the system to settle after the usual transient due to randomly selected initial conditions for the state variables.

Rolling through each batch in the time series sequentially, the JR model-simulated TEP ŷ was generated with the current set of parameter values, and its match to the empirical TEP y was calculated with the following mean-squared error (MSE) loss function.”

Essential revision 2:– I fail to recognize the distance-dependence in the subject-specific SCs. There are many strongly red-colored entries, not just the ones around the diagonal. It seems similar in the original SC, thus I would not take this as a key feature here to judge the fitted SCs.

We apologize if the Figure that we attached in the previous round of revisions was not clear. Please find below the same Figure which includes the matrices of the posterior mean structural connectivity weights for all subjects from our analysis. As the Reviewer might notice we have replace “coolwarm” color scale with “viridis”. This prevents to have blue for small +ve numbers and white for some arbitrary small number above 1 (like it was before). From this new Figure it should be more clear that the SC’s parameter estimates differ only very slightly across subjects in their overall spatial structure.

We have included this information in the revised supplementary materials section.

– In the visualization of the original SC matrix (Appendix 2- Figure 4), one can see that the range of values is [0:0.003], thus allowing a variance of 1/50=0.02 seems very extreme in my opinion.

We thank the Reviewer for raising this point that gives us the opportunity to correct a typo in the text. Indeed 0.02 does not represent the variance on the structural connectome weights (which as the Reviewer pointed out would have been very extreme), but rather the variance on the gain parameter that encodes the deviation from the prior value, computed with the element wise matrix product SC_updated_ = SC_prior_ * exp(gain). This is implemented in the code at the following location:.

https://github.com/GriffithsLab/PyTepFit/blob/main/tepfit/fit.py#L333

As the Reviewer might seen from the code line (also reported below),

w = torch.exp(self.w_bb) * torch.tensor(self.sc, dtype=torch.float32)

Basically 0.02 variance was applied on the connection gain (variable name w_bb)and not on the structural connectome (variable name sc). If the prior mean of w_bb was 0, the most deviation of w_bb is about 2*std which is 2*sqrt(0.02)=0.28. Thus after taking exponential exp(0.28)=1.32, the maximum possible change is about a 32% of the original SC. Thus, our choices here represent strong priors on the SC matrix posterior values.

We have added this information to the manuscript (please see below):

Line 526: “We set strong priors on the connection weights, such that individual fits allow for small adjustment of these values. Specifically, a prior variance of 1/50 was set for every connection, which was determined empirically to provide a strong constraint but allow for some flexibility during the fitting process. This prior variance was applied to the connection gain and not directly to the connectivity weights, preventing the final fitted structural matrix to differ much from the original. Importantly, after fitting every subject, a visual inspection was conducted to confirm that the (posterior mean) connection weight matrices retained the key topological features present in empirical neuroimaging measurements.”

We hope that this is much clearer right now and thanks again for giving us the opportunity to provide a better explanation.

– Also for better comparison of the original SC with the subject-specific SCs, could the authors please use the same color bar limits for all SCs?

We have included a brand new Figure 4 in the new Appendix 2.

– If the authors allowed such a 1/50 variance, did they only allow this variance in the positive direction or did their prior distribution also allow for negative entries in the SC?

Thanks for this comment. No, negative SC entries are not permitted in our scheme. As noted in our response to the Reviewer’s previous comment, connection weights are updated using SC_updated_ = SC_prior_ * exp(gain). This is a standard formulation in parameter estimation schemes that ensures the numerically desirable property that values near zero in the gain matrix exert minimal change (since exp(0) = 1), and also that values cannot go negative (since the range of the function exp(x) is [ 0, +infinity]).

– "We selected a prior variance for each of these connections of 1/50, which was determined empirically to provide a strong constraint but allow for some flexibility during the fitting process." What exactly is meant with „determined empirically" here?

Thanks, we acknowledge that this terminology may be unclear in the present context. By ‘determined empirically’ here we simply mean that we selected this number as one that was found to work well in terms of balancing constraint and flexibility through the manual, partially exploratory process of developing the model and algorithm. The variance is a model hyper-parameter and it is learned to experience during the fitting process. We know from previous studies in general what is the parameters range of the Jansen and Ritt model and how the optimization algorithm works which give us an insight on how to set up the prior/initial conditions on mean and the variance. Such priors simply determine how far you want the fitted SC from the original SC each step. In our case, as specified before, we added this variance to the connection gain and not directly to the SC. Moreover by adding the exponential function in front we assure that all the values will be close to 1 meaning that the final SC was not diverging that much from the original prio

– "It can be seen here that the distribution of the norm distances scores (see image below) acceptable, indicating a relatively small deviation from the prior connectivity weights." It would be much better if the authors describe in words what they calculated here instead of posting the code as a justification of acceptability of the norm distance scores.

We apologize – in general we find posting the code to be most helpful in specifying exactly what we have computed, but we appreciate that this is not necessarily the case for all readers.

We have calculated the norm distances between the original (initial condition) and the individually fitted SC matrices as the Frobenius norm between the upper triangles of the two matrices.

Essential revision 4: The x axis ‚brain regions' should probably be relabeled and also add numbers to the ticks please.

We apologize if the figure was confusing. Since it was not a figure included in the manuscript we haven’t added numbers to the x-ticks thinking that it was clear enough. See Author response image 5.

Essential revision 6: I applaud the authors for providing such a table with individually optimized parameters, which is an important step towards full reproducibility of their study.

We are delighted and flattered by the Reviewer’s positive evaluation of our efforts to maximize open science and reproducibility which represent sensitive aspects of our daily work in the lab. Thanks again!

– In the table, there is only one column of variances added for each variable, was this the variance of the prior or posterior distribution or were they the same?

We wish to express our gratitude to the Reviewer for pointing this out. The priors in the previous version of the table represented the prior means, for which we used the parameters for the classic Jansen and Rit (1995) paper. However, as the Reviewer corrected reported, there was missing information regarding the prior variance. For this reason we have added another column to the table for every parameter (please see Supplementary file 1).

We hope this is offer a comprehensive as well as a reliable source in order to promote reproducibility.

– How can there be a negative variance for the global gain?

Thanks for reporting this. We apologize, the values given for posteriors in the table were, erroneously, the posterior estimates before the final pass through the linear rectification (ReLU) function, which we employ in the optimization algorithm to prevent the values of parameters such as variance from going negative. In the code:

We initialize the Pytorch object for the ReLu here:

https://github.com/GriffithsLab/PyTepFit/blob/main/tepfit/fit.py#L704 Then we used that object across the lines below: https://github.com/GriffithsLab/PyTepFit/blob/main/tepfit/fit.py#L715

We have corrected this information in the table and we hope this helps to clarify this important point.

Essential revision 12:The 100 repetition times of their fitting algorithm show a large and to me alarming variability of optimized parameters and warrants further analyses. One solution to deal with this alarming variability would be to please always run the algorithm 100 times (at least 10 I would suggest) and take the mean of its distribution as the final optimized parameter in order to increase robustness of the presented fitting results.

We wish to thank the Reviewer for this important point and for giving us the opportunity to disentangle the variability in the optimized parameters. We respectfully disagree with the Reviewer’s comment, since we strongly believe that this variability is far away from being considered alarming. We refer the Reviewer to our comment below regarding ranges and posteriors for elaboration of this. It is also important to mention that, at the end of the day, the scope of the paper was testing the physiological origin of the recurrent activity induced by TMS for 4 different conditions (1 control with no lesion and 3 experimental where the lesion has been applied at different times). For this central comparison, the model parameters are equal in all the 4 different conditions (since we fit the data for no lesion first and then re-run with lesions applied), and so does not affect the results. Finally, we would also like to add in this regard that the for final fitted parameters (the ones reported in table 1 and that we used for running our simulations) we already took the average of the last 100 batches (an algorithmic detail which we have not emphasized previously), which is in fact similar to the Reviewer’s suggestion here. We have added this information to the manuscript (please see below):

Line 693: “When the batch window reaches the end of the TEP time series, it returns to the start and repeats until convergence. When the optimization was completed, the average value for every parameter was computed using the last 100 batches and then used for running the simulations. For an overview of all parameters used in the model, please refer to Appendix 2-figure 4 and Supplementary File 1. For a complete description of the parameter estimation algorithm, please see (Griffiths et al., 2022).”

We hope that the Reviewer would understand that re-running all the subjects for two independent dataset 100 times and re-doing all the Figures and statistical analyses from scratch would represent a huge amount of effort which is partially not in line with the aforementioned scope of the paper: testing the physiological origin of the TMS-evoked potentials. Moreover, the solution we have already implemented is similar to the one proposed by the Reviewer’s comment and should control for large variability in fitted parameters’ values. We hope that the Reviewer would embrace our vision also considering all the effort we made for re-doing the analyses using the other two datasets coming from Fecchio-Pigolini’s paper (low and high voltage motor-evoked potential data). We wish to thank again the Reviewer for their suggestions.

– "For completeness, the prior means for each parameter are also shown as vertical red lines. Shown below are parameter estimate distribution plots for a single subject, repeated 100 times. The vertical red line indicates the original parameters' value (the one in the manuscript) for the same subject." What do the red lines in the plots represent, the prior means or the parameter value chosen for this subject in the manuscript results?

We apologize that it was confusing. The red lines were the parameter value chosen for this subject so the average of the last 100 batches.

– "Reassuringly, these histograms are approximately normally distributed, and have a tight range of variation that spans only a few percent of the magnitude of each parameter" Even though the variation spans only a few percent of the magnitude of each parameter, the variation for this single subject (e.g. for parameter a: [94:102]) spans nearly over the whole range of optimal parameter values that were fitted over all subjects (e.g. for parameter a: [92:102], see Appendix 2 – Table 1) – or even a larger range than the one fitted over all subjects as in the case of c1. Thus, all results regarding individual variability should be interpreted with caution. The plot for c4 is probably mislabeled as no subject has optimal values in this range according to Table 1.

We thank the Reviewer for this comment, which indicates a close and thoughtful inspection of our results that is much appreciated. It is important to remember here that because we are employing Bayesian inference, the proper interpretation of the fitted parameters is as probability distributions, and that these are defined not only by the mean and support (i.e. range) but also their shape – how tightly distributed they are around the mean, as determined by the variance parameters. The fact that a small number of repeat runs arrive at an estimate of the mean that lie in the extremes of the ranges noted (e.g. [94:102] for a) is entirely expected given that the posterior variances are non-zero (i.e. there is some degree of spread in the posterior distributions). The Reviewer’s statement that ‘results regarding individual variability should be interpreted with caution’, made on the basis of the ranges in the figures and tables provided, is correct to the extent that the posterior distributions are uniform (i.e. flat), in which case the extreme values on the left and right hand side of the ranges would have equal significance to the mean. Whilst (from our experience) flat posteriors are not uncommon when Bayesian inference is employed in physiological modelling applications, this is not the case in our results. We therefore stand by our conclusion that the reported variability in estimated synaptic parameters \can account for differences the TMSevoked potentials across subjects, and given their physiological nature provide insight into mechanisms that are difficult to measure in-vivo in humans.

Regarding the Reviewer’s comment on plot c4, we wish to apologize and it was indeed incorrect. Please see Author response image 19 for the correct plot.

We hope this addresses the Reviewer’s comment and we wish to thank him/her again for giving us the opportunity to elaborate better on this aspect.

Essential comment 18:Results lack significance after Bonferroni correction, therefore they should in my opinion be moved to the Supplementary Material and not be mentioned in the main text as being significant.

We apologize but we respectfully disagree with the Reviewer’s comment. The fact that given that the results did not survive to Bonferroni correction does not necessarily imply that they should move to Supplementary Material. We explicitly stated in the manuscript that the results were not corrected and that the reader should critically interpret that:

Line 310: Interestingly, as shown in Figure 5D, by varying the optimal value of the inhibitory synaptic time constant, an increase in the amplitude of the first, early, and local TEP component; and a decrease of the second, late, and global TEP component was found. However, these significant results were not corrected for the Bonferroni corrected p-value of 0.007. For this reason the reader should critically interpret such results.

We apologize for disagreeing with Reviewer’s vision but we really hope he/she will allow us to keep that figure/results in the main manuscript because we firmly believe that section represents an valuable part of our work that can potentially have some value for future research even though is not corrected for multiple comparisons.

Reviewer #2 (Recommendations for the authors):The authors have done a good job in responding to the reviewer comments. In particular it is impressive that they successfully replicated the analysis on an independent dataset. Some residual comments:

We appreciate the Reviewer’s favorable evaluation on our manuscript. We updated our entire manuscript in accordance with the Reviewer’s residual comments

1. In general it is preferable that if a clarification is made to the reviewers, it is also made in some way in the text. I won't list them all, but for example the matrix laplacian clarifications don't appear to have made it to the text.

Totally agree and thanks for pointing this out. We have addressed this point in our response to Reviewer #1 point #3, above.

2. Re notation difference, figures and the appendix table refer to parameters a, b, c1, c2, c3, c4, g, but I do not see these in Equations 1-11.

Thanks for pointing this out and we apologize if there was some mismatches between the equations and the figure/table. We have corrected all of them.

3. Re Equation 4, Su(t) appears to be a typo since the other equations refer to the function S(v).4. Re Equation 4, why the t=0 boundary?

Thanks we have promptly corrected equation 4.

Reviewer #3 (Recommendations for the authors):First, I would like to congratulate the authors on the incredible amount of work done in order to address all the issues posed by myself and the other reviewers. In particular, I really appreciate the new set of analyses performed on a completely different dataset in order to control for possible confounding factors, such as residual sensory input. In general, the authors have addressed most of the issues I formerly raised, and significantly improved the manuscript with the revised version.

We are delighted by the Reviewer’s appreciation and consideration of our effort in addressing all the points raised in the previous round of revisions. We agree with the Reviewer’s comment and strongly believe that the manuscript has significatively improved and for that we wish to express our deep gratitude to the Reviewer because it would not be possible without his knowledge and expertise.

Yet, the first fundamental issue mentioned in my previous review has not been addressed (see point 19 of previous review). The authors mentioned in their response that they "believe that [testing different areas] is beyond the scope of the present manuscript…". I respectfully disagree. In my opinion, the interpretation of the results could be severely biased by the stimulation of one single area (primary motor cortex – M1), having strong efferent connections with periphery. To better clarify my point, by stimulating only M1 it is not possible to conclude that late TEP components are "driven by the recurrent activity within the rest of network" (as stated in the introduction); or at least this is not the only possible interpretation. Indeed, TMS of M1 – especially at Resting Motor Threshold (RMT), as in Biabani et al., or above RMT, as in Fecchio, Pigorini et al. (Experiment_1) – most often leads to muscle twitch, which in turn, induces proprioceptive (sensory) feedback from the periphery. So, TEP components under these experimental conditions are inherently confounded by the peripheral twitch induced by the stimulation. How can the authors address this problem?In my view, the most straightforward solution implies including other areas in the main analysis (as suggested in my previous review); however, I understand that this implies a huge amount of work, encompassing "diffusion-weighted MRI analysis, TMS-EEG analysis, e-field modeling, deep learning for parameter optimization and neural mass model implementations…". I also understand that "this question is among the scientific priorities" of the authors, which "are currently working on expanding this model further to other stimulation sites".So in my view, other two – less time consuming – possible solutions to disentangle this problem are:1) The authors could control whether TEP associated with high voltage and low voltage Motor Evoked Potentials are associated with different late components in the simulated data (data available in Fecchio, Pigorini 2017 – Experiment_2).2) The authors need to significantly rework the main message of the manuscript by incorporating this confound in the authors' interpretation of their main finding (e.g. the title should be changed because late components could be recurrent large scale network dynamics, but could be also feedback from the periphery…).I find this second option overly detrimental to the overall significance of the study and I strongly suggest the authors to implement the first in a revision.

We wish to thank the Reviewer for raising this important potential confounding factor. We hoped to have addressed this pivotal point in the previous round of revisions by rerunning the analyses in a completely independent dataset ((Fecchio et al., 2017)- https://figshare.com/articles/dataset/Data_for_Fecchio_Pigorini_et_al_/4970900/1).

As the Reviewer correctly pointed out, even though we were able to replicate the results we obtained with the Rogash group’s dataset, this did not allow us fully to reject the null hypothesis that our previous results were due to proprioceptive (sensory) feedback from the periphery. We thank the Reviewer for understanding and for sharing our perspective regarding including other areas in the main analysis, which in our opinion would have indeed been out of the scope of the current work. However, we agree that this point needed to be disentangled somehow, and we strongly believe that the Reviewer’s suggestion #1 is the best way to address this potential confounding factor. Indeed we are entirely on the same page as the Reviewer since we strongly believe that suggestion #2 would have decreased the impact of our manuscript.

For this reason we have re-run the analysis for the low voltage Motor Evoked Potentials as suggested by Reviewer’s comment.

Please see the new Appendix 2-figure 7 showing the Empirical (upper row) and simulated (lower row) TMS-EEG butterfly plots with scalp topographies for all 6 subjects of the Fecchio and Pigorini’s dataset – but this time using the low voltage data. Results show a robust recovery of individual empirical TEP patterns in model-generated activity EEG time series.

In accordance with the flow suggested by the Reviewer’s comment, we have also re-run the analysis for the virtual lesion approach using the low voltage data of Fecchio-Pigorini dataset. Please see Appendix 2-figure 9 (Panel B), which shows how removing recurrent connections from stimulated target nodes suppresses their late TEP activity in the same way as found for both the Rogash group dataset and the high voltage trials of Fecchio-Pigorini dataset (Panel A).

We edited the Appendix 2 Control analyses section accordingly:

“Appendix 2-figure 6, 7 shows empirical and fitted TEP waveforms and selected topography maps for the entire group of the Fecchio, Pigorini et al. dataset (Fecchio et al., 2017) using both high (Appendix 2-figure 6) and low (Appendix 2-figure 7) voltage motor-evoked potentials data.

Similarly to the analyses for the Rogasch’s dataset, it is visually evident in these figures that the model accurately captures several individually-varying features of these time series, such as the timing of the 50 ms and 100-120 ms TEP components, and the extent to which they are dominated by left/right and temporal/parietal/frontal channels.”

“Shown in Appendix 2-figure 8 are effects for one representative subject of the Fecchio, Pigorini et al. dataset (Fecchio et al., 2017) on the simulated TEP of virtual lesions to recurrent incoming connections of the main activated regions at 20 ms, 50 ms, and 100 ms after single-pulse TMS stimulation of left M1.

Similarly to the analyses for the Rogasch’s dataset, the network-level effects were evaluated statistically by extracting dSPM loadings from the source activity maps for each of the 7 canonical Yeo networks (Yeo et al., 2011) and entering them into an ANOVA with factors of NETWORK and TIME OF DAMAGE.

For the high voltage motor evoked potentials data (Appendix 2-figure 8A), a significant main effects were found for both TIME OF DAMAGE (F(6,30) = 229.87, p<0.0001, η2p = 0.97) and NETWORK (F(6,18) = 12.08, p < 0.0001, η2p = 0.71) – indicating that the effects of virtual lesions vary depending on both the time administered and the site administered to, as well as the combination of these factors (significant interaction NETWORK*TIME OF DAMAGE (F(3,18) = 10.26, p < 0.0001, η2p = 0.67)).

For the low voltage motor evoked potentials data (Appendix 2-figure 8B), a significant main effects were found for both TIME OF DAMAGE (F(6,30) = 204.07, p<0.0001, η2p = 0.95) and NETWORK (F(6,18) = 11.25, p < 0.0001, η2p = 0.68) – indicating that the effects of virtual lesions vary depending on both the time administered and the site administered to, as well as the combination of these factors (significant interaction NETWORK*TIME OF DAMAGE (F(3,18) = 9.58, p < 0.0001, η2p = 0.64)).”

The fact that the same results were found for both the high and the low voltage motor evoked potentials data indicates that the results observed are not affected by proprioceptive sensory feedback from the periphery.

We strongly believe that with this additional analyses we have re-run on the low voltage Motor Evoked Potentials helped us to reject the null hypothesis that the recurrence results we reported using the high voltage datasets (by Rogash and Fecchio-Pigorini) could be affected by proprioceptive sensory feedback from the periphery. We definitely agree that this suggestion made by the Reviewer improves the quality of the manuscript and for this we wish to express, once again, our deepest gratitude!

Apart of this main issue, I also found a couple of issues the authors may want to address before publication.1) Perhaps my previous comment about PCI was not clear enough. I did not say that "PCI should never be used for evaluating model fit"; and I fully understand that fitting "could be done with any metric at all that can be computed from empirical and simulated time series". However, while it is true that "the model may be considered a good one to the extent that it captures multiple data features of physiological interest that are reliably quantifiable", it is also true that some metrics are better than others to capture similarities among different time series. As mentioned in my previous review, "two completely different wave shapes could in principle have exactly the same PCI value". The authors can find an example for this in Decision letter image 1. Top and bottom panels report the butterfly plot (black) and the channel under the coil obtained by stimulating (in the same subject) occipital and prefrontal cortices, respectively. In both cases PCI is 0.47, but the wave shapes highlight major differences both in time and space which are not captured by the single PCI scalar. I hope this example clarifies my former concern and suggest the authors to report metrics other than PCI in the main figures (Pearson correlation coefficients are more than enough).

We thank the Reviewer for taking the time to elaborate more on his perplexity on using the Perturbational Complexity Index (PCI,(Casali et al., 2013)) as a metric for evaluating the similarity between real and simulated data. We absolutely agree with the Reviewer and thanks for providing an image with a compelling example of how PCI value can be misleading in evaluating the wave shape. That being said, we respectfully disagree that this should drive us to remove panel D from Figure 2. We believe that Figure 2 in its current state offers a compelling and comprehensive overview of the goodness-of-fit at the level of channels. Indeed, panel A shows 3 representative subjects waveforms for empirical and simulated time series. From their comments it seems the reviewer may have missed this, but Panel B does also show the Pearson correlation coefficients between simulated and empirical TMS-EEG time series for each subject. Panel C shows the corresponding stats (Time-wise permutation tests) of the Pearson correlation coefficient scores. Only Panel D reports PCI values. Moreover, in the Appendix 2-figure 1 we provided the empirical and simulated TMS-EEG waveforms for every study subject, reinforcing the demonstrations in Figure 2 that the model-generated EEG activity time series achieve robust recovery of individual subjects’ empirical TEP propagation patterns. The same strategy has been used for showing the goodness of fit for an independent dataset. We strongly believe that figure 2 in its current state provides a full overview of (1) TEP waveforms (Panel A and Appendix 2figure 1), (2) Pearson correlation coefficients (Panel B and C) and only at the end PCI scores which is a small (and marginal) part of Figure 2. We would agree with the Reviewer if we only have used PCI to justify and evaluate the goodness of fit but – as specified above – we have provided a comprehensive overview including multiple quantitative metrics as well as extensive qualitative visual assessments. As part of this combination, and given that it is a well-known measure in the analysis of TMS-EEG TEPs, we feel that including PCI as a small part of the goodness-of-fit assessment adds value to the analysis.

For this reason, we kindly ask the Reviewer to include PCI as a small part of Figure 2 (Panel D).

2) Something must be wrong with the axes in Figure S5. See Decision letter image 2 the authors can find a plot with the correct time and voltage scales for each subject. For example, in Subj_1 the first peak is a 4 ms and not at 38 ms (as in Figure S5).

We apologize and we also thank the Reviewer for pointing this out. It was only an issue with axis x that we have promptly corrected in the new Figure S5.

3) Related to my former comment on the original manuscript: "Figure 2B shows a correlation that can be partially driven by the spectral content of the waveforms. It could be relevant to verify whether these correlations are significant with respect to a comparable null hypothesis (e.g. IAAFT, or correlating different empirical and simulated sessions)." I mean that correlation analysis could be strongly affected by low frequencies (characterized by higher power). Thus, a statistical test accounting for different frequencies (e.g., IAAFT) could be preferable (yet not mandatory at this stage).

We wish to thank the Reviewer for further explaining this point that was not clear to us in the previous round of revisions, and we think it is a constructive comment. The goal of Figure 2 in the manuscript is to “prepare the field” for exploring the main subject of the paper: examining the origins of recurrent activity in TEPs. We do agree that correlation coefficients can be misleading at times if assumptions are made about equivalency of frequency content that are not checked. We believe we have provided enough information to confirm, at least informally, that there is not a clear discrepancy in frequency content between the empirical and simulated waveforms, as is clear from the comprehensive overview of TEP waveforms given in Figure 2 and the supplementary materials.

With this being said, we will take into account this possibility for further improving our model in future work, and perhaps such comparisons can be potentially included as a part of the model fitting process. We hope that the Reviewer would agree that at this stage implementing this further analysis would require a great deal of effort, in a direction outside the main focus of the paper, and as such would not impact its main message regarding recurrent activity.

References

Abdelnour F, Dayan M, Devinsky O, Thesen T, Raj A. 2018. Functional brain connectivity is predictable from anatomic network’s Laplacian eigen-structure. *NeuroImage* 172:728–739. doi:10.1016/j.neuroimage.2018.02.016

Atasoy S, Donnelly I, Pearson J. 2016. Human brain networks function in connectomespecific harmonic waves. *Nat Commun* 7:10340. doi:10.1038/ncomms10340

Casali AG, Gosseries O, Rosanova M, Boly M, Sarasso S, Casali KR, Casarotto S, Bruno M-A, Laureys S, Tononi G, Massimini M. 2013. A Theoretically Based Index of Consciousness Independent of Sensory Processing and Behavior. *Sci Transl Med*. doi:10.1126/scitranslmed.3006294

Fecchio M, Pigorini A, Comanducci A, Sarasso S, Casarotto S, Premoli I, Derchi C-C, Mazza A, Russo S, Resta F, Ferrarelli F, Mariotti M, Ziemann U, Massimini M, Rosanova M. 2017. The spectral features of EEG responses to transcranial magnetic stimulation of the primary motor cortex depend on the amplitude of the motor evoked potentials. *PLOS ONE* 12:e0184910. doi:10.1371/journal.pone.0184910

Griffiths JD, Wang Z, Ather SH, Momi D, Rich S, Diaconescu A, McIntosh AR, Shen K. 2022. Deep Learning-Based Parameter Estimation for Neurophysiological Models of Neuroimaging Data. doi:10.1101/2022.05.19.492664

Raj A, Verma P, Nagarajan S. 2022. Structure-function models of temporal, spatial, and spectral characteristics of non-invasive whole brain functional imaging. doi:10.3389/fnins.2022.959557